# Representing part-whole hierarchy with coordinated synchrony in neural networks

## Abstract

Human vision flexibly extracts part-whole hierarchy from visual scenes. However, how can a neural network with a fixed architecture parse an image into a part-whole hierarchy that potentially has a different structure for each image is a difficult question. This paper presents a new framework to represent the part-whole hierarchy by the hierarchical neuronal synchrony: (1) Neurons are dynamically synchronized into neuronal groups (of different timescales) to temporarily represent each object (wholes, parts, sub-parts, etc.) as the nodes of the parse tree. (2) The coordinated temporal relationship among neuronal groups represents the structure (edges) of the parse tree. Further, we developed a simple two-level hybrid model inspired by the visual cortical circuit, the Composer, which is able to dynamically achieve the emergent coordinated synchronous states given an image. The synchrony states are gradually created by the iterative top-down prediction / bottom-up integration between levels and inside each level. For evaluation, four synthetic datasets and three quantitative metrics are invented. The quantitative and qualitative results show that the Composer is able to parse a range of scenes of different complexities through dynamically formed neuronal synchrony. It is promising that the systematic framework proposed in this paper, from representation and implementation to evaluation, sheds light on developing human-like vision in neural network models.

## 1 Introduction

Representing hierarchical structure is a key problem for neural networks. While there is strong evidence in psychology that people parse a visual scene into part-whole hierarchies with many different levels (e.g. scene level, object level, part level, sub-part level, sub-sub-part level, etc.) (Hinton, 1979; Kahneman et al., 1992; Thompson, 1980), the representation and manipulation of part-whole hierarchical information in fixed hardware is a profound challenge for artificial neural networks (Hinton, 2021). On the other hand, constructing such a part-whole hierarchy enables the neural networks to understand the visual scenes in a compositional way like human (Hinton, 2021), and facilitates the interpretability of the network representation (Garau et al., 2022).

The part-whole hierarchy is an inclusion relationship and is conceptually organized as a parse tree since each part object (child node) should belong to a single whole object (parent node) (Hinton, 2021). Such compositional structure of multiple simultaneously presented objects of different levels profoundly complicates the problem of visual perception (Fig.1 b). More importantly, the structure of the parse tree could switch among multiple reasonable forms even given a single scene and is likely to dynamically reform itself on the fly when the scene changes. Such dynamical and multi-stable nature challenges neural networks of fixed architecture (Hinton, 2021). Moreover, it renders simple feedforward networks (Deng et al., 2021) and supervised learning unlikely to ultimately conquer the problem (Greff et al., 2020).

The challenge of the problem could be decomposed into three aspects: First (Node), how to dynamically group information that is distributed in neural networks to form each object representation that potentially acts as tree nodes? Second (Levels), how to distinguish node representations into the whole level and part level? Third (Edges), how to specify the relationship among whole-object representation and part-object representation as the edges in the parse tree. It is notable that when parsing different images, the three aspects should be achieved while keeping the network structure unchanged, e.g. the number of neurons.

To solve the problem, we seek inspirations from the brain: First (Nodes), neuronal synchrony is exploited to dynamically group distributed information into object representation (Malsburg, 1994; Singer, 2007) in a wide range of regions of the brain, so-called cell assemblies (Palm, 1982; Buzsáki, 2010; Camera et al., 2019; Miehl et al., 2022) (Fig.1d, colored shadows). Second (Levels), the neocortex is spatially organized into hierarchical levels of columns (V1, V2, etc, Fig.2g), potentially corresponding to the levels of part-whole hierarchy (Gross et al., 1972; Gross, 2002; Tsao et al., 2006; Hinton, 2021). In other words, the level is explicitly distinguished by spatial separation (Fig.1d different colors along the y-axis and Fig.2f,g). Third (Edges), the temporal structure of neuronal activity (cell assemblies and neuronal oscillations) is organized into a 'timescale hierarchy' (Manea et al., 2021), of different frequency bands (Buzsáki & Draguhn, 2004), along the cortical hierarchy (Mahjoory et al., 2019). Moreover, the timescale hierarchy (from milliseconds to seconds) is related to information of hierarchical levels (e.g. words to sentences) in the neocortex and the transient nestedness (coordination) of different timescales indicates the presence of consciousness (Northoff & Huang, 2017b). Therefore, the nested relationship among parts and wholes (Fig.1b) could be represented as the nestedness among synchronized neuronal groups of hierarchical timescales in neural networks (Fig.1d).

In this paper, we systematically study how to represent the part-whole hierarchy in neural networks through coordinated synchrony, from representation (framework) to implementation (model) and to evaluation (dataset and metric). We first develop a novel framework to deal with part-whole hierarchy at the representation level, where each object is represented as synchronized neuronal groups and the hierarchical relationship among objects is represented as the nestedness (coordination) among neuronal groups of different timescales. Then, at the implementation level, We provide a cortical-circuit-inspired model, called the **Composer** (short for **CO**rtical-like e**M**ergence of **P**art-wh**O**le relation**S**hip through n**E**uronal synch**R**ony) to show how the hierarchical synchrony state is emerged given an input image. The Composer integrates spike timing dynamics into a deep learning framework to exploit the core advances of both sides. More specifically, the Composer consists of two levels of columns and each column contains a visible spike coding space (SCS), which is delay coupled by a denoising autoencoder(DAE) (Vincent et al., 2008). The coordinated synchrony is reached through iterative top-down prediction and bottom-up integration within each level and across different levels. In order to understand the representation of the Composer, four synthetic datasets of different complexities and three metrics to measure different aspects of the part-whole representation are invented to explicitly evaluate the emergent neuronal activity. Quantitative results and qualitative visualization confirm the validity of the Composer and the plausibility of the framework. Lastly, for comparison, we show that the Composer outperforms the SOTA, the Agglomerator (Garau et al., 2022), when the representation for the part-whole hierarchy is explicitly evaluated. **The main contributions** are as follows:

(1) We developed a bio-plausible framework to deal with the part-whole hierarchy at the representation level.

(2) We developed the Composer, integrating both deep learning (denoising autoencoder and self-supervised learning) and neuroscience (spiking code, dendritic computation, and rhythmic dynamics) to show how the coherent state emerges to represent the part-whole relationship.

(3) We invented four synthetic datasets and three quantitative metrics to explicitly interpret the learned representation, which also shows that the Composer outperforms the recent state-of-the-art model.

## 2 FRAMEWORK AND INTUITIONS

In this section, we develop the framework of how to represent the part-whole hierarchy with synchrony and provide intuitions to understand how the neuronal synchrony emerges, step by step.

### 2.1 REPRESENTATION

Firstly, neurons have receptive fields that are selective for different features of objects; Secondly, the set of neurons responsive to features of the same object are dynamically synchronized into the neuronal group to form the object representation (Fig.1d, y-axis and colored shadows). Third, neurons are explicitly distinguished into columns of different levels (part/wholes, Fig.2f,g) and each column contains neurons that represent the objects at the respective level. Columns in higher levels have

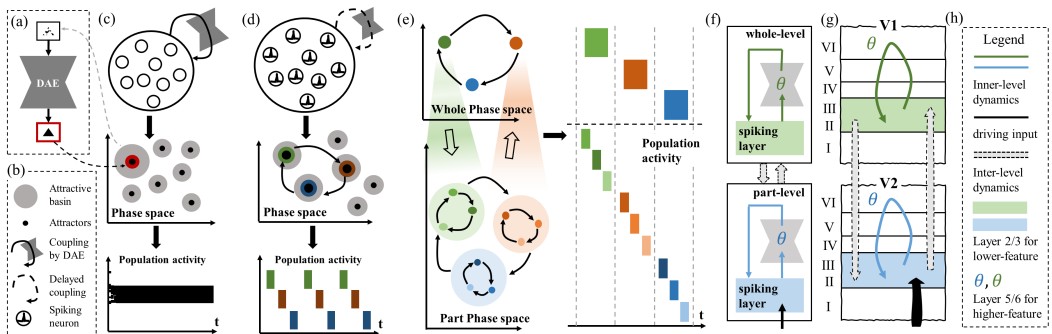

Figure 1: (a) The visual scene of a house. (b) The mental parse tree of the visual scene. (c)(d) Representing the parse tree as emergent neuronal synchrony. Synchronized neuronal groups are indicated by colored shadows in (d). Colors stand for levels in both (b) and (d). Neurons are indicated by selectivity along y-axis in (c),(d).

Figure 2: How coordinated neuronal synchrony emerges. (a) Denoising autoencoder (DAE). (b) Legend for (c)(d). (c) Building up attractor dynamics by DAE (top and middle), which results in stationary population activity (bottom). (d) Building-up metastable rhythmic dynamics when spiking neurons and delay coupling show up. (e) The phase space (left) and population activity (right) of the whole system. Attractive basin is not shown for clarity. (f) General architecture of the Composer, which is highly inspired from the visual cortical circuit shown in (g). (h) Shared legend of (f)(g).

longer timescales so that synchrony events are much sparser(Fig.1d). Fourth, the temporal inclusion relation (nestedness) of neuronal groups represents the inclusion relation among parts and wholes (see nested colored shadows in Fig.1d and colored nodes in Fig.1b), so that the part-whole hierarchy is represented as coordinated neuronal synchrony.

## 2.2 INTUITIONS OF THE MECHANISM

But how could the coordinated temporal structure emerge in a neural network given a visual scene? The intuition starts from the close relationship between the denoising autoencoder and the attractor dynamics. As shown in Fig.2a, denoising autoencoder (DAE) denoises noisy patterns. If it is exploited to parameterize a recurrent neural network so that $x_{t+1} = DAE(x_t)$, noisy pattern $x_0$ in the neighbourhood of original pattern $x$ will be 'attracted' to the original pattern by the recurrent dynamics (Fig.2c). $x_t$ is the network state at time step $t$. Therefore, a large number of attractors are explicitly embedded into the network dynamics by DAE. However, the attractor dynamics results in stationary population activity of a single set of active neurons (Fig.2c, bottom).

To represent multiple objects, the network needs to be metastable and it is where spiking dynamics and delay coupling show up. As shown in Fig.2d (top), if the neurons become spiking, their refractory period will prevent persistent firing so that the attracted states become transient. The delay coupling renders the dynamics non-Markovian and provides the essential time window for attracted states to switch (Fig.2d middle) so that the population activity becomes non-equilibrium and oscillatory (Fig.2d bottom). In other words, the synchronized neuronal groups to represent objects are transient attractive states of the network dynamics in nature. The same mechanism could be exploited to build

up neuronal groups in both part and whole levels. The subtle difference is that the whole level has longer timescales so that its dynamics is slower than the part level.

Provided that appropriate metastable dynamics is created in both part and whole levels so that all candidate object representations for the node of the parse tree are at hand, it is also important to coordinate the neuronal groups of the two levels to shape the parse tree. The general picture is that whole-level states condition the part-level states, e.g. by gating effect (Fig.2e, left, green arrow), so that during the 'lifetime' of each slow whole-level neuronal group (transient attractive state), corresponding fast part-level neuronal groups switch with smaller timescales (Fig.2e,left). On the other side, the temporal integration of part-level activity smooths out the finer-grained details and can in turn reinforce the whole-level states (Fig.2e, left, orange arrow). In a nutshell, vision in the Composer is a sampling process on an imagined energy landscape, which is shaped by both DAE and biological constraints like refractoriness, delay coupling, top-down gating, and bottom-up integration. The overall effect is to enforce the coordinated synchrony states as the local minimums of the entire dynamical system so that it is searched along the iterations. Once searched, hierarchical neuronal synchrony emerges as the population activity (Fig.2e, right).

## 3 MODEL

Overall, the Composer consists of two levels of columns, interconnected by top-down modulation and bottom-up integration (Fig.2f). Each column contains a visible spiking layer, named spike coding space (SCS, Fig.3c), which is delay coupled by respective DAEs. The SCS of both levels has the same dimensions as the image ($d$), corresponding to the topographical mapping in neocortex (Kaas, 1997).

The general architecture is inspired by the circuit organization in the visual cortex (Fig.2g). As shown in Fig.2fg, layer 2/3 in the cortical column encode low-level features with sparser firings while layer 5/6 encode higher-level features with denser firings. The former is modeled as superficial spike coding space (SCS) and the latter is modeled as the real-valued latent space of DAE. Besides, the bottom-up integration and top-down modulation 'inside each column' are modeled as encoders and decoders of the DAEs (Fig.2fg). More specifically, bottom-up integration is sensitive to spike timings, so-called coincidence detectors (König et al., 1996), which is modeled as an integrative function $I$ (Fig.3f) before feeding SCS activities into DAEs. Besides, the top-down feedback modulates the activity of pyramidal cells in layer 2/3 by acting on distal synapses (away from the soma) (Sherman & Guillery, 1998). These dendritic computations are modeled as simplified pyramidal cells in SCSs (Fig.3ab). In the following section, we dive into more details of the Composer step by step.

### 3.1 PART-LEVEL COLUMN

Pyramidal cells in the visible SCS of part column receive inputs from three sources (Fig.3b): the input image $x \in \{0, 1\}^d$, the inner-level feedback $\gamma_1 \in \mathbb{R}^d$ and the inter-level feedback $\Gamma \in \mathbb{R}^d$:

$$\rho_1(t) = x \cdot \gamma_1 \cdot \Gamma \tag{1}$$

where '$\cdot$' is pixel-wise and $\rho_1$ is the firing rate, which determines the firing activity $s_1 \in \{0, 1\}^d$:

$$P(s_1 = 1) = \rho_1(t) \cdot g_1(t - \hat{t}), \quad t \in [0, T]. \tag{2}$$

where $g_1(t - \hat{t})$ is the relative refractory function of neurons in the part level (Fig.3e) and $\hat{t}$ is the timing of the latest spike firing event of each neuron. As shown in Fig.3e, after firing a spike, the neuron goes into an absolute refractory period of timescale $\tau_{r1}$ and then a relative refractory period of timescale $\tau_{\delta 1} - \tau_{r1}$, where firing probability is inhibited by a factor $g < 1$. The inner-level feedback $\gamma_1$ in eq.1 is the denoised output of the $\mathrm{DAE}_1 = G_1 \circ F_1$, yet with delay $\tau_d$:

$$\gamma_1 = \mathrm{DAE}_1((I_1 * s_1)(t - \tau_d)). \tag{3}$$

where $*$ is the convolution operator and $I_1$ is the integrative function for $s_1(t)$, of timescale $\tau_1$ (Fig.3f). In a word, the spiking activity $s_1(t)$ in the visible SCS is integrated within a short time window $\tau_1$ before fed into the $DAE_1$, and the feedback from DAE to SCS is delayed by $\tau_d$.

### 3.2 WHOLE-LEVEL COLUMN

Since the whole-level column is the top level in the current two-level Composer, it does not receive top-down modulation from even higher levels. Besides, the image has a partial influence on SCS in the whole-level column through skip connections (Fig.3c), which is also common in the cortical circuit:

$$\rho_2 = (\lambda \cdot x + (1 - \lambda) \cdot D) \cdot \gamma_2 \quad (4)$$

where $\lambda < 1$ is the factor of partial influence from skip connection. $D$ is the integrated driving input from the part-level column. $\rho_2$ determines the spike firing probability by:

$$P(s_2 = 1) = \rho_2(t) \cdot g_2(t - \hat{t}) \quad (5)$$

Lastly, the delayed feedback from $DAE_2$ is also similar to eq.3:

$$\gamma_2 = \text{DAE}_2((I_2 * s_2)(t - \tau_d)) \quad (6)$$

### 3.3 LINKING THE LEVELS

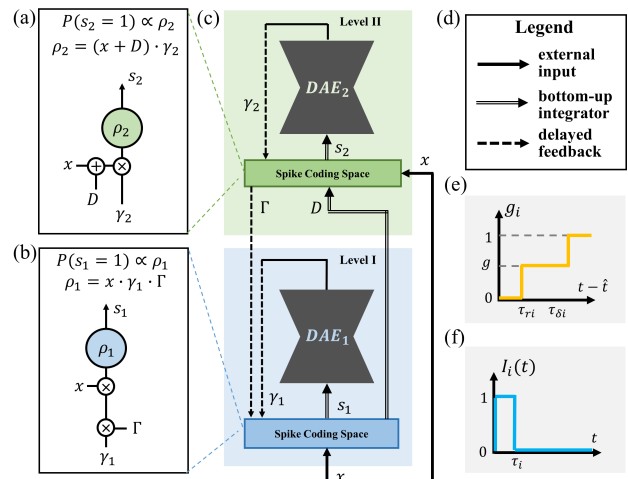

Figure 3: (a)(b) Pyramidal neuron models in the visible spike coding space of part level and whole level. $\otimes$ stands for multiplication and $\oplus$ stands for addition on the dendrites. (c) Detailed information flow. Note that delayed coupling exists both within each column and between different columns. Levels are indicated by color. (d) The legend for (c). (e) Relative refractory function $g$. (f) Integration function $I_i(t)$ of timescale $\tau_i$.

Up to now, we have introduced operations within each column of the Composer except for two variables: $\Gamma$ and $D$, which are interactions between levels:

$$\Gamma(t) = (I_\Gamma * s_2)(t - \tau_{d'}) \quad and \quad D(t) = (I_D * s_1)(t). \quad (7)$$

where $\tau_{d'}$ is the delay timescale from whole-level to part-level. $\tau_\Gamma, \tau_D$ in the $I_\Gamma, I_D$ is the timescale of integration function. It is notable that: (1) While only two levels are considered in this paper for simplicity, the Composer can be naturally extended to account for more levels (eg. Up to five levels are enough for human vision (Hinton, 2021)) (2) While the inter-level projection could account for a wide range of computational goals like coordinate transformation (Hinton, 2021), in this paper, we only focus on a minimal realization as pixel-wise gating (eq.1) and driving (eq.4) between SCSs since we aim to demonstrate how to group information through temporal coherence to represent hierarchical structures in neural networks. Further computational goals like coordinate transformation can be realized in the future by parameterizing the inter-level pathway also as neural networks (Hinton, 2021).

## 4 EVALUATION

As far as we know, most related works on the part-whole hierarchy are evaluated on images containing only a single object without a clear part-whole relationship (e.g. MNIST) (Hinton et al., 2018). Therefore, it is difficult, if not impossible, to distinguish the representation from general feature extraction (Garau et al., 2022) or object-centric attention (Sun et al., 2021), which are much easier problems. The lack of explicit part-whole datasets and quantitative metrics to measure the representation hinders the development of models capable of visual parsing. This challenge motivates us to invent datasets and metrics to explicitly evaluate the Composer.

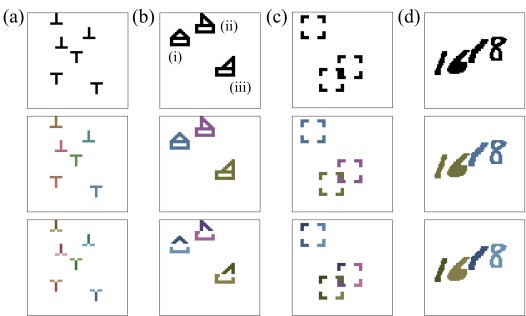

Figure 4: Examples in datasets (a) Ts (b) SHOPs (c) Squares (d) Double-digit MNIST. Top, input. Middle / Bottom, ground truth of wholes/parts. Similar color is used for parts of the same whole.

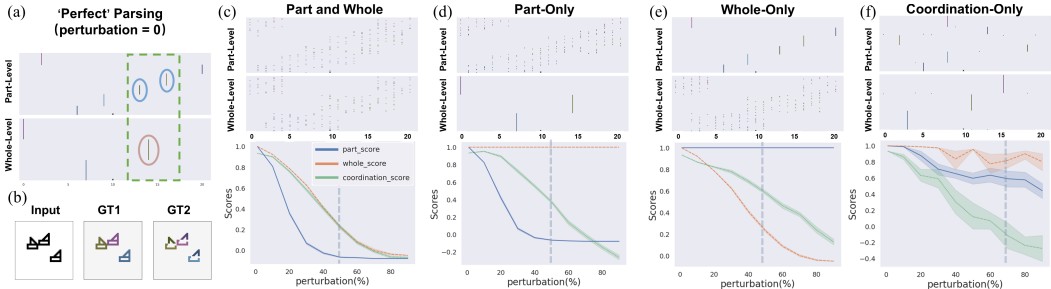

Figure 5: Different Scores measure different aspects of the part-whole hierarchy. (a) Ideal parsing (spike raster plot of two levels, similar to Fig.1d) given the input in (b) (GT stands for ground truth in (b)). Only a single period of the oscillatory pattern is drawn in (a) for clarity. From (c) to (f): We perturb the ideal representation in (a) on different aspects (title on the top) and of different levels (x-axis of bottom figures) to further show what the score measures and their sensitivity. Top, perturbed spiking pattern; Bottom, scores as functions of the perturbation level (**Orange: whole score; Blue: part score; Green: coordination score.**). Dashed vertical line indicates the perturbation level where the spiking pattern (top) is drawn.

## 4.1 DATASET

We invent four synthetic part-whole datasets of different complexities (Fig.4), each containing 60000 samples. Each image consists of multiple whole objects, each of which is further composed of well-defined parts. Whole objects are randomly located in the image. The DAE in part / whole level columns are trained to denoise single part / whole objects (Appendix A.7).

**Ts** dataset (Fig.4a) consists of three letter $\top$ and three reversed letter $\bot$ as whole-level objects. Each $\top$ or $\bot$ is composed of a horizontal bar segment and a vertical bar segment as parts. $T$s dataset has relatively large whole number, but each whole has small part number. Similar stimuli are used as target templates in perceptual tasks like visual search in neuroscience literature (Wolfe, 2021).

**Squares** dataset (Fig.4c) consists of three randomly-located squares as wholes, each of which consists of four corners. The objects in dataset have relatively more parts. Besides, it could demonstrate the role of spatial connected-ness / closure in forming the parsing tree. Similar stimuli are used to study illusory contour (Lee & Nguyen, 2001) in Gestalt perceptual tasks in psychology literature.

**SHOPs**, short for (**S**hoes (Fig.4b-ii), **H**ouse (Fig.4b-i), **OP**era (Fig.4b-iii)) consists of three types of whole objects that are further composed of more elementary rectangular and triangles. Each image contains three randomly selected and located objects. This dataset accounts for the complexity that parts could heavily overlap with each other to construct a whole object. Overlapped regions are not assigned to either object at part level in the ground truth (Fig.4b, bottom).

**Double-Digit MNIST** (Fig.4d) mimics the more realistic scenes when dealing with double-digit numbers. Each image contains two randomly selected and located double digits, and each double-digit is composed of two randomly selected, closely located MNIST digits. This dataset contains objects of higher complexity and diversity.

## 4.2 SCORES

The neural representation of a parse tree can be decomposed into 3 characteristics: (1) The grouping of part-level objects (child node, blue circles in Fig.5a); (2) The grouping of whole level objects (parent node, orange circle in Fig.5a) ; (3) The coordination among parts and wholes (Edges, green box in Fig.5a). Since all three aspects are coherence measures of clusters in nature, we exploit Silhouette Score (Rousseeuw, 1987) to develop the metrics: (1) Part Score (2) Whole Score and (3) Coordination Score to measure the three aspects based on ground truth segmentation. See Appendix.A.4 for more details. To understand how scores work, we perform perturbation studies. As shown in Fig.5c, all scores decrease smoothly from 1 to 0, when the ideally structured pattern gradually gets globally perturbed, finally into total random firing (e.g. Fig.1c). If only the part level is perturbed, the Whole

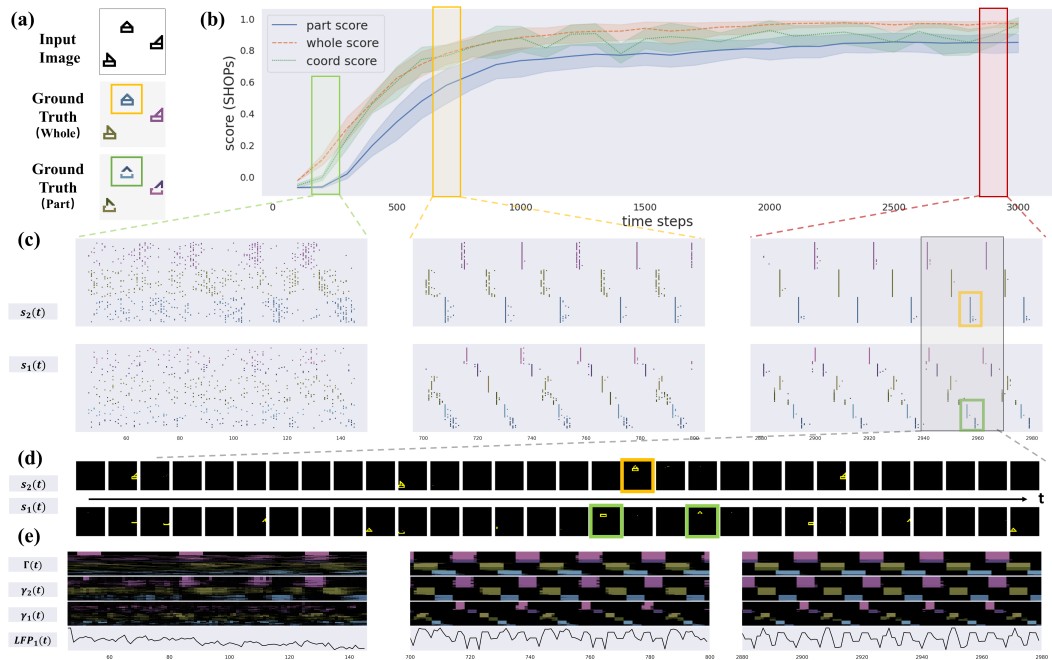

Figure 6: Emergence of the part-whole hierarchy with coordinated neuronal synchrony. Exemplified by one SHOPs sample. (a) Input image and ground truth; (b) Evolution of the Scores. (c) The spike raster plot of three selected phases in (b): phase I (initial, green box), phase II (middle, yellow box), phase III (final, red box). $s_2(t), s_1(t)$ stand for spiking representations in SCSs of whole/part levels. (d) zoomed in spiking pattern during the period marked by black box in (c), to visualize what each synchronized group represents (e.g. yellow/green boxes in (c),(d)and(a)). (e) Evolution of the top-down attention maps and the local field potential (LFP) during the three phases in (b).

Score remains constant while both the Part Score and Coordination Score are decreased from 1 to 0 (Fig.5d). Results are inversed when only the whole level is perturbed (Fig.5e). Lastly, if we only perturb the relative order of parts and wholes, each of which is perfectly grouped as in (a). It is shown that only the Coordination Score decreases saliently while Part / Whole Scores remain mostly unchanged. Since the perturbed order of well-grouped spikes results in systematic wrong assignment of clusters (Fig.5f top), the score decreases to even lower than 0 (Fig.5f bottom). Taken together, Part Score and Whole Score evaluate the build-up of tree nodes, which is the 'pre-requisite' to represent parse trees and the Coordination Score further evaluates the structure of the tree. Following the Silhouette Score, the best score (coherence) is 1 and the worst score (incoherence) is -1. A score near 0 indicates randomness like Fig.1c.

## 5 EXPERIMENTS

### 5.1 QUALITATIVE RESULTS AND VISUALIZATION

**Emergence of the parse tree in SCS**. We visualize the simulation on a randomly selected sample in the SHOPs dataset in Fig.6. As indicated by the convergence of Part Score, Whole Score and Coordination Score (Fig.6b), the Composer gradually achieves a state of neuronal coherence that represents the parts and wholes as synchronized neuronal groups, which is further visualized in Fig.6c right and Fig.6d. More specifically, three two-level binary-tree (corresponding to three SHOPs objects in Fig.6a) periodically emerges in the final phase III (Fig.6c right), one of which is marked out by one yellow (for whole object) and two green (for part objects) boxes. The spikes of parts/wholes in Fig.6c are reordered (on the y-axis) and colored corresponding to the ground truth of parts/wholes (Fig.6a) for more vivid visualization, so that the same neuronal groups are arranged closely and the color of spikes is consistent with the ground truth. For more visualization results, see Appendix A.10.7.

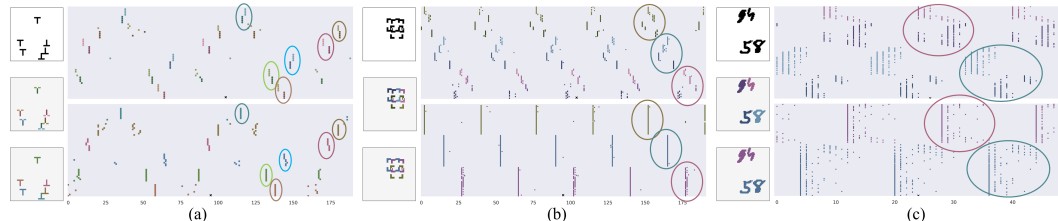

Figure 7: Similar to Fig.6c right, but for other datasets: (a) Ts (b) Squares (c) Double-Digit MNIST. Left: input image, part ground truth and whole ground truth. Right: spike raster plot in final phase III, top for part level and bottom for whole level. Neuronal groups are circled for clarification.

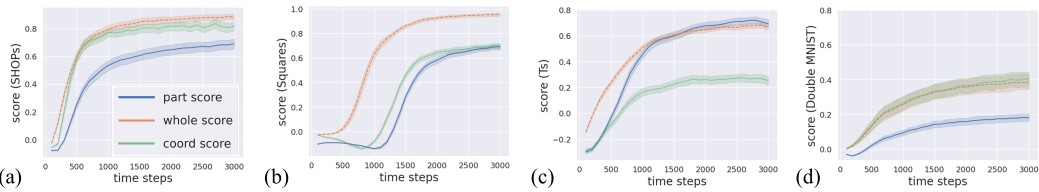

Figure 8: Convergence of Scores. (a) SHOPs (b) Squares (c) Ts (d) Double-Digit MNIST.

**Emergence of the DAE attention map** is observed in Fig.6e: Starting from randomness (left), the cross-level feedback $\Gamma$ or inner-level feedback from the DAEs $\gamma_1/\gamma_2$ gradually converge to structured patterns, similar to the spiking patterns (Fig.6c, right), yet of longer timescales. Therefore, the top-down attention from DAEs and bottom-up integrations of spikes work together as a whole system in Composer. Besides, rhythmic population activity ($LFP_1$) emerges (Fig.6e) at the part level.

**Visualization results on other datasets** are shown in Fig.7. Interestingly, the emergent synchrony structure differs across the datasets. While in the Ts dataset (consists of 6 Ts), 6 binary trees emerge periodically, 3 quadtrees emerge in the Squares dataset (consists of 3 Squares). In Double-Digit MNIST (consists of 2 Double-MNISTs), 2 binary trees emerge. Taken together, the Composer successfully and flexibly represents the part-whole hierarchy of scenes of different complexities.

## 5.2 QUANTITATIVE ANALYSIS

**Convergence** of the scores during iterations are evaluated on 100 randomly selected samples in each dataset and are shown in Fig.8. Interestingly, while scores consistently converge on all datasets with low error bars, the convergent process slightly differs across cases. For Squares (Fig.8b), whole objects group much faster than part objects, similar to human vision (Lee & Nguyen, 2001). For Ts (Fig.8c), the large object number imposes combinatorial burdens on the coordination, so that the Coordination Score lags behind. For Double-Digit MNIST (Fig.8d), Composer has more difficulties in distinguishing part-level MNISTs, partially due to the diversity of the dataset. See Appendix.A.10.5

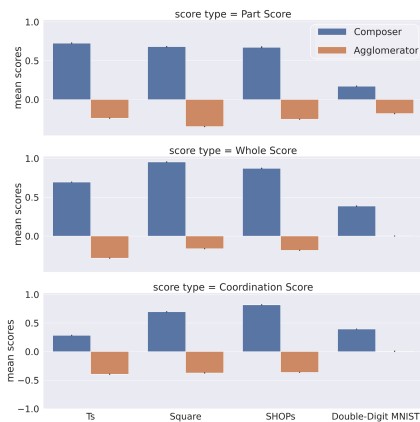

Figure 9: Benchmarking

**Benchmarking**. We compare the Composer with a recently implemented SOTA, the Agglomerator (Garau et al., 2022), which also attempts to exploit the idea of neuronal coherence (similarity among vectors) to group neuronal representation (at different levels) into tree nodes (islands of vectors). 1000 random samples and 5 random seeds are used to evaluate the Composer and the Agglomerator on the four datasets. The 3 coherence-based metric is naturally generalized to evaluate the Agglomerator. As shown in Fig.9, the Composer outperforms the Ag-

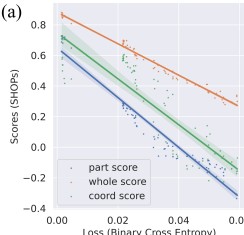
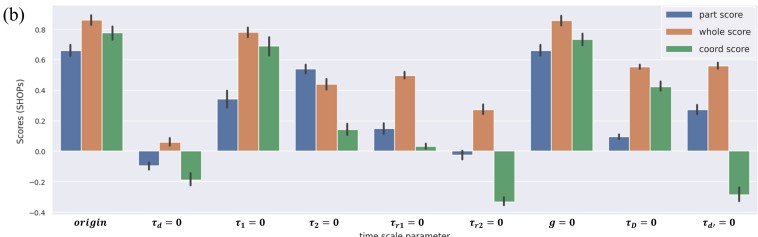

Figure 10: Ablation study on SHOPs. (a) Loss vs Scores (b) Ablation of time scale parameters.

glomerator on all datasets. Actually, the Agglomerator even failed to form the node representation as pre-requisites. See Appendix.A.5 for more details on benchmarking.

**Loss vs Scores**. Since denoised feedback from DAEs is an essential mechanism in the Composer, it is instructive to examine the relationship between the denoising performance of DAE and the parsing scores. For this purpose, we trained 100 DAEs with the same architecture on the SHOPs dataset with random learning rates, and then performed parsing using each of them. Fig.10a shows the relationship between the denoising loss and parsing scores. It is observed that lower loss positively correlates with higher scores on all metrics, indicating that there is a direct interplay between denoising and parsing.

**Ablation study of timescale parameters** are shown in Fig.10b, where parameters are set to zeros in isolation. For example, $g = 0$ stands for the removal of the relative refractory period. Compared with the original model, all ablated models have lower scores. Specifically, the removal of delay $\tau_d$, whole-level refractory period $\tau_{r2}$, and cross-level feedback delay $\tau_{d'}$ has the destructive effect, indicated by reversed Coordination Score. Besides, changes at the part level affect Part Score more than the Whole Score (e.g. $\tau_1, \tau_{r1}$). The removal of relative refractory period $g$ slightly degrades the coordination and the removal of cross-level integration $\tau_D$ globally degrades all scores. (Appendix.A.10.4)

## 6 RELATED WORK

**Object-centric representation** is a line of research that explores how to bind distributed information into single-level objects as reusable entities in neural networks (Greff et al., 2015; 2016; 2017; 2019; 2020; Locatello et al., 2020), some of which also exploit the idea of neuronal synchrony (Zheng et al., 2022; Löwe et al., 2022). While single-level object representations are essential prerequisites to form building blocks (nodes), they can't account for hierarchical structures like part-whole hierarchy.

**Graph neural network (GNN)** can explicitly represent part-whole relationships as a specific type of graph of patches. However, their architecture either is not fixed but changed with the grouped object number (Bear et al., 2020) or needs an object detector to transform the image into node representations (Xu et al., 2017). In contrast, we study how to implicitly represent part-whole hierarchy with neuronal activities in neural networks of fixed architecture that directly process the image, which is more consistent with human vision. Besides, the over-smoothing phenomenon limits the depth of part-whole levels being represented in GNNs (Han et al., 2022).

**Hierarchical latent variable models** can explicitly capture the tree structure within its latent space (Deng et al., 2021). However, current models are feedforward networks without iterative message passing as the Composer, which limits their potential to ultimately conquer the problem.

**Other visual parsers** include capsule-like (Hinton et al., 2018; Garau et al., 2022), transformer-based (Sun et al., 2021), and recursive neural programmer (Fisher & Rao, 2022), etc. The common weaknesses of these works are the evaluations: single-object datasets without clear part-whole relationships (e.g. MNIST) are used and the evaluation lacks metrics to measure the parsing. Therefore, it is unlikely to distinguish the proposed part-whole representation from feature extraction or single-level object-centric representation. In contrast, the Composer's representation is interpreted explicitly.

## 7 CONCLUSION

We present Composer, together with the framework of representation, physical intuition, biologically inspired implementation and explicit evaluation. Results show that Composer uses emergent neuronal synchrony to parse a range of scenes of distinct composite structures, complexities and diversities.

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

## A  APPENDIX

The appendix is organized as follows: We first discuss the **limitation**, broader impact, and potential future work of this paper in Section.A.1 and Section.A.2. Followed by showing the **code availability**, Section.A.3, where the detailed realization of the model and main results can be found. In Section.A.4 (**Metric**), we provide a step-by-step introduction to the evaluation method in this work. Then, we introduce the **benchmark** details in Section.A.5. In Section.A.6, details of the model, especially the organization of time-scale parameters are stressed. Following the model detail, we list the details about the **training** process of DAEs in Section.A.7 and the **hyper-parameters** details in Section.A.8.

We also enumerate the biological motivation and correlates of the Composer in Section.A.9 (**Bio-plausibility**). Lastly, we provide more detailed discussions and additional results (**Visualization**) related to the experiments in the main text, in Section.A.10.

Also, we provide a zip file containing videos visualizing the dataflow of the Composer in SI, about (60MB)

### A.1 LIMITATION AND FUTURE WORK

In this section, we highlight several limitations that could be addressed in future works.

**Visible layer**. In this work, the part-whole hierarchy is represented and evaluated based on the visible space (SCS), which has a one-to-one mapping to the image's pixel space (topographical mapping). Such a setting makes the representation much more interpretable so that the part-whole hierarchy could be explicitly evaluated and visualized as Fig.6, Fig.7. While topographical mapping (a one-to-one spatial relation to the physical world) is a common feature of the cortical representation, the representation could be more abstract. For example, instead of a binary neuron associated with the 'existence' of an object occupying the location, a population of binary neurons can be assigned to each location, so that different aspects of features (associated with the object at each location) could be accounted for by the population vector. Assigning a population of neurons with locations is similar to the capsule idea in Capsule Network (Hinton et al., 2018) and the mini-column organization in GLOM (Hinton, 2021). This suggests a future direction to combine the neuronal synchrony with identical islands of neurons in the GLOM.

Besides, the object could be represented in the latent layer of the DAE instead of the visible SCS in each column, similar to Locatello et al. (2020). Representing objects in the latent layer could enable transforms between levels as in GLOM, by replacing pixel-wise gating in this paper into a neural network that potentially parameterizes a coordinate transformation.

Besides, the input to the SCS of the lowest level is not restricted to be the pixel-level image but could be the output feature map of an encoder, which is called tokenization in Agglomerator (Garau et al., 2022). This kind of generalization has also been discussed in Hinton (2021) and also applies to our model.

In sum, the limitation of the part-whole hierarchy as a pixel-level relationship in visible space could potentially be generalized in three directions: (1) To allocate each location a column of spiking neurons to form a representation space at each location. (2) replacing the simplified cross-level interaction between visible layers as a proper neural network between latent layers and visible layers. (3) the input to the model could be generalized to the tokenized embeddings from the upstream encoder. Notably, all these generalizations are compatible with the model and could be explored as future works.

**Coordination transformation**. As originally motivated in Hinton (1979) and restated in Hinton (2021), part-whole hierarchy contains two challenges: (1) the dynamic emergence of the part-whole tree structure and (2) the implementation of a part-whole coordinate transformation. The insight behind this paper is that the first challenge is the core challenge of the problem while the latter one could be solved by implementing the transformation as a neural network. In other words, the flexible forming of a symbolic tree structure (capable of capturing the basic nested part-whole relationship) within a pure neural network is the hard problem that challenges the neural network models. The second problem, implementing a coordinate transformation, is more compatible with the neural network: such transformation could be realized as a (feedforward) neural network.

In this work, we focus on how to represent the part-whole hierarchy within a pure neural network model through emergent nested neuronal synchrony. The coordinate is assumed to be already aligned between the whole and part so that the coordination transformation is reduced to the inclusion mapping (similar to identity mapping). However, since the parsing tree is realized within a pure neural network, the mechanism is compatible with more general coordination transformation: it could be realized by replacing the pixel-wise conditioning with a neural network, which parameterizes the coordinate transformation.

**More layers**. In this paper, we show the part-whole hierarchy of two levels: whole and part. However, this minimal structure could be naturally extended to account for more levels, since the form of

interaction between levels is mostly irrelevant to how many levels are there or which level it is in. All levels could share a unified form of cross-level interaction and within-level interaction. Therefore, by stacking the columns along the hierarchy, more levels are accounted for. As discussed in Hinton (2021), up to five levels are sufficient to realize human-like vision.

**Synthetic image**. In this paper, we use synthetic images to demonstrate how to represent the part-whole hierarchy. The benefit of using the synthetic image is that: (1) a common sense reasonable part-whole relationship is known beforehand as ground truth, therefore it is more convenient to explicitly evaluate the representation and test the capability. (2) The ground truth assignment of objects (part/whole) is known, which could be utilized to evaluate the neuronal coherence. The weak side of a ground truth is that such explicit assignment of part-whole ignores the ambiguity of parsing the scene: the parsing could depend on many factors like prior knowledge, attention, goal, internal state, and so on. Besides, parsing a real-world image without explicit part-whole hierarchy might be challenging for other reasons (overlap, background, etc.). However, recent models (Hinton et al., 2018; Sun et al., 2021; Garau et al., 2022) that claim to solve the part-whole problem actually resemble performing hierarchical feature extractions. Such confusion is partly due to the ambiguity of the part-whole relation, object definition and the complexity of features in the real-world images, which makes the symbolic structure harder to distinguish. Therefore, taking the present status of the problem[1] and the challenges the problem implicates[2] into account, it is desirable to focus on explicit evaluation based on synthetic data first (so that it is easier to interpret whether the mechanism works) and then gradually generalize the outcome to increasingly complex datasets in the future.

**Learning scheme**. In this work, we treat the 'sense' of what the object should look like as prior knowledge embedding in the parameters of DAE's weight, which in turn determines the dynamical property of the Composer. Indeed, such prior is needed for humans to parse a visual scene as well. For example, given a visual scene of a house (Fig.1a in the main text), a human observer should have already had the concept of the door, the window, and the roof in their mind, so that a house is parsed in the way Fig1.b (main text) shows. Therefore, in this work, we consider how a parsing structure could emerge as coordinated neuronal coherence in a pure neural network given the prior knowledge of objects. On the one hand, some of the priors are indeed hard-wired in the brain through a long period of evolution (related to Gelstalt psychology (Wagemans et al., 2012), like proximity, similarity, enclosure, continuation, closure, symmetry, common fate, etc.); on the other hand, some of the others may be gradually learned during evolution. Therefore, the learning scheme could be improved to capture how the part-whole hierarchy could emerge during the unsupervised perception of the multi-object world. A preliminary insight is that: the model architecture in this paper (column organization, time scale relationship) could be regarded as inductive bias to form a hierarchically factorized representation during the unsupervised training of the whole model as a recurrent spiking neural network to reconstruct what it sees on average during a temporal period. More details are discussed in Section.A.7.4

## A.2 BROADER IMPACT

On the positive side, the model parses objects with neuronal coherence in the visible space composed of spiking neurons without explicit supervision. The mechanism by principle is not limited to a certain modality or certain object type. Thus, it may help develop human-like perception systems. Besides, with biological relevant features (eg. delayed coupling) and phenomena (eg. synchrony), the model may also act as a data-driven biological model to understand the perception process in the brain.

On the negative side, since the model is not supervised, it is harder to control what it learns. The current model is only trained on simple synthetic datasets and learns to group at the superficial pixel level, therefore the representation is highly explainable. However, grouping in latent space on real-world datasets requires to develop evaluation and visualization methods to make the representation in latent space more understandable. We believe this may serve as a step toward more transparent and interpretable predictions.

---

[1]Representing the part-whole hierarchy in a pure neural network is still an unsolved problem (**?**)

[2]In essence, the part-whole problem requires a general solution to the sub-neuro-symbolic architecture and to realize hierarchical split of computational problem in a divide-and-conquer way. This paper explores the temporal aspect of the solution.

## A.3  CODE AVAILABILITY

The source code for results in the paper and a video demonstrating the whole simulation process can be found at:

https://drive.google.com/drive/folders/1GTHhpdafze6rExjD9NtV8beLfruMCMJR?usp=sharing.

Codes will also be updated to Github:

https://github.com/codingbugmonster/part_whole_hierarchy/ after clean-up.

## A.4  METRIC

In this section, we introduce how the metrics for quantitative evaluations are defined based on the Silhouette Score, including the Part Score, Whole Score, and Coordination Score. Since the Silhouette Score is based on the similarity measure among samples, we first introduce how the similarity among spike trains is measured, where the Victor-Purpura metric shows up. Then we introduce the Silhouette Score and how it could be extended to account for varied aspects of the part-whole representation. Finally, we discuss how the metrics can be generalized to evaluate other models with similar attempts to group neural representation by similarity measure.

### A.4.1  HOW TO MEASURE THE DISTANCE BETWEEN SPIKE TRAINS: VICTOR-PURPURA METRIC

The Victor-Purpura metric (VP-metric) is a classical non-Euclidean metric to measure the distance between arbitrary spike trains for evaluating the temporal coding in the visual cortex (Victor & Purpura, 1996). Three types of operations are identified (Fig.11): 1. add a spike (cost=1); 2. delete a spike (cost=1); 3. shift a spike for length $\Delta t$ (cost=$\Delta t/\tau$). By sequentially applying the three operations ($T(u)$ in Fig.11), a spike train can be transformed to the other. $\tau$ is a parameter to control the temporal precision of the spiking code (or the temporal sensitivity of the metric). The Victor-Purpura distance is defined as the minimal cost to transform a spike train to the other (Fig.11):

$$D_{VP}(s_i, s_j; \tau^{-1}) = min_T(\sum_{u=1}^{|T|} cost(T(u))) \tag{8}$$

$$T(u) \in \{delete, add, shift\} \tag{9}$$

$$cost(delete) = cost(add) = 1 \tag{10}$$

$$cost(shift, \Delta t; \tau) = \Delta t/\tau \tag{11}$$

where $T(u), u = 1...|T|$ is a sequence of basic transformations to transform $s_i$ to $s_j$ (or vice versa). The costs of the three basic transformations are different. The most special one is the shift operation: there is a time scale parameter to control the punishment of shifting the spike to its neighbourhood.

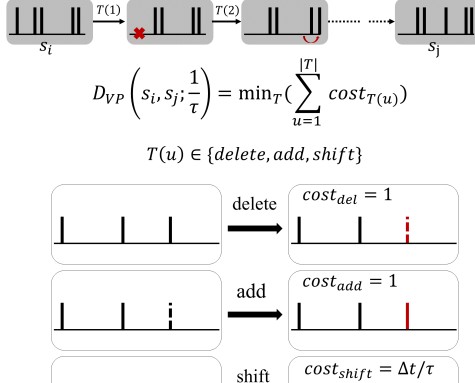

Notably, it is proved that the definition satisfies the three principles of a metric: positivity, symmetry, and triangle inequality (Victor & Purpura, 1996). Thus, it induces a metric space of arbitrary spike trains, even if not embedded in a vector space of specified dimension. Since spike trains are non-Euclidean in nature, the VP-metric provides a more direct measure of these entities. The minimal cost is computed through a dynamic programming method.

A desirable feature of VP-metric is that the parameter $\tau$ explicitly controls the temporal sensitivity of the metric and the expected temporal precision to be considered. If the $\tau$ is chosen to be $\infty$, then shifting a spike will cause no cost ($1/\infty \sim 0$). Thus the distance is exclusively due to spike count[3], therefore spiking rate is measured. If the $\tau$ is chosen to be 0, then it measures the number of spikes that are not in absolute synchrony [4]. So small $\tau$ measures the spike train distance based on the very precise temporal synchrony structure. By varying the $\tau$, it is plausible to find the optimal coding scheme of the visual cortex (Victor & Purpura, 1996). In sum, $\tau$ is treated as a timescale parameter, to control the precision of temporal coding. In this paper, the part level is evaluated with smaller $\tau$ (part-level time scale) while whole level are evaluated with larger $\tau$ (whole-level time scale). The coordination between the levels is evaluated with the time scale of the whole level (the child node should stay within the time window of their parents.)

### A.4.2 General coherence measure of clusters: Silhouette Score

The Silhouette coefficient (Rousseeuw, 1987) is a score to evaluate the quality of clustering by measuring the inner-cluster coherence. Given a clustering assignment, the score is calculated using average intra-cluster distance (a) and average nearest-cluster distance (b). The score is computed as $(b - a)/max(a, b)$. The document can be found at https://scikit-learn.org/stable/modules/generated/sklearn.metrics.silhouette_score.html. The best score is 1 and the worst score is -1. The values near 0 indicate overlapping clusters.

### A.4.3 Victor-Purpura Metric + Silhouette Score

How could the neuronal synchrony be measured? Given ground truth assignments of neurons (to the parts or wholes), synchrony is measured as temporal coherence of (ground-truth assigned) neuronal groups: the inner-group similarity and inter-group separability.

Specifically, if we take (1) each neuron as a **sample**, (2) the spike train of each neuron as **features**, (3) the ground truth assignment as the **clustering assignment**, (4) the VP-metric as the **distance measure**, then, the inner-cluster coherence of the clustering (Silhouette Score) is exactly the coherence measure of neuronal synchrony. In other words, the high Silhouette Score indicates that the spike trains of neurons of the same group are closer to each other in terms of VP-metric, which can be interpreted as neurons of the same group synchronizing better. Therefore, the VP-induced Silhouette Score sufficiently measures the grouping quality. The VP-induced Silhouette Score is also from -1 to 1. The best value is 1 (perfect grouping) and values near 0 indicate overlapping clusters (purely random firing without any temporal structure). Negative values generally indicate that a sample has been assigned to the wrong cluster, as a different cluster is more similar (neurons synchronize to incorrect groups).

During the whole simulation, only a segment of simulation is used for evaluating the Silhouette Score, See Table.1.

### A.4.4 Scores to measure the part-whole hierarchy

Coherence scores are all defined based on the VP-Silhouette Score and Ground Truth assignment.

**Part Score** is defined as the VP-Silhouette score with respect to the part-level spiking pattern and the part-level ground truth assignment:

$$Part - Score = Silhouette(VP(spk_1, spk_1; \tau_p), label_1) \tag{12}$$

where $spk_1 \in \{0, 1\}^{(N, \tau_l)}$ means the total spike trains in level 1 (part-level). $\tau_l$ is the length of each spike train for evaluation and $N$ is the number of neurons at part level. $VP(spk_1, spk_1)$ is the distance matrix whose elements (i,j) are the VP-distance between the i-th spike train and the

---

[3]All transforming cost comes from adding/deleting spikes

[4]only total synchronous spike trains have 0 distance while the slight shift of spikes has cost 1

j-th spike train in the part level; The $label_1$ means the ground truth assignment of neurons in level 1 (part-level). $Silhouette$ is the Silhouette Score. $\tau_p \sim \tau_1$ is close to the (integration) time constant of part-level (Table.1), controlling the temporal sensitivity of VP-metric (eq.8). Therefore, the Part Score measures the coherence level of the part level exclusively, independent of the activity in the whole level. Part Score indicates the quality of the grouping of tree nodes in the part level. For example, $Part - Score = 1$ indicates that neurons are synchronized perfectly into separated groups corresponding to the part objects. On the other hand, $Part - Score = 0$ indicates that the neurons fire randomly and no temporal structure emerges. In rare cases, $Part - Score < 0$ indicates that (on average) neurons with different assignments are synchronized and neurons with the same assignments are not synchronized.

**Whole Score** are similarly defined as:

$$Whole - Score = Silhouette(VP(spk_2, spk_2; \tau_w), label_2) \tag{13}$$

where $spk_2 \in \{0, 1\}^{(N, \tau_l)}$ means the total spike trains of level 2 (whole-level). $VP(spk_2, spk_2)$ is the distance matrix whose elements (i,j) are the VP-distance between the i-th spike train and the j-th spike train in the whole level; The $label_2$ means the ground truth assignment of neurons in level 2 (whole-level). $\tau_w \sim \tau_2$ is close to the time constant of the whole-level: $\tau_w > \tau_p$ (Table.1).

On the one hand, the forming of tree nodes is a necessary condition to form the entire tree, so Part Score and Whole Score are important measures of the representation. On the other hand, since the Part Score and Whole Score measures the grouping in part/whole level independently (Fg.5de in the main text), they do not reveal the correlation between levels. For example, the emergent groups can be arbitrarily permuted or translated (together) without affecting the scores (Fg.5f in the main text). Obviously, such arbitrary operations are serious enough to destroy a well-defined tree structure (Fg.5f in the main text).

Therefore, we provide the additional score to capture the cross-level coordination: the Coordination Score.

**Coordination Score** is defined as the coherence score between part-level spiking patterns and whole-level spiking patterns based on whole level assignments:

$$Coordination - Score = (4/3) \cdot Silhouette(VP(spk_1, spk_2; \tau_w), label_2) \tag{14}$$

where $spk_1, spk_2 \in \{0, 1\}^{(N, \tau_l)}$ means the total spike trains of level 1 (part-level) and level 2 (whole level) respectively. $VP(spk_1, spk_2)$ is the distance matrix whose elements (i,j) are the VP-distance between i-th spike train in the part level and j-th spike train in the whole level. $(4/3)$ is a normalization factor, introduced below.

Intuitively, the coordination means that synchronized neuronal groups at the part-level are coordinated within the lifetime of whole-level neuronal groups (Fig.1 in the main text), which is also called nestedness. Therefore, it could be formalized as the coherence measure between part-level spike trains and whole-level spike trains: $(b - a)/\max(a, b)$. Here, $a$ is the average distance between part-level spike trains and whole-level spike trains that share the same whole-level assignment (Fig.12 top, green dashed box). $b$ is the mean distance between a whole-level spike train and the nearest part-level neuronal group that the spike train is not a part of (the averaged distance between sets, Fig.12 top, blue dashed box). This formulation is similar to the Silhouette, but replaces the $VP(spk_1, spk_1), VP(spk_2, spk_2)$ to the $VP(spk_1, spk_2)$.

However, since part-level and whole-level should not be exactly the same, the derived $(b - a)/\max(a, b)$ do not reach the 1 in best cases. To normalize the score into the range of $(-1, 1)$, we compute a compensatory factor. To simplify the problem, we assume that in the ideal parsing case, both part-level and whole-level spikes are synchronized perfectly and arranged uniformly along the time dimension (Fig.12 bottom). Assume the nearest time interval between part-level and whole-level spikes is $\tau$ (Fig.12 bottom), then for a binary tree (Fig.12 a):

$$a = \tau, b = (3\tau + 5\tau)/2 = 4\tau \tag{15}$$

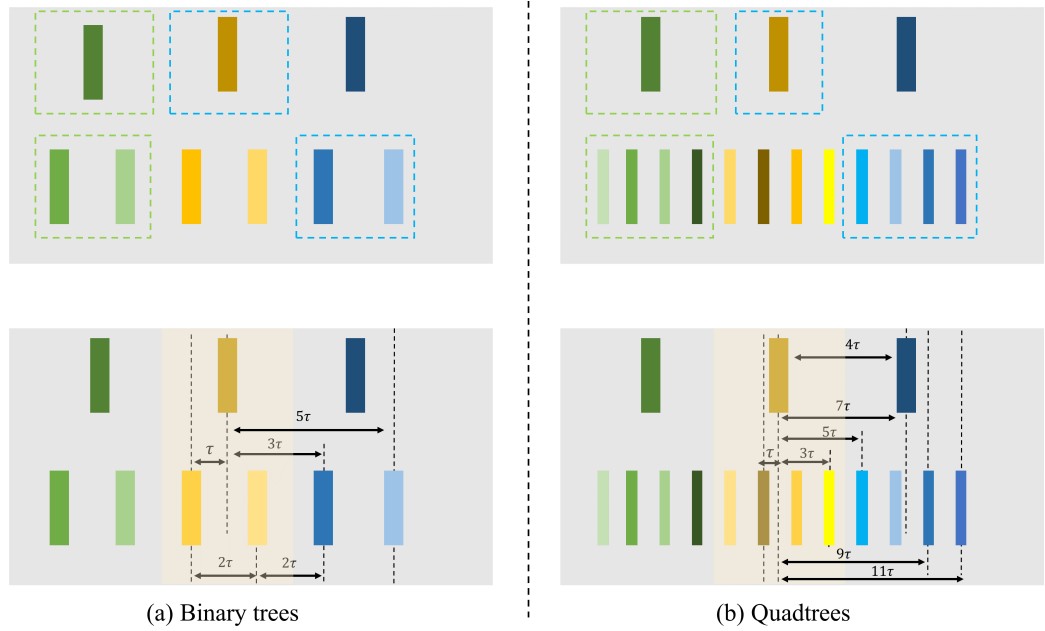

(a) Binary trees        (b) Quadtrees

Figure 12: Illustration of the Coordination Score. Each colored bar stand for a population of synchronized neurons, as in Fig.2de in main text.

$$Coordination - Score = (b - a)/\max(a, b) = 3/4 \qquad (16)$$

and for quadtree:

$$a = (\tau + \tau + 3\tau + 3\tau)/4 = 2\tau \qquad (17)$$

$$b = (5\tau + 7\tau + 9\tau + 11\tau)/4 = 8\tau \qquad (18)$$

$$Coordination - Score = (b - a)/\max(a, b) = 3/4 \qquad (19)$$

Interestingly, in both ideal cases, the Coordination Score is $3/4$, as a result, we normalize the derived Silhouette Score by a factor (3/4), which is exactly the eq.14. Here binary tree accounts for SHOPs, Ts, and double-digit MNIST while quadtree accounts for the Squares. The validity of the normalization is confirmed in Fig,5 in the main text, which achieves 1 in ideal cases.

Lastly, the reason why it is valid to use whole-level assignment to ground both part-level and whole-level neurons is that we have assumed a one-to-one spatial relation between part-level visible SCS and whole-level visible SCS (topographical mapping to the physical world), and the complete object has a 'copy' at each level along the hierarchy (main text). It is also a conventional assumption of the cortex (Hinton, 2021).

### A.4.5 EXPERIMENTALLY VERIFY THE SCORES: THE PERTURBATION STUDY

In order to verify the proposed scores, we conduct a perturbation study in Fig.5 in the main text. Here, we provide more details on how the perturbation is made and more discussions about the experiment.

Given an input image and its ground truth as in Fig.5b, we firstly 'artificially' build up the 'perfect parsing' spike pattern in one oscillation period ($\tau_{total} \sim 20$-time steps). More specifically, all neuronal groups are synchronized perfectly and arranged uniformly along the time axis as in Fig.12.

Part-level neuron groups are coordinated within the lifetime of whole-level neuronal groups (Fig.5a). It is the ideal case, with 0 perturbation level in Fig.5cdef.

Then, for Fig.5bde, we randomly and independently perturb the timing of spikes into nearby time points:

$$t_i \longrightarrow t_i \pm \Delta t, \Delta t \leqslant \tau \tag{20}$$

where $t_i$ is the spike timing of i-th neuron, $\tau$ is the timescale controlling the perturbation level. If $\tau = 0$, perturbation is zero. If $\tau$ equals the length of the whole period $\tau_{total}$, the perturbation will lead to pure random firings like Fig.1d. As a result, we define the perturbation level in Fig.5bde as $\tau/\tau_{total}$, ranging from $0\%$ to $100\%$. A more detailed perturbation process is shown in Fig.13 to Fig.15. In Fig.5c, the perturbation is applied to both part and whole level (Fig.5c top) so that all scores smoothly decrease from 1 to near 0. In Fig.5d, the perturbation is only applied to the part level (Fig.5d top) so that the Whole Score is not affected but both Part Score and Coordination Score smoothly decrease from 1 to near 0. In Fig.5e, the perturbation is only applied to the part level (Fig.5e top) so that the Part Score is not affected but both Whole Score and Coordination Score smoothly decrease from 1 to near 0. In a word, in Fig.5d and Fig.5e, we isolately verify the property of Part / Whole Score, which shows that they are capable of capturing the quality of node-level representation. In Fig.5c, we provide more common cases where both part and whole level degrades, which shows that three scores consistently measure the coherence of neuronal representation.

For Fig.5f, to isolately verify the role of Coordination Score. We build up perfect synchronized neuronal groups as in the perfect parsing case (Fig.5a), but then perturb the timing of each 'neuronal group' at different levels. All spikes within the same neuronal group are perturbed with the same $\Delta t$ and different neuronal groups are perturbed by independent $\Delta t s$ (Fig.5f top). Similarly, we define perturbation level as $\tau/\tau_{total}$, where $\tau$ is the timescale controlling the perturbation. A more detailed perturbation process is shown in Fig.16. As shown in Fig.5f, the Coordination Score decreases smoothly while Part / Whole Scores remain almost constant. The slight decrease in Part / Whole Scores and the slightly higher variance of the score is likely due to the perturbation scheme instead of the property of scores: Perturbing the entire neuronal groups can potentially synchronize different neuronal groups so that Part / Whole Scores get slightly decreased. This effect also increases the variance. Notably, the perturbation can lead to wrong coordination: whole-level neuronal groups are synchronized with part-level neuronal groups of different assignments. Thus, the Coordination Score can decrease into values even lower than 0.

### A.4.6 THE SHIFT: FROM SYNCHRONIZATION TO POLYCHRONIZATION

In the neural system of the brain, the synchronization matters since the coincident arrival of spike trains could have a much larger effect on the target neuron. Therefore, it is the 'synchronization' in the viewpoint of the reader neuron that really matters (Buzsáki, 2010). However, due to the diverse axonal delay (tens of milliseconds) of different neurons, coincidently arrived spikes are usually fired at different timings, yet with fixed temporal shifts. This phenomenon is called polychonization, or polychonized neuronal groups (PNG) (Izhikevich, 2006), which generalizes the concept of synchronization and is a more natural outcome of a real-world neural system, with potentially heterogeneous parameter settings. In other words, polychony and synchrony bear the same spirit of fixed temporal correlation, but polychonization could tolerate a fixed temporal shift among spike timings. While in an external observer's viewpoint, two things are different, in the viewpoint of the readout neuron, both can be the same thing.

In other words, if we shift all timing patterns with a fixed shift parameter, it is equivalent to the original pattern in the sense that the temporal shift can be compensated by the fixed axonal delay when being read out by a downstream module. Motivated by this fact, such slight fixed shifts are compensated before computing the Coordination Score. In other words, in the Composer, whole-level and part-level neuronal groups are allowed to have a slight fixed temporal shift (translation slightly along the time axis). The representation is regarded as unaffected as long as the shift is a constant.

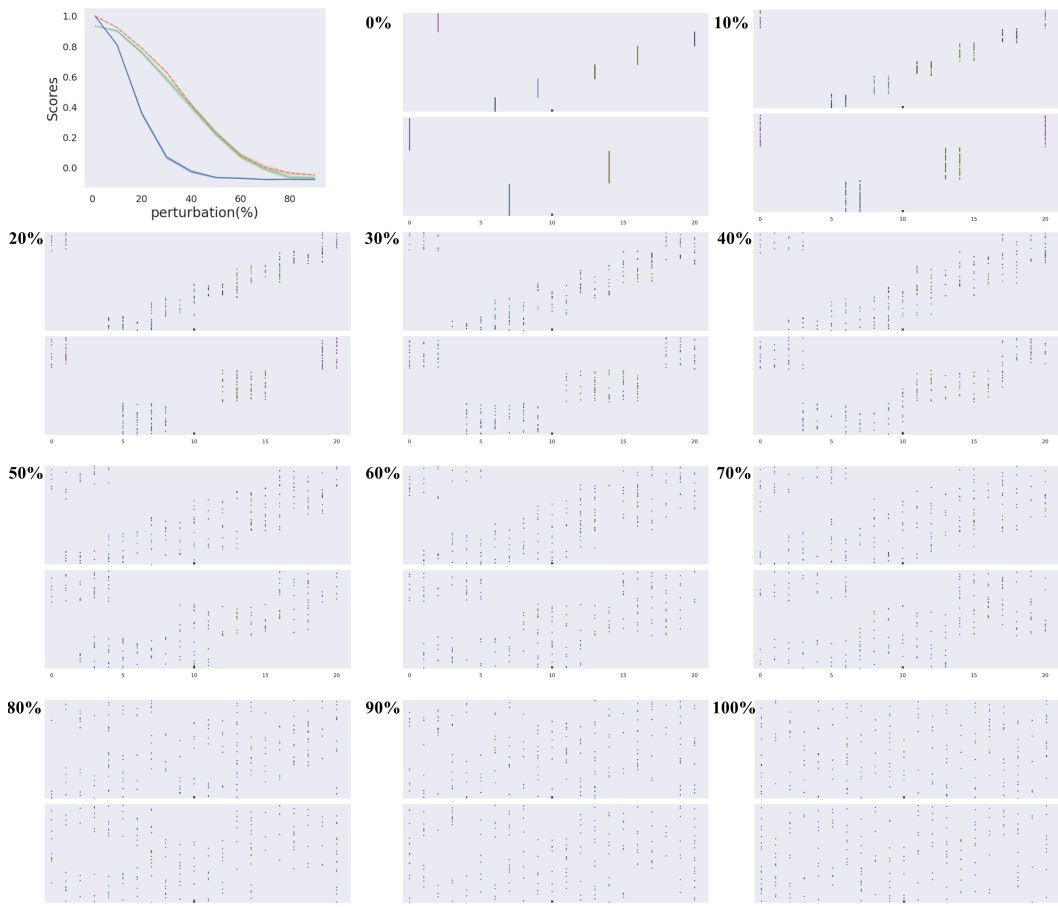

Figure 13: Visualization of the perturbed spiking pattern at different perturbation levels ($0\%\,to\,100\%$), corresponding to the Fig.5c. Both part-level and whole-level gradually degrades into random firings.

Table 1: Parameters of the evaluation. $\tau_p$, $\tau_w$ is the time scale parameter for computing the VP-distance in each case. Shift is the fixed modified time steps for computing Coordination Score and for visualization. The segment length of spike trains for computing scores ($\tau_l$) and for visualization is also shown.

| Dataset | Ts | Squares | SHOPs | Double MNIST |
|---|---|---|---|---|
| $\tau_p$ | 2 | 2 | 3 | 2 |
| $\tau_w$ | 6 | 7 | 6 | 4 |
| shift | 10 | 9 | 6 | 2 |
| segment length (score,$\tau_l$) | 160 | 75 | 42 | 32 |
| segment length (visualization) | 200 | 100 | 100 | 70 |

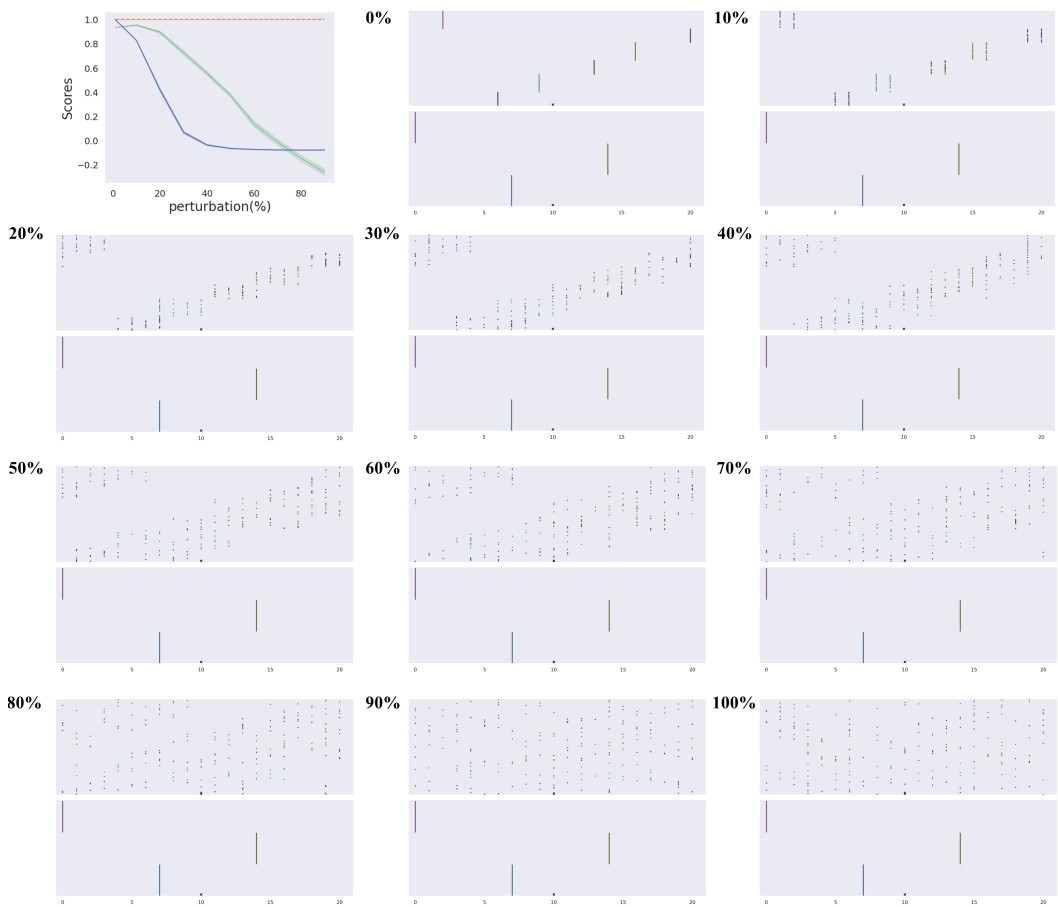

Figure 14: Visualization of the perturbed spiking pattern at different perturbation levels ($0\% to 100\%$), corresponding to the Fig.5d. Part-level is degraded gradually while whole level remains unchanged.

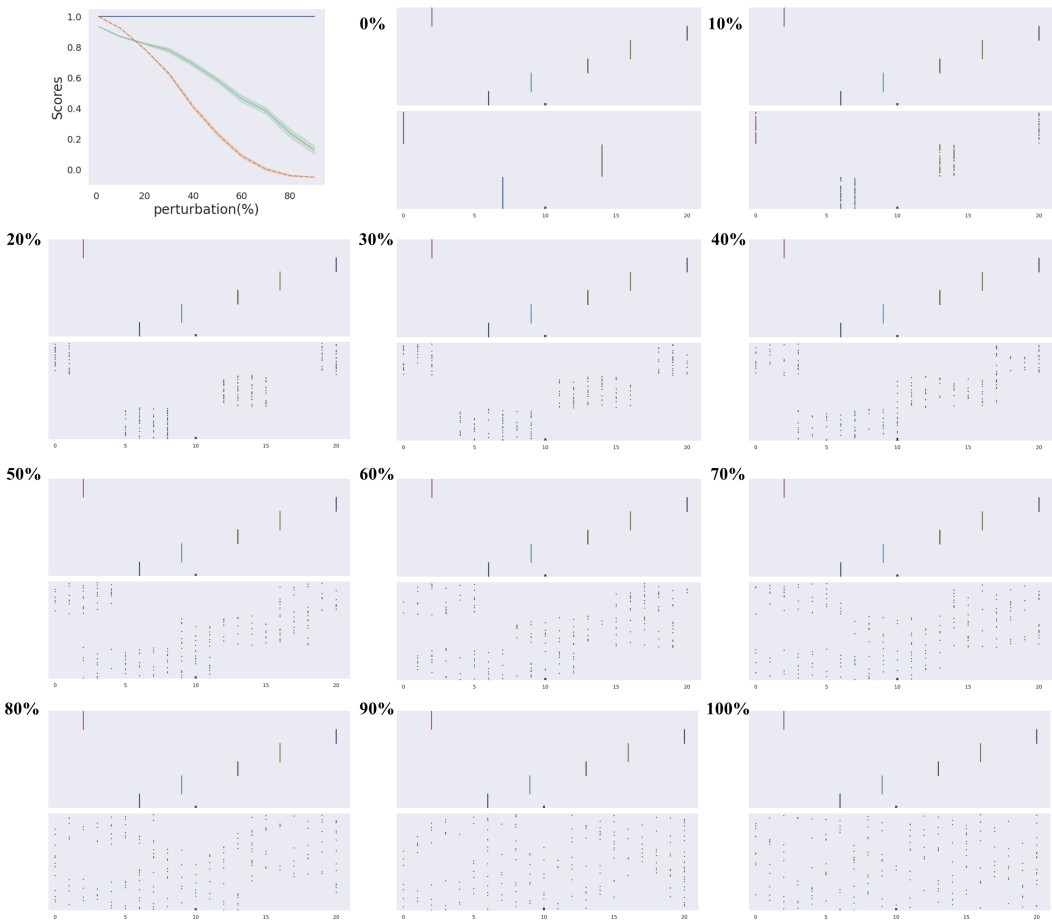

Figure 15: Visualization of the perturbed spiking pattern at different perturbation levels ($0\%\,to\,100\%$), corresponding to the Fig.5e. Whole-level is degraded gradually while part level remains unchanged.

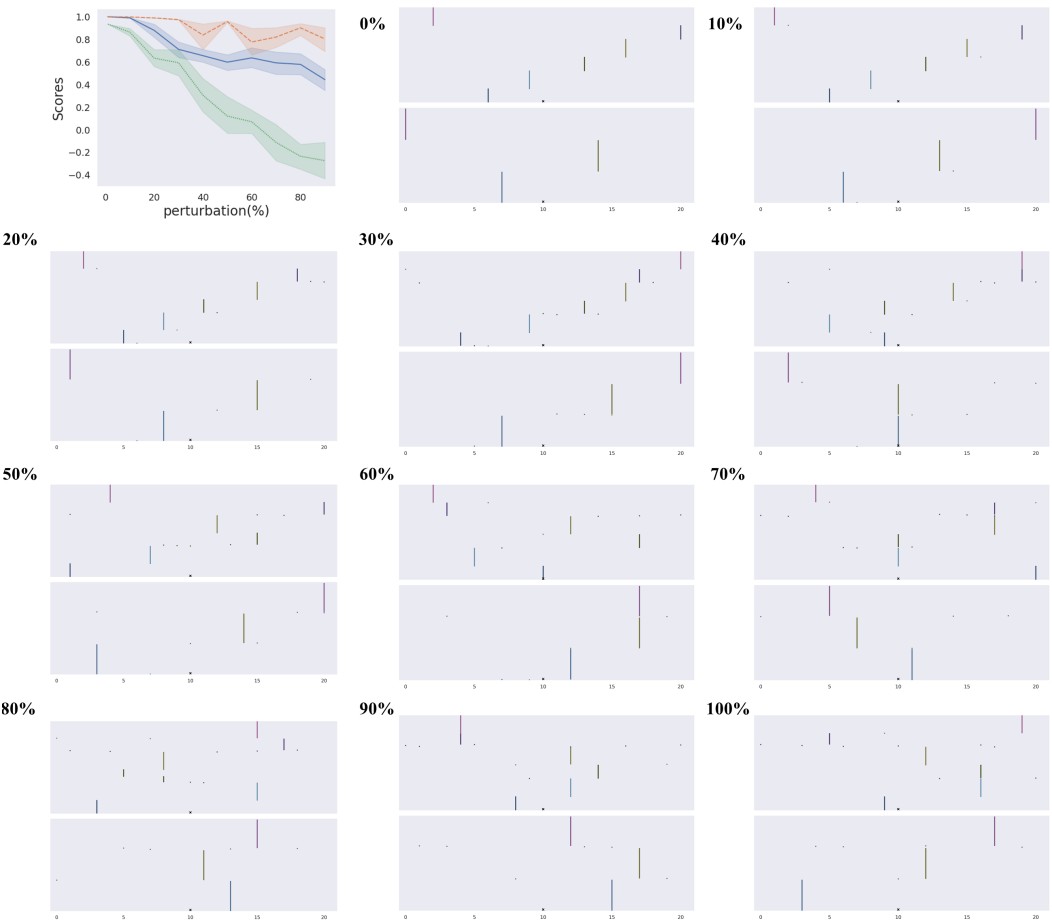

Figure 16: Visualization of the perturbed spiking pattern at different perturbation levels ($0\%$ $to$ $100\%$), corresponding to the Fig.5f. Relative coordination among neuronal groups is gradually changed while the synchronization of each neuronal group is unchanged.

Table 2: Scores of the Composer

| Dataset | Ts | Squares | SHOPs | Double MNIST |
|---|---|---|---|---|
| Part Score | $0.73 \pm 0.005$ | $0.67 \pm 0.007$ | $0.67 \pm 0.008$ | $0.17 \pm 0.003$ |
| Whole Score | $0.69 \pm 0.001$ | $0.87 \pm 0.003$ | $0.87 \pm 0.003$ | $0.39 \pm 0.005$ |
| Coordination Score | $0.28 \pm 0.004$ | $0.81 \pm 0.003$ | $0.82 \pm 0.004$ | $0.39 \pm 0.002$ |

Table 3: Scores of the Agglomerator

| Dataset | Ts | Squares | SHOPs | Double MNIST |
|---|---|---|---|---|
| Part Score | $-0.24 \pm 0.002$ | $-0.35 \pm 0.002$ | $-0.25 \pm 0.001$ | $-0.18 \pm 0.003$ |
| Whole Score | $-0.28 \pm 0.002$ | $-0.16 \pm 0.001$ | $-00.18 \pm 0.002$ | $0.00 \pm 0.001$ |
| Coordination Score | $0.40 \pm 0.003$ | $-0.37 \pm 0.003$ | $-0.36 \pm 0.001$ | $0.01 \pm 0.004$ |

### A.4.7 GENERALIZE TO EVALUATE OTHER MODELS

The metric proposed in this paper is also applicable to neural models that exploit similarity or coherence measures to group neural representations into part-whole hierarchies. GLOM (Hinton, 2021) and GLOM-inspired Agglomerator (Garau et al., 2022) is one interesting example, which is compared as the benchmark. To measure the similarity among vectors, the Victor-Purpura metric is not needed anymore. Therefore, it is more direct to take each vector as a sample and the different dimensions of the vectors as the features in a clustering algorithm. In this way, three Silhouette-based coherence measures of spike trains could be naturally generalized to account for real-valued vectors.

An interesting point is that the islands of identical vectors in GLOM are parallel to the synchronized neuronal groups in the Composer, as long as we take each temporally unfolded spike train as the (binary) vector in each GLOM's column.

## A.5 BENCHMARKING

### A.5.1 THE SCORES OF THE COMPOSER

Here we list the scores of the Composer corresponding to Fig.9 in the main text. The value in Table.2 is the mean averaged scores on 1000 randomly selected samples and 5 random seeds are used. The error bar is very low.

### A.5.2 THE SCORES OF THE AGGLOMERATOR

Here we list the scores of the Agglomerator corresponding to Fig.9 in the main text. The value in Table.3 is the mean averaged scores on 1000 randomly selected samples and 5 random seeds are used. The error bar is very low.

### A.5.3 INTRODUCING THE AGGLOMERATOR

The Agglomerator (Garau et al., 2022) is a GLOM (Hinton, 2021) inspired implementation to deal with the part-whole hierarchy. The basic idea is to use similarity measures among vectors, which are called columns, to dynamically group neuron representation into "identical islands of vectors". The spatial inclusion relationship among islands is interpreted as the part-whole hierarchy. We show the simplified architecture and representation scheme of the Agglomerator in Fig.17, in case the reader is not familiar with the model.

More specifically, columns are organized into different levels. At each level, columns are spatially located on a grid mesh, like the topographical mapping. Different levels have different spatial scales, reflected by different radii of horizontal connection among columns within the levels.

To put it in a nutshell, the Agglomerator also attempts to exploit (spatial) neuronal coherence to dynamically form the tree node at each level and to coordinate the nodes naturally by spatial nestedness. It is the similarity of the representation/architecture that we choose Agglomerator as an

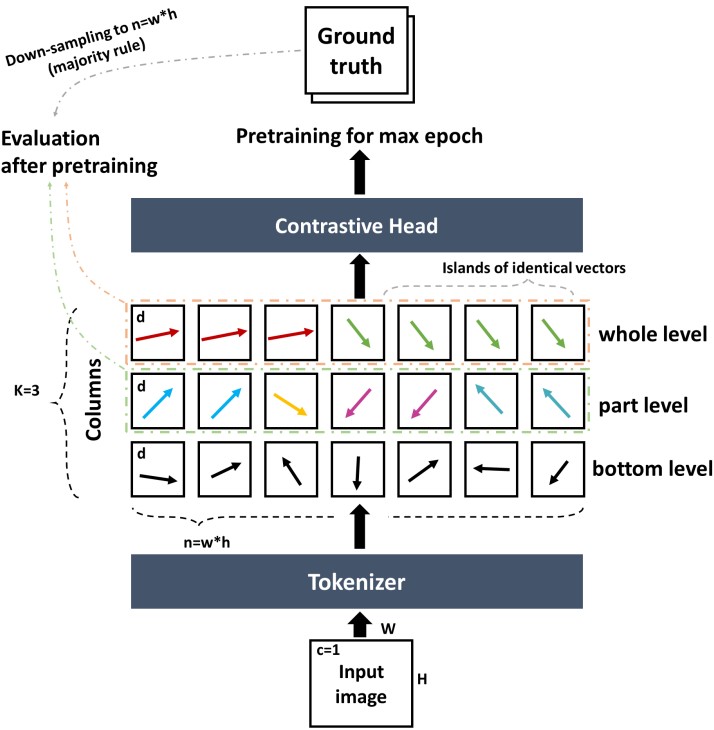

Figure 17: Illustration of the architecture of the Agglomerator, how part-whole hierarchy should be represented as spatially nested islands of identical vectors and how evaluation is achieved.

appropriate benchmarks and recent benchmark. However, as shown in Table.3, they failed to form the node-level representation at all in the four explicit datasets, indicated by the Part Score and Whole Score lower than 0. Since they do not coordinate the cross-level nodes by similarity measure among vectors (but by spatial relationship of islands), the Coordination Score does not reveal more insight except that representation between levels is not as coherent as in the Composer.

### A.5.4    HOW COHERENCE IS MEASURED IN THE AGGLOMERATOR?

Actually, it is direct to generalize the coherence metrics to account for evaluating the identical islands:

$$Part - Score = Silhouette(D_2(l_p, l_p), label_1) \tag{21}$$

$$Part - Score = Silhouette(D_2(l_w, l_w), label_2) \tag{22}$$

$$Coordination - Score = Silhouette(D_2(l_p, l_w), label_2) \tag{23}$$

where $D_2$ is the Euclidean distance metric, $l_p \in \mathbb{R}^{(w*h,d)}$ is the part-level column representaion. $w$ and $h$ is the size of the grid mesh of columns, $d$ is the dimension of each column. Here, the dimension $d$ is parallel to the time dimension in our model. If identical islands are formed, Part / Whole Scores should reach values near 1.

### A.5.5    THE ARCHITECTURE DETAIL OF THE AGGLOMERATOR

We bear the most parameters and architecture settings from the original paper. The detailed parameter setting is shown in Table.4. An illustration of these parameters is shown in Fig.17.

Table 4: Parameter details for benchmarking

| Parameter | Ts | SHOPs | Squares | Double-Digit MNIST |
|---|---|---|---|---|
| K | 3 | 3 | 3 | 3 |
| d | 48 | 48 | 48 | 125 |
| T | 5 | 5 | 5 | 5 |
| w,h | 10 | 15 | 15 | 8 |
| c | 1 | 1 | 1 | 1 |
| max epoch | 100 | 100 | 100 | 300 |
| batch size | 32 | 32 | 32 | 128 |
| learning rate | 0.05 | 0.05 | 0.05 | 0.05 |
| W, H | 40 | 60 | 60 | 80 |
| number of training objects | 6 | 3 | 3 | 2 |

Here, $K = 3$ is the number of levels, where the first level (bottom) is the output of the tokenizer and only extracted features are represented at the bottom level. Therefore, we consider the second level as the part-level and the third (top) level as the whole-level. $T$ is the number of iteration steps.

$d$ is the dimension of representation at each column and $w, h$ are the width of the grid mesh of columns (shared among levels). $n = w * h$ is the total number of columns. $c$ is the input channel number, since the datasets are all binary we choose the input channel number to be 1. $W, H$ are the size of the image. The max epoch = 100 is also consistent with the pre-training process in the original paper. The evaluation is conducted after 100 epochs of the contrastive pre-training.

The following section presents a detailed description of the key networks employed in Agglomerator. The network structure of the convolution tokenizer is outlined in Table 5. Within this table, the variable $e_d$ denotes the embedding dimension of Agglomerator, which is specifically set to 12, 12, 5, and 12 for SHOPs, Squares, Double-Digit-MNIST, and Ts datasets, respectively. The output of the convolution tokenizer network is subsequently rearranged and fed into the bottom-up and top-down column networks, which exhibit the structure presented in Table 6. Here, the variable $d_1$ in the bottom-up column network is set to 96, 96, 250, and 96 for the four datasets accordingly; $d_1$ in the top-down column network is set to 48, 48, 125, and 48. $d_2$ in the bottom-up network is set to 384, 384, 1000, and 384 respectively; $d_2$ in the top-down network is set to 192, 192, 500, and 192. Additionally, $n_p$ signifies the number of patches (same as $n = w * h$ in Table.4) and takes the values 225, 225, 64, and 100, respectively. Lastly, the structure of the network utilized in contrastive learning is depicted in Table 7. In this context, $p_d$ represents the patch dimension and assumes the values 48, 48, 125, and 48 for SHOPs, Squares, Double-Digit-MNIST, and Ts datasets, correspondingly. It is worth noting that the $n_p$ values remain consistent with the settings in column networks.

Table 5: Parameter details for convolution network

| Network Structure |
|---|
| Conv2d(1, $e_d$ // 2, kernel_size=(3, 3), stride=(2, 2), padding=(1, 1), bias=False) |
| BatchNorm2d($e_d$ // 2) |
| ReLU() |
| Conv2d($e_d$ // 2, $e_d$ // 2, kernel_size=(3, 3), stride=(1, 1), padding=(1, 1), bias=False) |
| BatchNorm2d($e_d$ // 2) |
| ReLU() |
| Conv2d($e_d$ // 2, $e\_d$, kernel_size=(3, 3), stride=(1, 1), padding=(1, 1), bias=False) |
| BatchNorm2d($e_d$ // 2) |
| ReLU() |
| MaxPool2d(kernel_size=(3, 3), stride=(1, 1), padding=(1, 1), bias=False) |

Table 6: Parameter details for column network

| Network Structure |
|---|
| LayerNorm($n_p$) |
| Conv1d($d_1$, $d_2$) |
| GELU (for bottom-up network) or Siren (for top-down network) |
| LayerNorm($n_p$) |
| Conv1d($d_2$, $d_1$) |

Table 7: Parameter details for contrastive head

| Network Structure |
|---|
| LayerNorm($p_d$) |
| Dropout(p=0.3) |
| Rearrange('b n d - b (n d)') |
| LayerNorm($n_p * p_d$) |
| Dropout(p=0.3) |
| Linear($n_p * p_d$, $n_p * p_d$) |
| LayerNorm($n_p * p_d$), |
| GELU() |
| LayerNorm($n_p * p_d$) |
| Dropout(p=0.3) |
| Linear($n_p * p_d$, 512) |

### A.5.6 DOWN-SCALING OF THE GROUND TRUTH

In the Agglomerator, original $W * H$ images are firstly tokenized into $n = w * h$ patches, with $w = W/4$ and $h = H/4$. These tokenized embeddings are treated as bottom-level column representations. Due to the down-sampling effect of the tokenizer, the number of columns is smaller than the original pixels. Further, since the Agglomerator is super-computationally expensive, scaling as $O(w^4)$, reserving the original dimensionality of images ($w = W, h = W$) is not computationally plausible. Therefore, we impose down-sampling to the ground truth with the same reduction ratio, so that the dimension of column embedding ($w \times h \times d$) matches the down-sampled ground truth ($w \times h$). The down-sampling is based on majority rule.

### A.5.7 THE FAILURE OF THE AGGLOMERATOR

As shown in Table.3, the Agglomerator failed in all cases. As far as we know, in the original paper of the Agglomerator (Garau et al., 2022), the representation is not quantitatively evaluated in terms of the hierarchical structure but for classification accuracy and object detection. Also, in the visualization, the representation across levels is more likely to extract features at different scales, instead of forming an interpretable part-whole hierarchy. As far as we understand, although the motivation and basic idea are very promising, it is questionable whether the Agglomerator is capable of capturing hierarchical object-centric representation as a part-whole hierarchy at all. As shown in Table.3, it is worth re-evaluating the parsing ability of the Agglomerator on images with more 'explicit' part-whole relationships and more appropriate 'quantitative metrics' as we do.

Another explanation is the symmetry in our dataset. In the original paper, the seeming parsing of different parts is likely due to the different colors associated with the parts. In other words, It is observed that the Agglomerator only groups locations of similar color into islands (like feature extraction), instead of parsing the object based on knowledge of the object-centric representation (e.g. same object can have different colors and different objects can share the same color). To put it in another way, the grouping in the Agglomerator is due to the external asymmetry in the scene, e.g. different colors. However, in our dataset, all objects have the same color (black), and the symmetry-breaking process for grouping needs to occur internally. Therefore, the symmetry (shared color among objects) can challenge the Agglomerator to parse the scene.

Table 8: time constant of the Composer

| Dataset | Ts | Squares | SHOPs | Double MNIST |
|---|---|---|---|---|
| $T$ | 3000 | 3000 | 3000 | 3000 |
| $\tau_d$ | 80 | 75 | 42 | 16 |
| $\tau_{\delta 1}$ | 36 | 36 | 20 | 16 |
| $\tau_{\delta 2}$ | 35 | 35 | 20 | 15 |
| $\tau_{r1}$ | 15 | 24 | 12 | 16 |
| $\tau_{r2}$ | 14 | 24 | 12 | 15 |
| $\tau_1$ | 2 | 2 | 3 | 2 |
| $\tau_2$ | 6 | 12 | 6 | 8 |
| $\tau_D$ | 18 | 30 | 10 | 8 |
| $\tau_\Gamma$ | 15 | 16 | 8 | 8 |
| $\tau_{d'}$ | 80 | 75 | 42 | 16 |

On the other hand, failure on our synthetic dataset and metrics neither excludes its potential validity in other cases nor excludes the possibility that it can be improved to solve the problems. Compared with our model, the Aggglomerator's architecture is more flexible in dealing with real-world images, and training on larger real-world datasets can be very different from training on small-scale synthetic datasets. Therefore, the limitation of our dataset and metric is also notable. Besides, the downsampling of the ground truth is likely to magnify the failure of the Agglomerator since the Silhouette Score is more sensitive to incoherence when the sample number is lower. It is partially the reason why the score tends to be lower than 0. However, the results indeed show that SOTA models can fail when explicitly evaluated.

To put it in a nutshell, we highlight that the problem of representing part-whole hierarchy needs to be more explicitly evaluated to confirm the validity of the model. As far as we know, representing the part-whole hierarchy is far from being solved and the Composer is the first model to deal with the problem implicitly in pure neural networks with fixed architecture.

## A.6 MODEL DETAILS

In this section, we provide more details about the formulation of the model. A zip file containing videos visualizing the dataflow of the model is provided in SI (60MB).

### A.6.1 INITIALIZATION OF THE DYNAMICS

.

Here, Eq.1 and eq.4 in main text are slightly extended to clarify the initialization process:

$$\rho_1(t) = x \cdot (\gamma_1 \cdot \Gamma_1 + r_1(t) \cdot \epsilon_1) \tag{24}$$

$$\rho_2 = (\lambda \cdot x + (1 - \lambda) \cdot D) \cdot (\gamma_2 + r_2(t) \cdot \epsilon_2) \tag{25}$$

where, the term $r_i(t) \cdot \epsilon_i$ is only for random initialization (See Table.9). $\epsilon_i$ is sampled from uniform distribution $U[0, 1]$ and $r_i(t)$ is the temporary amplitude of the noise, which is decayed rapidly along the simulation (decay rate $\sim 0.8$, Table.9). In other words, $r_i(t) = r_i \cdot (0.8)^{-t/\tau_d}, i = 1, 2$, where $r_i$ is the initial amplitude of the noise. During simulation, the noised is delayed every $\tau_d$ time steps for simplicity.

### A.6.2 TIME SCALES

The time scale parameters of the model are shown in Table.8, which has appeared in eq.1 to eq.7 in the main text. $T$ is the entire simulation length. $\tau_d$ is the coupling delay of the top-down feedback inside the column, shared for both part-level and whole level. $\tau_{\delta 1}$ is the total refractory period of part-level spiking neurons and $\tau_{\delta 2}$ is that of the whole level neurons. $\tau_{r1}$ is the absolute refractory

period of part-level spiking neurons and $\tau_{r2}$ is that of the whole level neurons. $\tau_1$ is the integrative time window of the part-level column (from visible SCS to DAE) and $\tau_2$ is that of the whole level column. $\tau_D$ is the integrative time window from the part-level column to the whole-level column. $\tau_\Gamma$ is the time window of the top-down feedback from the whole-level column to the part-level column. $\tau_{d'}$ is the coupling delay of the cross-level top-down feedback from the whole-level column to the part-level column. In this work, we set $\tau_{d'} = \tau_d$ for simplicity. Roughly speaking, we have:

$$\tau_d = \tau_{d'} > \tau_{\delta 1} \sim \tau_{\delta 2} > \tau_{r1} \sim \tau_{r2} > \tau_2 \sim \tau_D \sim \tau_\Gamma > \tau_1 \tag{26}$$

More specifically, part-level and whole-level columns are characterized by two timescale parameters: $\tau_1 < \tau_2$, which determines the timescale (fast or slow) of the intra-column dynamics, which is inspired by the timescale hierarchy along the cortical hierarchy (Mahjoory et al., 2019).

If we take each time step as 1 millisecond in the brain, then the refractory period $\tau_\delta$ is around tens of milliseconds and the absolute refractory period $\tau_r$ is around ten milliseconds. The coupling delay is around 50 millisecond (Singer, 2021). The integrative time window matches that of the coincidence detector (several millisecond (König et al., 1996)). The frequency of oscillatory activity is around ten of milliseconds, within the Gamma band (Tallon-Baudry & Bertrand, 1999).

### A.6.3 ABLATION STUDY OF THE TIMESCALE PARAMETERS

In Fig.10b in the main text, we provide the ablation study of the timescale parameters. Here we provide more discussions.

Firstly, the coupling delay $\tau_d = \tau_{d'}$ is most essential for the capability of the Composer. As shown in Fig.10, once removed, the parsing representation fails directly.

Secondly, the refractory period ($\tau_{r1}, \tau_{r2}$) has a secondary effect on the Composer, especially for the whole-level dynamics. Besides, the integration timescales ($\tau_1, \tau_2, \tau_D$) also matters significantly.

As pointed out in Fig.2e in the main text, refractory period and delay coupling are essential to change the attractor dynamics into metastable rhythmic dynamics (equilibrium states into non-equilibrium states). Thus, the removal of these parameters indeed degrades the system.

Thirdly, the removal of the relative refractory period slightly degrades the coordination of the Composer. The explanation is that: Representing the part-whole hierarchy is a combinatorial problem in nature, which needs to be iteratively searched. For example, when the object number increases as in the Ts dataset, the possible configuration of the parse tree gets exponentially larger. However, while hard refractory period forces the system to switch among different states (spike fires at wrong timings), the 'hardness' could prevent efficient self-correcting once the system gets into a wrong state (because the hard refractory period constraints the available next firing timing). Thus, introducing a relative refractory period can help the system jump out of the local minimum, once it 'finds' much better states. It is likely that for this reason, enforcing $g = 0$ in Fig.10 in the main text slightly degrades the Coordination Score.

For additional results of ablation study on other datasets, See Fig.20

### A.6.4 OTHER HYPER-PARAMETERS

Table.9 shows other parameters of the Composer (The parameter for the evaluation can be found in Table.1).

### A.7 TRAINING DETAILS

### A.7.1 RESOURCES

Our experiments have been performed on ubuntu 16.04.12 with devices: CPU (Intel(R) Xeon(R) CPU E5-2640 v4 @ 2.4GHz) and 4×GeForce RTX 2080 Ti. The python version is 3.6.3.

Table 9: Other hyper-parameters of the Composer. g (eq.2) is the (inhibitory) gating effect of relative refractory period ($\tau_{delta} - \tau_r$), same for whole-level and part-level for simplicity. $\lambda$ in eq.4 describes the skip connection. noise decay, $r_1, r_2$ describes the initialization process (eq.24).

| Dataset | Ts | Squares | SHOPs | Double MNIST |
|---|---|---|---|---|
| $g$ | 0.5 | 0.3 | 0.3 | 0 |
| $\lambda$ | 0.3 | 0.4 | 0.4 | 0.4 |
| noise decay | 0.8 | 0.8 | 0.8 | 0.8 |
| $r_1$ | $\frac{1}{40}$ | $\frac{2}{3}$ | $\frac{1}{9}$ | $\frac{1}{8}$ |
| $r_2$ | $\frac{1}{40}$ | $\frac{2}{3}$ | $\frac{1}{9}$ | $\frac{1}{8}$ |

### A.7.2 NETWORK ARCHITECTURE AND TRAINING HYPERPARAMETERS

The details of training neural networks are shown in Table.10. All networks are trained with stochastic gradient descent (SGD).

### A.7.3 DATASET FOR TRAINING DAE

The details of training dataset are shown in Table.11. Examples of the training data are visualized in Fig.18.

### A.7.4 LOSS FUNCTION

The DAE (either part or whole) are trained to minimize the MSE loss between the output of DAE and original image:

$$loss(x) = (x - DAE_i(\tilde{x}))^2, \quad i = 1, 2 \tag{27}$$

where $x$ is the original single-object image in Section.A.7.3 or Fig.18. $\tilde{x}$ is the denoised version of $x$. Notably, the training of DAE has an unsupervised form and does not provide any explicit information on how to group the tree nodes or to form the parsing tree. These all emerged during the simulation dynamics. All the training does is to provide the minimal prior about what (on average) the object (part/whole) looks like, so that the model could make sense of the multi-object scene at all. It is plausible that such prior also exist in the brain to help parse the scene. For example, before parsing the house (Fig.1a in the main text), a person should have a prior about the door, window, and roof. Such prior should also influence the outcome of the parsing process.

In this paper, we treated those senses of the object (part or whole) as prior knowledge and explored how the parsing structure emerges on the condition of the prior knowledge. While some of the prior may be hard-wired in the brain through evolution, others may also be learned during development. The learning aspect of these priors is not discussed in this preliminary model, but we could provide insight into how it could potentially be achieved: the general architecture in this work, including the explicit separation of columns and hierarchical organization of time-scale constant, could be treated as the inductive bias of the end-to-end training. Instead of training DAE separately, we could treat the entire model as a recurrent spiking neural network and train the model by back-propagation through time (BPTT) (Wu et al., 2018). The loss function is modified minimally: the MSE loss between the multi-whole-object input and the **averaged** (part-level) top-down feedback (eg.$\gamma_1(t)$). Due to the hierarchical temporal structure of the model (inductive bias), it is more efficient to learn a part-whole hierarchy representation to predict the whole image. Then, the single-object prior is possible to be learned in a fully unsupervised manner. We leave it as a promising future work.

### A.8 HYPERPARAMETERS

The (hyper) parameters of the paper are listed as:

(1) DAE related: Table.10

(2) Model related: Table.8 (time-scale constants), Table.9 (initialization)

Table 10: Details of training DAE

| Dataset | encoder | decoder | learning rate | noise | minibatch size | epoch num |
|---|---|---|---|---|---|---|
| Ts (part) | FC(1600, 1000) Sigmoid() | FC(1000, 1600) Sigmoid() | 1e-3 | 0.5 | 16 | 200 |
| Ts (whole) | FC(1600, 1000) Sigmoid() | FC(1000, 1600) Sigmoid() | 1e-3 | 0.5 | 16 | 200 |
| Squares (part) | FC(3600, 400) Sigmoid() | FC(400, 3600) Sigmoid() | 1e-3 | 0.8 | 16 | 200 |
| Squares (whole) | FC(3600, 400) Sigmoid() | FC(400, 3600) Sigmoid() | 1e-3 | 0.6 | 16 | 200 |
| SHOPs (part) | FC(3600, 400) Sigmoid() | FC(400, 3600) Sigmoid() | 1e-3 | 0.7 | 16 | 200 |
| SHOPs (whole) | FC(3600, 400) Sigmoid() | FC(400, 3600) Sigmoid() | 1e-3 | 0.7 | 16 | 200 |
| Double-MNIST (part) | FC(6400, 2000) Sigmoid() | FC(2000, 6400) Sigmoid() | 1e-3 | 0.5 | 16 | 200 |
| Double-MNIST (whole) | FC(6400, 2000) Sigmoid() | FC(2000, 6400) Sigmoid() | 1e-3 | 0.5 | 16 | 200 |

Table 11: Training dataset details

| Dataset | Training size | Input dimension | Object number |
|---|---|---|---|
| Ts (part) | 60000 | $40 \times 40$ | 1 |
| Ts (whole) | 60000 | $40 \times 40$ | 1 |
| Squares (part) | 60000 | $60 \times 60$ | 1 |
| Squares (whole) | 60000 | $60 \times 60$ | 1 |
| SHOPs (part) | 20000 | $60 \times 60$ | 1 |
| SHOPs (whole) | 20000 | $60 \times 60$ | 1 |
| Double-MNIST (part) | 60000 | $80 \times 80$ | 1 |
| Double-MNIST (whole) | 60000 | $80 \times 80$ | 1 |

(3) Evaluation related: Table.1.

(4) Benchmarking related: Table.4

## A.9 BIO-PLAUSIBILITY

In this section, we list and provide detailed discussion about the biological correlates of the design of the network.

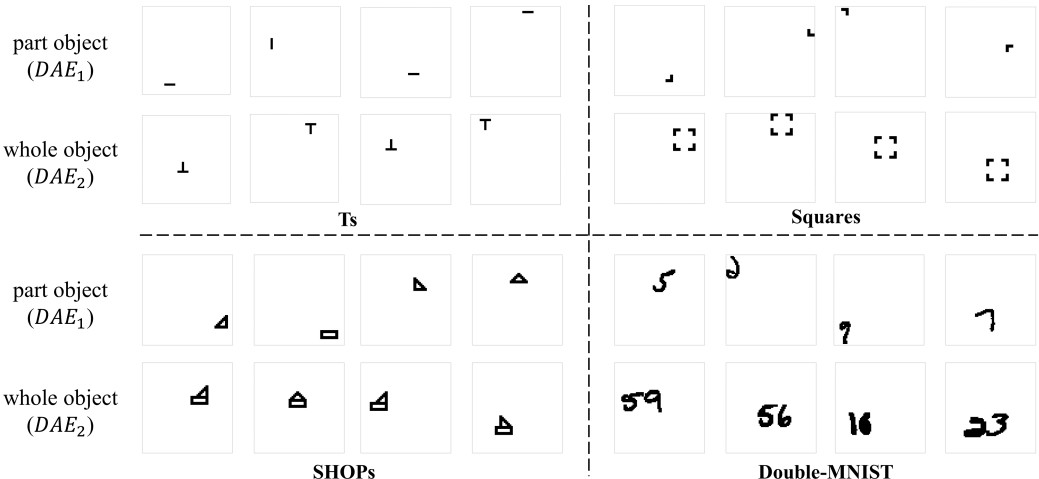

Figure 18: Examples of the training data to train part/whole level DAE.

1. **Delay-coupled oscillatory neural network**: In Singer (2021), the author describes the cerebral cortex as a delay-coupled recurrent oscillator network, which is very different from the architecture in the deep learning field. In this work, such architecture is captured, integrated within the deep learning framework, and acting as an essential ingredient of the mechanism. The delay within the column enlarges the time window of the feedback loop ($\tau_d$), so that alternative cell assemblies could emerge and disappear in order. In other words, the coupling delay makes the system non-Markovian and of infinite dimension (approximately, (Izhikevich, 2006)), so that the coding scheme and the capability of associative memory is much enlarged[5]

2. **Feed-forward and feed-back pathway along the cortical hierarchy**: In general, the cortex is organized into similar columns (Douglas & Martin, 2004), which is composed of six layers from layer I to layer VI. Cortex are spatially organized corresponding to the spatial structure of the physical world and hierarchically organized into levels. These basic features are captured in our model and act as essential elements for representation theory: the representation of the part-whole hierarchy depends on such spatial and hierarchical organization. Notably, in Markov et al. (2013), the author also specifies the organization of the feedforward and feedback hierarchy. In detail, there are recognizable feedforward and feedback pathways between layer II/III of higher and lower level columns. This corresponds to the cross-level interaction between the visible layer in our model. Lastly, Markov et al. (2013) also shows that a long-distance feedforward path from lower level to high level exists in layer IIIb. These are realized as the skip connections from driving input to the whole level visible layer in our model (eq.4). More generally, the feedback from higher levels contains signals originating from both layer II/III and layer V/VI (corresponding to the latent space in the Composer). It is left to future work to study the cross-level interaction between visible SCS and the latent space of DAEs.

3. **Time scale hierarchy**: Along the cortical hierarchy, there is a gradient of timescale hierarchy (Mahjoory et al., 2019)—'*We found that the dominant peak frequency in a brain area decreases significantly, gradually and robustly along the posterior-anterior axis, following the global cortical hierarchy from early sensory to higher order areas*'. Such time scale hierarchy is exploited in our model as the basis for representing hierarchical inclusion relationships among part-neuronal-groups and whole-neuronal-groups. However, since the frequency spectra are not unlimited, the capability of the part-whole representation may be limited by the range of the total frequency bands. Here, we treat such limitation as a shared weakness of our model and the brain, since the temporal resolution of cross-frequency coupling has shown to be a constraint for the capability of working memory of humans

---

[5]Each memory is realized as a trajectory instead of a single fixed point. The trajectory is the combination of transient fixed points so that the attractive states expand combinatorially or exponentially.

$(7 \pm 2)$ (Nicola & Clopath, 2019). Indeed, human also has a limited range of the hierarchy depth to represent instantaneously ($\sim$ 5 levels) (Hinton, 2021). At least three frequency bands could be explored in the future: gamma band, alpha band, and theta band.

4. **Topographical mapping**: As mentioned above, the spatial organization of the cortical column has a topographical correspondence to the physical world, called the topographical mapping (Eickhoff et al., 2017). Such location-wise representation is exploited in GLOM (Hinton, 2021) as a core basis to represent the part-whole hierarchy and is similarly essential for our model. The topographical relationship enables the representation of objects as grouped features distributed in a local spatial range (identical island of vectors in GLOM). Besides, the location-wise representation also helps to clarify the inclusion relationship between whole and part across the hierarchy, both in our model and in GLOM For example, part neuronal groups should also be spatially aligned to their parents.

5. **Top-down attention as autoencoder**: Predictive coding (Rao & Ballard, 1999) was first proposed by Dana H. Ballard and Rajesh P. N. Rao to explain the extra-classical receptive-field effects in primary visual cortex. Then, the predictive coding theory was mapped to the canonical circuit of cortical circuit (Bastos et al., 2012) and served as a unified theory of brain function (Friston, 2010). In the predictive coding model, the bottom-up and top-down feedback attention is formalized as the autoencoder architecture, and the reconstruction error should be minimized to achieve minimal 'prediction error' or 'surprise'. Such architecture is exploited in our model to realize the inner-column bottom-up/top-down pathways and reconstruction error is minimized as the objective function of training. Interestingly, such predictive feedback is also related to the temporal synchrony in the cortex (Engel et al., 2001).

6. **Sparse code and dense code**: The dual coding scheme in the cortical circuits has been recognized when representing features: ultra-sparse coding in the superficial layer (layer II/III) and dense coding in deeper layer (layer V/VI) (Tang et al., 2018; Wang, 2018). While the latter encodes the statistical aspects of features, the former might additionally encodes the relationships. In this work, the dual coding scheme is realized as the sparse spike coding in visible space and real-valued dense vector coding in latent space, with the synchrony in the visible space additionally encoding the relationship among objects.

7. **Relative refractory period**: Strictly speaking, the absolute refractory period (ARP) refers to the phase immediately after a spike initialization ($\sim$ 2 ms). The later phase where a spike is harder to be triggered (though not impossible) is referred to as the relative refractory period (RRP) (Dayan & Abbott, 2001). If we take 1-time step as 1 millisecond in real-world time, then the absolute refractory period is around 10 milliseconds (Table.8) in the Composer, which is much longer than the strict absolute refractory period. Therefore, the picture should be clarified as follows: the excitability of spiking neurons after a spike increases gradually, in the form of $1 - e^{-t/\tau}$. At the beginning phase, the excitability is low enough to prevent the neuron from firing a second spike given the conventional stimulus strength, but since the excitability increases rapidly during this phase, the relative period length is small compared to the whole refractory period. This beginning phase where excitability is low enough compared to the stimulus strength in our experiments but increases fast is treated as an absolute refractory period. In contrast, during the rest period, the excitability has recovered to the extent that neurons might generate a second spike but with a much lower probability. Since the recovery is much slower during the second phase, the temporal range is much longer than that of the first phase. This slow recovery phase is modeled as the relative refractory period in this work. The total refractory period can expand from tens of milliseconds to much longer, depending on the channel type on the axon of the neuron (Gerstner et al., 2014). On the other hand, it is also conventional in numerical modeling that the absolute refractory period is modeled no less than 5 ms. In sum, the time scale of refractoriness fits the biological systems.

8. **Dentritic computation of pyramidal cell**: The driving signal and modulatory signal are distinguished in the cortical circuit (Lee & Sherman, 2010), where the driving signal acts on the proximal site of dendrites and the modulatory signal acts on the distal sites (Spruston, 2008). The two types of inputs interact in a non-linear way. Such non-linear interaction between driving input and modulatory input is captured as the multiplication between the bottom-up integration and top-down modulation, realized as the pyramidal cell in the visible

space (Fig.3ab in the main text). Such a gating effect inside the column is essential for the binding of neuronal groups and the gating effect across levels is essential for the coordination of neuronal groups.

9. **Coincidence detector**: Abeles (1982) argued that cortical neurons in superficial layers are coincidence detectors, which detect sparse synchronous events within a narrow time window. In our model, the time constant of the integrative time window is small ($\tau_1 \sim 2ms, \tau_2 \sim 5ms$). As a result, the inner-level bottom-up integration of spiking activity in the superficial (visible) layer is modeled as coincidence detectors. Such a narrow time window enables two things: (1) stochastic spikes fired at extremely adjacent time steps should be detected as a single event; (2) the temporal resolution of the synchronous event is kept within a small time-scale ($\sim \tau_1, \tau_2$). Both are important to form a high-quality parse tree. Interestingly, a similar concept has also been developed in GLOM (Hinton, 2021), named as 'coincidence filtering'.

10. **Meta-stability of cortical network**: '...*Single-trial analyses of ensemble activity in alert animals demonstrate that cortical circuit dynamics evolve through temporal sequences of metastable states. Metastability has been studied for its potential role in sensory coding, memory, and decision-making. Yet, very little is known about the network mechanisms responsible for its genesis...*' (Mazzucato et al., 2015). In this work, we build such a system of metastable states by integrating the spiking neural network (SNN) and artificial neural network (DAE as ANN) and further demonstrates its computational role in vision.

11. **Neuronal assembly as code words**: '*A widely discussed hypothesis in neuroscience is that transiently active ensembles of neurons, known as "cell assemblies," underlie numerous operations of the brain, from encoding memories to reasoning. However, the mechanisms responsible for the formation and disbanding of cell assemblies and the temporal evolution of cell assembly sequences are not well understood...I suggest that the hierarchical organization of cell assemblies may be regarded as a neural syntax...*' (Buzsáki, 2010). Besides, assemblies are shown to be able to realize arbitrary computation function (Papadimitriou et al., 2019). In this work, we show how assembly transiently formed and disbanded, and be organized into a sequence at each level, and hierarchically organized to express the neural syntax. More generally, various features, even of a continuous nature are represented as neuronal assemblies in the brain (population binary code), this provides the basis to enable the model to deal with continuous features (RGB color) with its coding scheme (Stockman, 2019). The reservoir of neuronal assemblies could be more efficiently realized in neuromorphic devices (Pei et al., 2019). From the viewpoint of the Miehl et al. (2022), our model generate the assemblies by DAE-induced symmetry-breaking.

12. **Temporal binding theory and feature integration theory**: Temporal binding theory (Engel & Singer, 2001) and feature integration theory (Wolfe, 2020) are two mainstream theories to solve the binding problem: how distributed information is bound together to form the whole. The former is based on time coding and neuronal synchrony while the latter is based on top-down attention searching on a spatial map. The temporal synchrony, temporal coding, top-down attention, and spatial map are all captured in this model. Thus it is promising to explore whether it could serve as a canonical model to unify the two theories.

13. **Temporal-spatial theory of consciousness**: '*We postulate four different neuronal mechanisms accounting for the different dimensions of consciousness: (i) "temporospatial nestedness" of the spontaneous activity accounts for the level/state of consciousness as the neural predisposition of consciousness (NPC); (ii) "temporospatial alignment" of the prestimulus activity accounts for the content/form of consciousness as the neural prerequisite of consciousness (preNCC); (iii) "temporo-spatial expansion" of early stimulus-induced activity accounts for phenomenal consciousness as neural correlates of consciousness (NCC); (iv) "temporo-spatial globalization" of late stimulus-induced activity accounts for the cognitive features of consciousness as the neural consequence of consciousness (NCCcon).*' (Northoff & Huang, 2017a). In this work, the nested temporospatial nestedness emerges and indicates the perceptual awareness (eg. recognizing the part-whole relationship), and the temporospatial alignment clarifies the content/form of the scene.

14. **Gamma oscillation and perceptual awareness**: '...*One theory suggests that rhythmic synchronization of neural discharges in the gamma band (around 40 Hz) may provide the necessary spatial and temporal links that bind together the processing in different brain*

*areas to build a coherent percept. In this article we propose that this mechanism could also be used more generally for the construction of object representations that are driven by sensory input or internal, top-down processes...*' (Tallon-Baudry & Bertrand, 1999). In this work, the spiking activity in the visible layer approximately oscillates at the gamma band (tens of milliseconds if each time-step is regarded as 1 millisecond.) The gamma oscillation dynamically groups neurons into object representations (the representation theory in the main text).

15. **Preconfigured brain**: In a recent 'inside-out' framework to view the brain, in Gyorgy Buzaki's words, he says—'...*This is the organization I call the preformed or preconfigured brain: a preexisting dictionary of nonsense words combined with internally generated syntactical rules. The neuronal syntax with its hierarchically organized rhythms determines the lengths of neuronal messages and shapes their combinations. Thus, brain syntax preexists prior to meaningful content..."Preconfigured" usually means experience-independent. The backbone of brain connectivity and its emerging dynamics are genetically defined. In a broader sense, the term "preconfigured" or "preexisting" is also often used to refer to a brain with an existing knowledge base, ....In the preconfigured brain model, learning is a matching process, in which preexisting neuronal patterns, initially nonsensical to the organism, acquire meaning with the help of experience...*' (Buzsáki, 2019). Thus, the well-trained DAE in this paper could be treated as an essential preconfigured structure due to genetic codes or the life-long calibration of the sensory-action loop. Plasticity may only provide a secondary role to increase the precision of the 'good-enough' model (Buzsáki, 2019).

16. **Plasticity**: One of the designs that may depart from biology is that the connection weights are trained based on a gradient-based method instead of a correlation-based method, like Hebbian rule or spike timing plasticity (Gerstner et al., 2014). However, this could be explained from two points of view. First, as argued above, the well-trained DAE could be regarded as the preconfigured structure which is gradually searched from evolution (amount to stochastic gradient-based search). Second, since the DAE structure in this model is relatively simple, the training objective (minimizing reconstruction error, the difference between input and feedback) could be interpreted as increasing the correlation between sensory neurons and modulatory neurons, so that the gradient-based training equals correlation-based plasticity. Indeed, Melchior & Wiskott (2019) shows that gradient-based learning and Hebbian plasticity can be unified. Further, we could imagine that there is a two-stage learning algorithm, like the wake-sleep cycle: during the day, the system infers entities based on learned weight, during the night, the learned objects replay and the system efficiently updates the weight by association, which corresponds to the training phase of the DAE. Similar treatment has also been discussed in GLOM (Hinton, 2021).

17. **Inner-layer recurrent connection**: Another design feature that may depart from the biological brain is that the spiking visible layer itself is not recurrent in our model. However, this could also be explained from at least two points of view. First, the feedforward and recurrent connection usually have different functional roles in the cortical circuit, and have different levels of domination. For example, layer IV in the visual cortex are mainly feedforward and the recurrent effect are relatively weak. As a result, the inner-layer recurrence of visible spiking neurons are treated as secondary compared to the recurrence of inner-level top-down feedback or inter-level top-down feedback. So that it is temporally ignored. Further, the localized inner-level recurrence may play a secondary role (different from that of top-down feedback) to speed up the convergence by forming a grid frame (by local connection) to encode the prior of the proximity property of objects. Secondly, the entire column could be recognized as a single layer, with DAE parameterizing the recurrent connection weight among spiking neurons. And the general mechanism still works. In other words, there is no restriction to view the two-level system as a column or a layer. In either case, the models maintain their bio-plausibility.

18. **Polychronization** refers to the generalization of absolute synchronization into structured asynchrony. As argued in Izhikevich (2006), due to the heterogeneity and conduction delay of the neural system, polychrony is more plausible than absolute synchrony. While the externally observed spike firing time is asynchronous, the arriving time of asynchronous spikes to downstream readout neurons is (internally) synchronous. In other words, the shift in spiking time is balanced out by the shift in conduction delay. According to Buzsáki (2010),

the more rigorous definition of cell assemblies should be based on internal observation (downstream readout neurons) instead of external observation (human observer). Therefore, polychronous representation is in essence also synchronous representation. This is the biological basis why we ignore the slight fixed shift in our model.

## A.10 DETAILS ON EXPERIMENTS AND ADDTIONAL RESULTS.

### A.10.1 THE PERTURBATION STUDY

The detailed discussion on Fig.5 in the main text (the perturbation study) is provided in Section.A.4.5.

### A.10.2 THE BENCHMARKING

The detailed discussion on Fig.9 in the main text (the benchmarking) is provided in Section.A.5.

### A.10.3 LOSS VS SCORE

In Fig.10a, we conduct ablation studies to the DAE module in order to find out the relation between the parsing score and the total training loss of part-level and whole-level DAEs. We randomly selected 100 learning rates from $(10^{-3}, 1)$ and for each selected learning rate we trained one part-level DAE and one whole-level DAE. So there are 100 part-level DAEs and 100 whole-level DAEs for each dataset (100 DAE pairs of different denoising capabilities). Then we evaluate the parsing score of the Composer equipped with each of the 100 DAE pairs. Specifically, the x-axis in Fig.10a is the summed loss of both $DAE_1$ and $DAE_2$ that are trained with the same randomly selected learning rate. During the evaluation, exceptional data points where loss gets unreasonably large due to sick learning rate are removed.

Lastly, we find that the overall relationship between the DAEs and Scores is consistent across datasets and not closely dependent on which DAE is used for comparison. For example, we show more results in Fig.19. The relationship is consistent across different cases. For this reason, we show one of the results (Fig.19)a in the main text without losing generality.

Taken together, the positive relationship between lower denoising loss and higher scores indicates that there are direct interplays between the DAE and the parsing ability of the Composer.

### A.10.4 ABLATION STUDY OF TIMESCALE PARAMETERS

The detailed discussion on Fig.10b in the main text (ablation study of timescale parameters) is provided in Section.A.6.3. The ablation studies on all datasets are shown in Fig.20.

In the followings, we provide details on other experiments.

### A.10.5 CONVERGENCE

In Fig.8, we show the convergence of scores along the iteration. 100 randomly selected samples are used,and the score are evaluated every 100 time steps (so 3000/100=30 data point in total for each score).

Convergence on the SHOPs dataset achieves the best overall results. However, the potential overlap of part-level objects when composing the whole-level object imposes additional challenges on recognizing the part-level objects, indicated by the relatively lower Part Score in Fig.8a.

Convergence on the Squares dataset is very interesting. On the one hand, the Whole Score takes the lead all the time, indicating that global information is firstly recognized by the Composer, which is very similar to human vision (Lee & Nguyen, 2001) and is also consistent with Gestalt psychology. On the other hand, the Part Score and Coordination Score undergo an initial descending period before going up. Here, we explain this phenomenon: Compared with the very starting phase, where spikes are randomly and densely fired, the emergence of whole-level squares around $500 \sim 1000$ time steps provide new conditions on the part level. While this has benefits in the long run, it could degrade the representation in the short run, because the Composer needs to rethink its representation and make modifications. For example, the part-level firing becomes sparser, and there are more incorrect synchronizations. This may degrade the part-level grouping and coordination. In other words, the fact

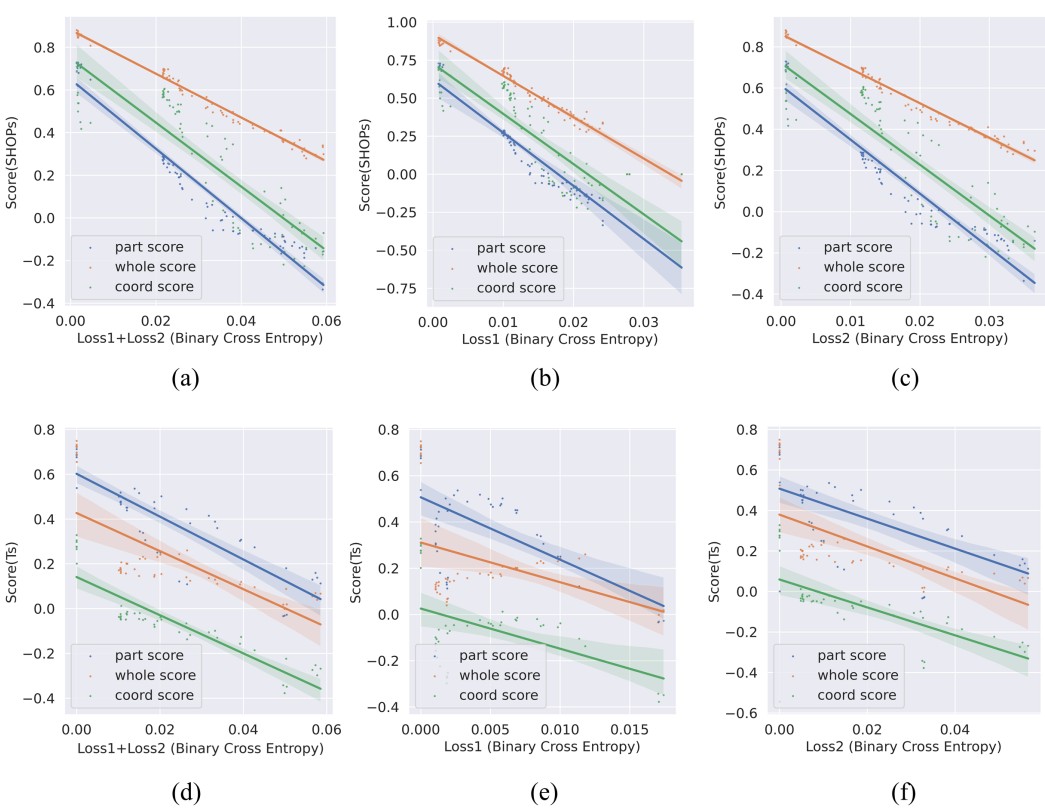

Figure 19: Loss vs Score. More results. (a)(b)(c) results on SHOPs dataset; (d)(e)(f) results on Ts dataset. (a)(d) Relations between scores and total loss of part / whole level DAE ($loss_1 + loss_2$); (b)(e) Relations between scores and total loss of part-level DAE ($loss_1$); (c)(f) Relations between scores and loss of whole-level DAE ($loss_2$); All results are consistent.

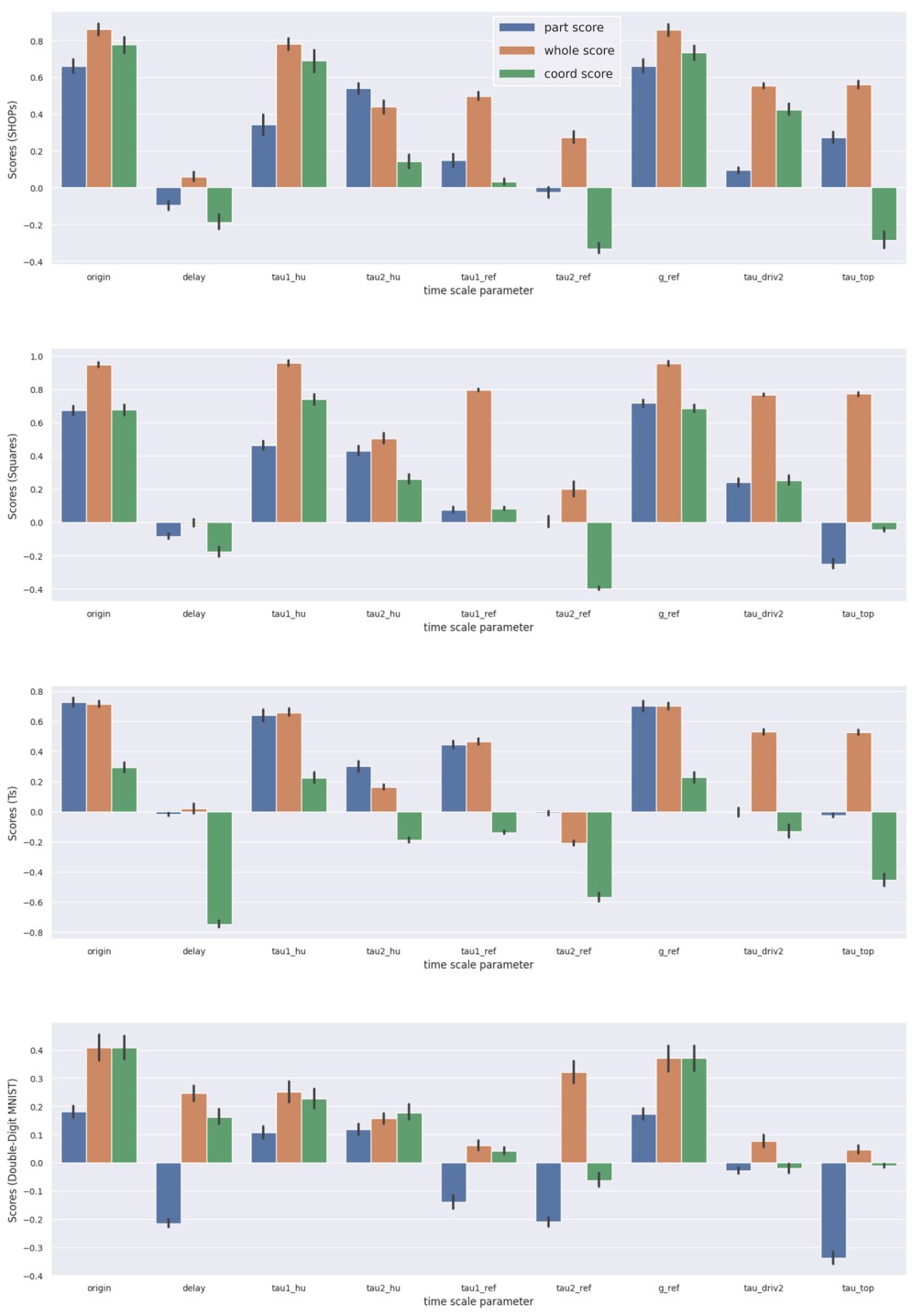

Figure 20: Ablation study on all datasets same as Fig.10b in main text.

that each whole object is composed of four parts complicates the self-correcting / searching process, after the whole-level objects are recognized. Fortunately, after a short period of self-correcting, the Scores go up again and gradually converge to expected synchrony as in other cases.

Convergence on the Ts dataset is very challenging due to the object number is much larger. On the one hand, the Composer needs to distinguish 6 wholes and 12 parts. On the other hand, 6 whole objects and 12 part objects impose $6^12$ potential combinations of part-whole relationships (each part can choose to belong to one of the six wholes). Therefore, it takes time to search for / sample the optimal configuration. Even if the parts/wholes are grouped by synchrony, there is a high possibility that the parts and wholes are not well coordinated. Since the neural computation in the brain can also be regarded as sampling (Buesing et al., 2011), these challenges may also cause problems in perception like the binding problem (Engel & Singer, 2001; Von der Malsburg, 1999). Therefore, on the Ts dataset, the Coordination Score lags behind the other scores.

Convergence on Double-Digit MNIST is also challenging for the Composer because the objects are of much higher diversity. Therefore, it is harder for the Composer to clearly distinguish the objects and to form well-synchronized neuronal groups. Therefore, the Part Score is lower than other scores and the variance is higher than in other cases. However, it is surprising that the Composer still achieves good parsing, indicated by the convergent Coordination Score, even though objects are less recognizable.

### A.10.6 VISUALIZATION

In Fig.6 in the main text, we visualize the spiking pattern, attention map, and local field potential along the convergent process. To better visualize the neuronal group, we reorder the index of neurons on the y-axis (Fig.6c) so that neurons encoding the same object are close on the y-axis. Besides, in order to distinguish different neuronal groups, we color the spikes fired by neurons based on the ground truth assignment of the neurons, so that the color of the neuronal groups indicates what object the neuronal group represents. In other words, the represented object can be directly read by comparing the color of the neuronal group and ground truth. This fact can be verified by comparing the circled neuronal group in Fig.6c, the circled zoomed-in spike pattern in Fig.6d, and the circled object in the ground truth (Fig.6a). It is clear that the synchronized neuronal group gradually emerges from randomness along the simulation. Each synchronized neuronal group represents the parts/wholes of the object. Neuronal groups at different levels are coordinated properly according to the part-whole relationship.

In Fig.6e, it is also observed that different types of top-down attention also emerge into structured patterns. To keep consistent with Fig.6c, the neuron indexes are also reordered and the attention map is also colored based on the ground truth. The depth of the color reflects the value of the attention map. The structured pattern has the same order as the spiking pattern, yet of long timescales. This indicates that attention plays a role in modulating the spike timing in SCS. However, such modulation is not single-way, but a iterative interplay between bottom-up integration and top-down modulation. Therefore, both DAE and SCS play essential roles in solving the parsing problem.

In Fig.6, we also shows the emergence of the oscillatory LFP at the part level, which is the summed top-down feedback: $LFP_1(t) = \sum_{i=1}^{N} \gamma_{1i}(t)$, where $i$ is the neuron index in the part level.

### A.10.7 MORE VISUALIZATIONS

In Fig.7, we briefly show the visualization results on other datasets. Here we provide more detailed visualization results on the four datasets Fig.24 to Fig.28. Two cases are provided for each dataset, including one normal case and one fail case.

We also provide a zip file containing videos to visualize temporal evolution of neuronal activities in SI, about 60MB.

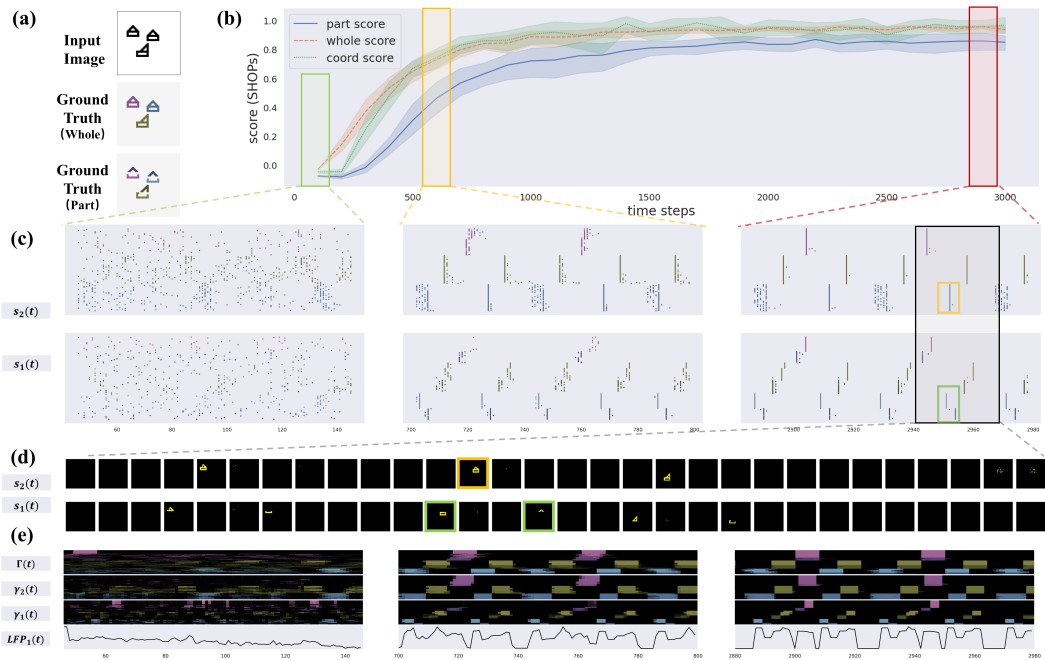

Figure 21: Visualization on SHOPs dataset, same as Fig.6 in main text, but for a different sample.

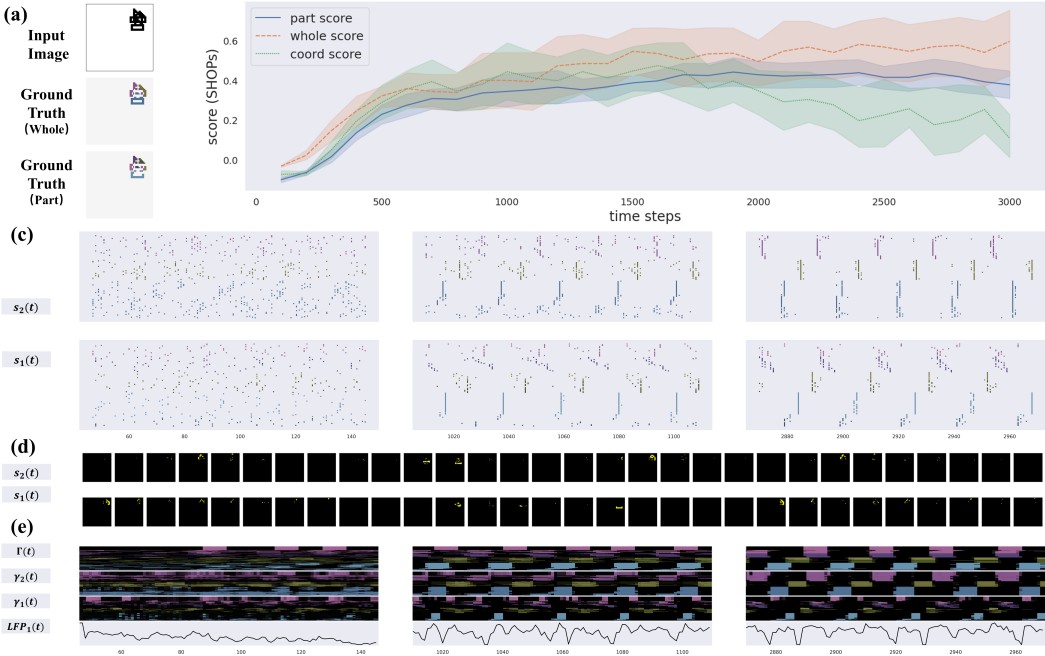

Figure 22: Visualization on SHOPs dataset, same as Fig.6 in main text. The fail case, when objects sickly overlap

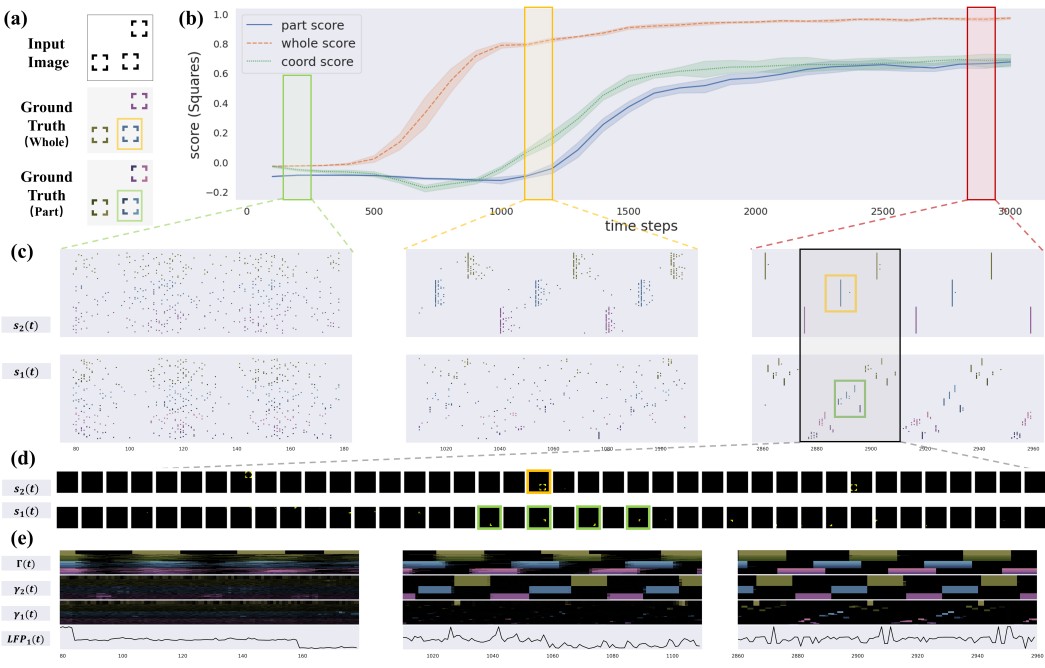

Figure 23: Visualization on Squares dataset, same as Fig.6 in main text. Squares are not overlap.

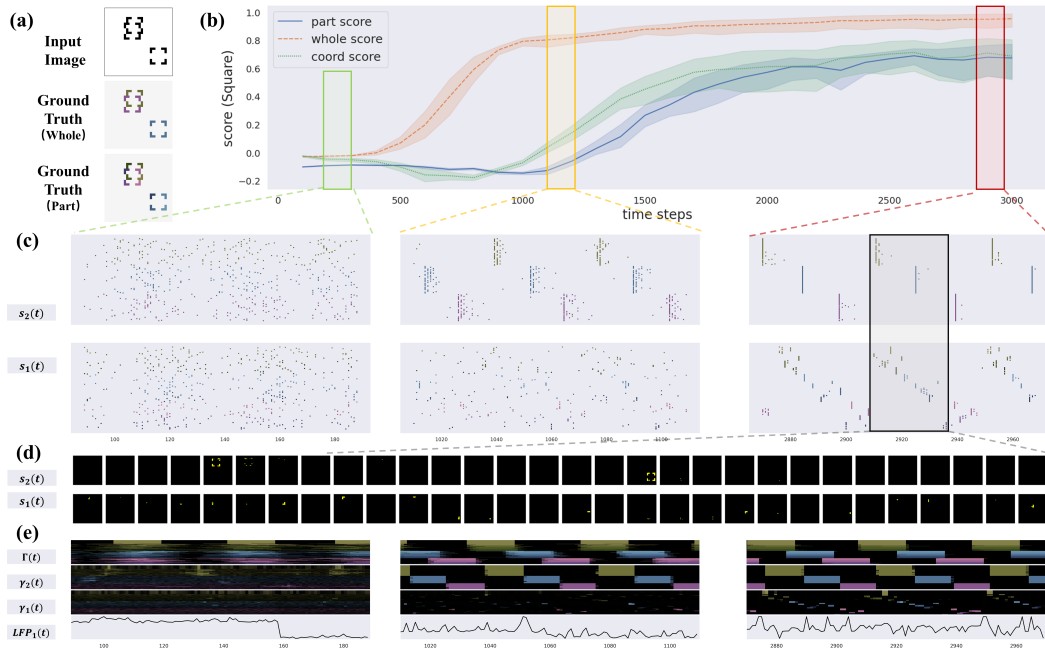

Figure 24: Visualization on Squares dataset, same as Fig.6 in main text. Two Squares heavily overlap.

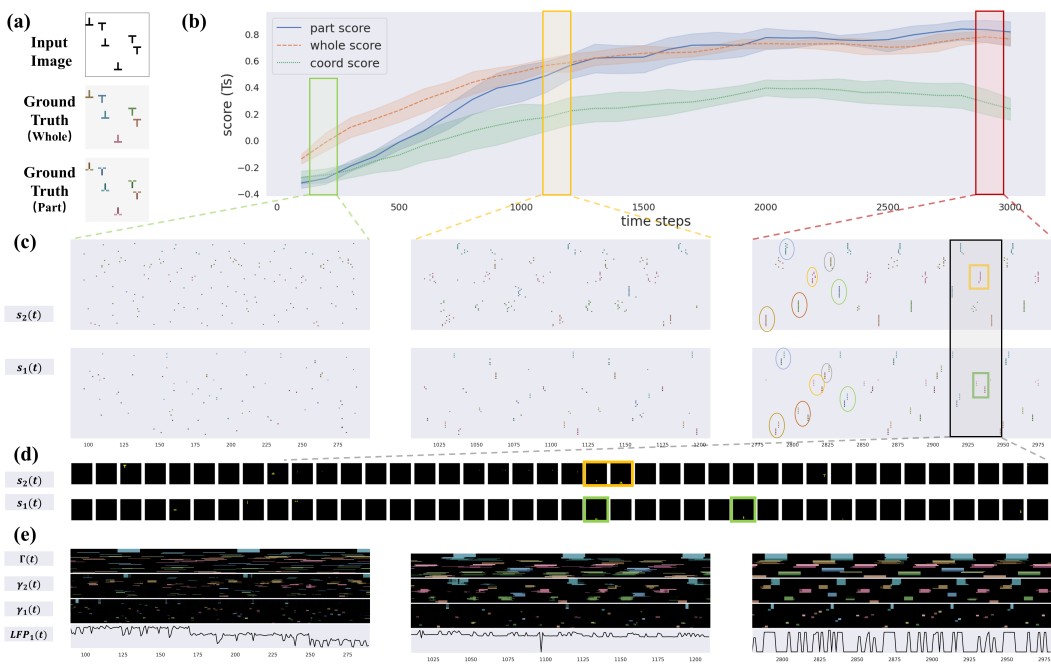

Figure 25: Visualization on Ts dataset, same as Fig.6 in main text. colored circled indicates the coordinated neuronal groups. Same color indicates the part-whole relationship.

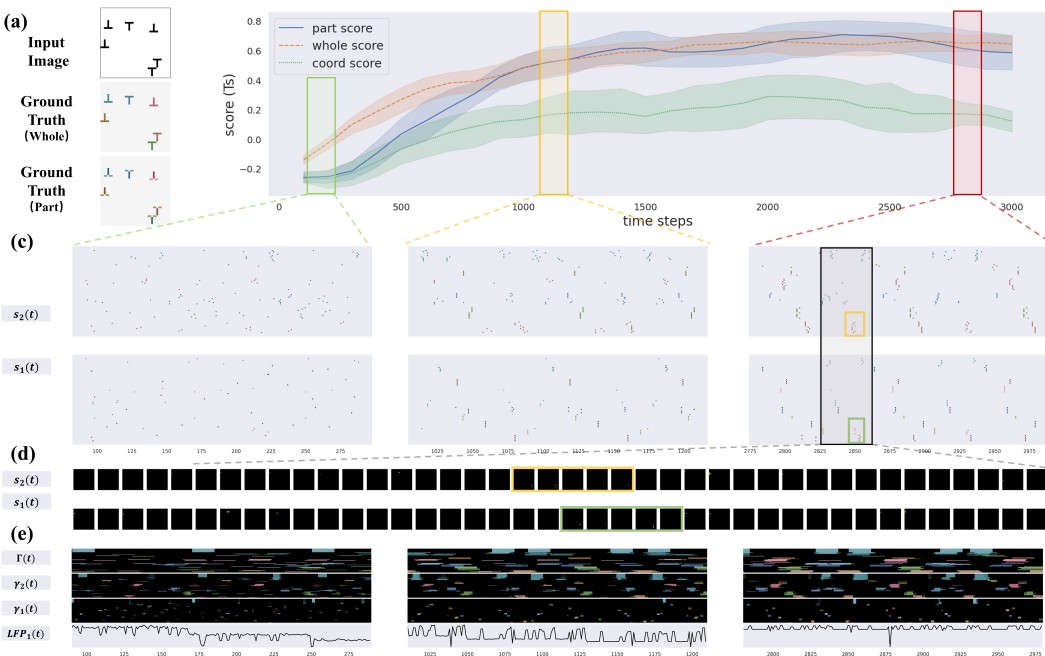

Figure 26: Visualization on Ts dataset, same as Fig.6 in main text. Coordination is not clear as fig.25.

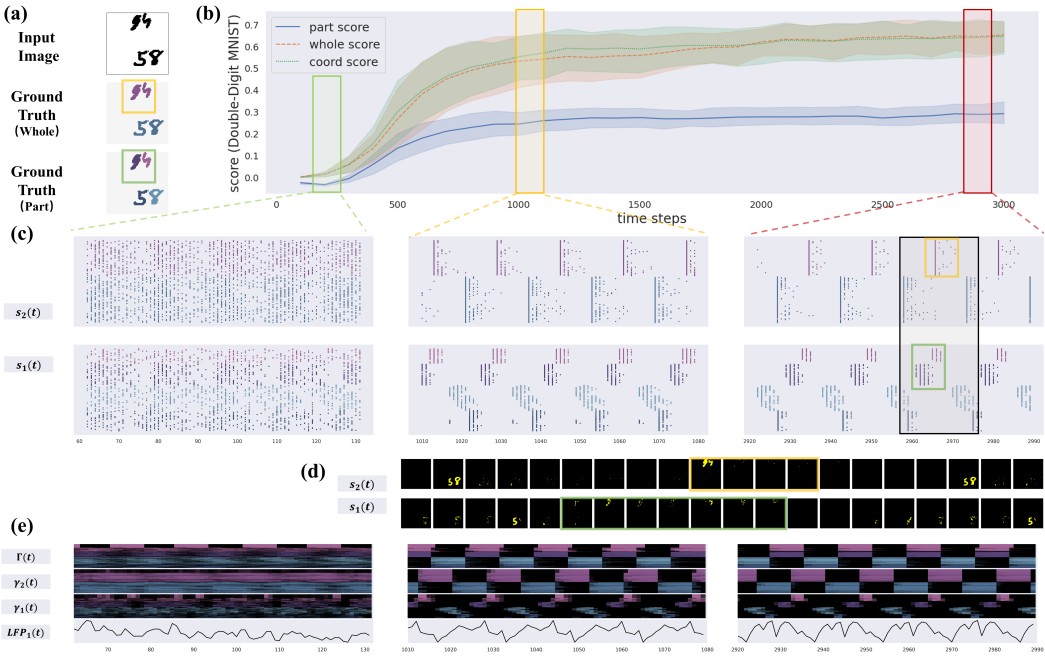

Figure 27: Visualization on Double-Digit-MNIST dataset, same as Fig.6 in main text.

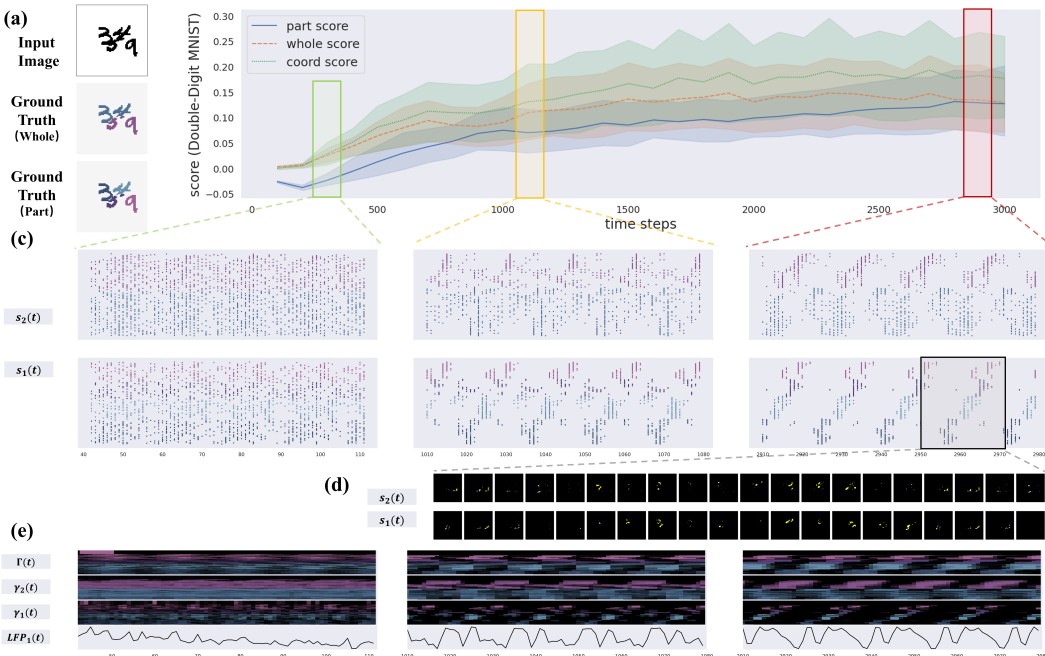

Figure 28: Visualization on Double-Digit-MNIST dataset, same as Fig.6 in main text. Digits are crowded.

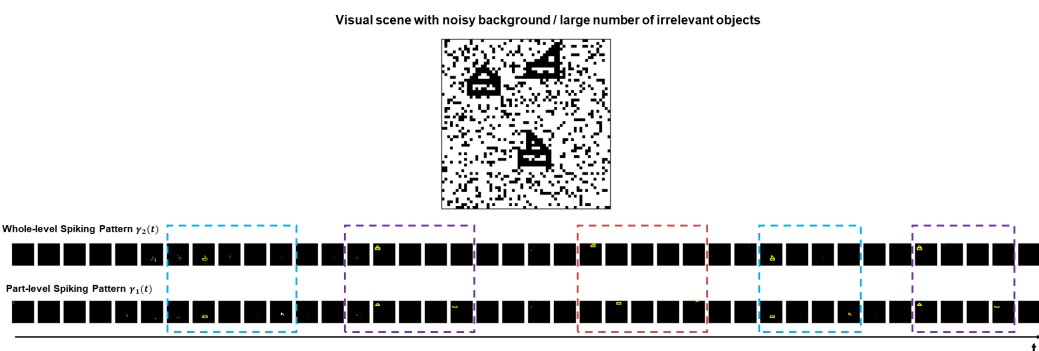

Figure 29: Top, the input image with salt&pepper noise as background noise (p=0.1); Bottom, the spiking pattern of part level (bottom) and whole level (middle). It can be seen that while the noise is visible in the input, the representation in SCS (middle and bottom) is not as noisy.

## A.11 FIGURES FOR REBUTTLE

Q. **With noisy background**. The result when there are background or more objects and parts than fit into different phases.

A. As shown in Figure.29, the model works pretty well when there are background noise or many irrelevant objects. Actually, the performance of the model should not be affected by the irrelevant information, because there are selective attention mechanisms (by DAE) to filter those, and focus on a number of objects of interests. Besides, 'fit' is not very accurate here, because the rhythms and synchrony are emergent "reference frame" that group features into objects, instead of predefined slots, where the number of objects need to fit the slot number.

Q. **Parameter sensitivity test**. "....if the latter case, how sensitive are the results to specific values of the parameters. While the ablation study is nice, setting the various values to zero is somewhat dramatic and uninformative, and I'm rather wondering how precise the parameter values must be, e.g., $\tau_r$, for the network to work.....Is there no parameter tuning? how does the "whole"-level population naturally fire with longer periods (e.g., Fig7b?)? Or is it very sensitive to specific parameter values (e.g., delay and refractory timescales), and if so, are the findings of the study generalizable to either learning something about the brain or improving practical ML algorithms?"

A. As shown in Fig.30, the score is relatively robust with the perturbation of the parameters, as long as the parameter is within the suitable range described by eq.26 in Appendix. Due to the limited time and computing resources during rebuttle, we can only provide the complete sensitivity results on SHOPs dataset for illustration.

### A.11.1 COPIED FIGURES RELATED TO ABLATION STUDIES

Here, We collect the Figures and discussions in Appendix that related to the ablation study, so as to be more available to the reviewer.

**Ablation of spiking modules**

In Fig.10b in the main text (same as Fig.31top), we provide the ablation study of the timescale parameters. Here we provide more discussions. The ablation studies on all datasets are shown in Fig.31.

Firstly, the coupling delay $\tau_d = \tau_{d'}$ is most essential for the capability of the Composer. As shown in Fig.31, once removed, the parsing representation fails directly.

Secondly, the refractory period ($\tau_{r1}, \tau_{r2}$) has a secondary effect on the Composer, especially for the whole-level dynamics. Besides, the integration timescales ($\tau_1, \tau_2, \tau_D$) also matters significantly.

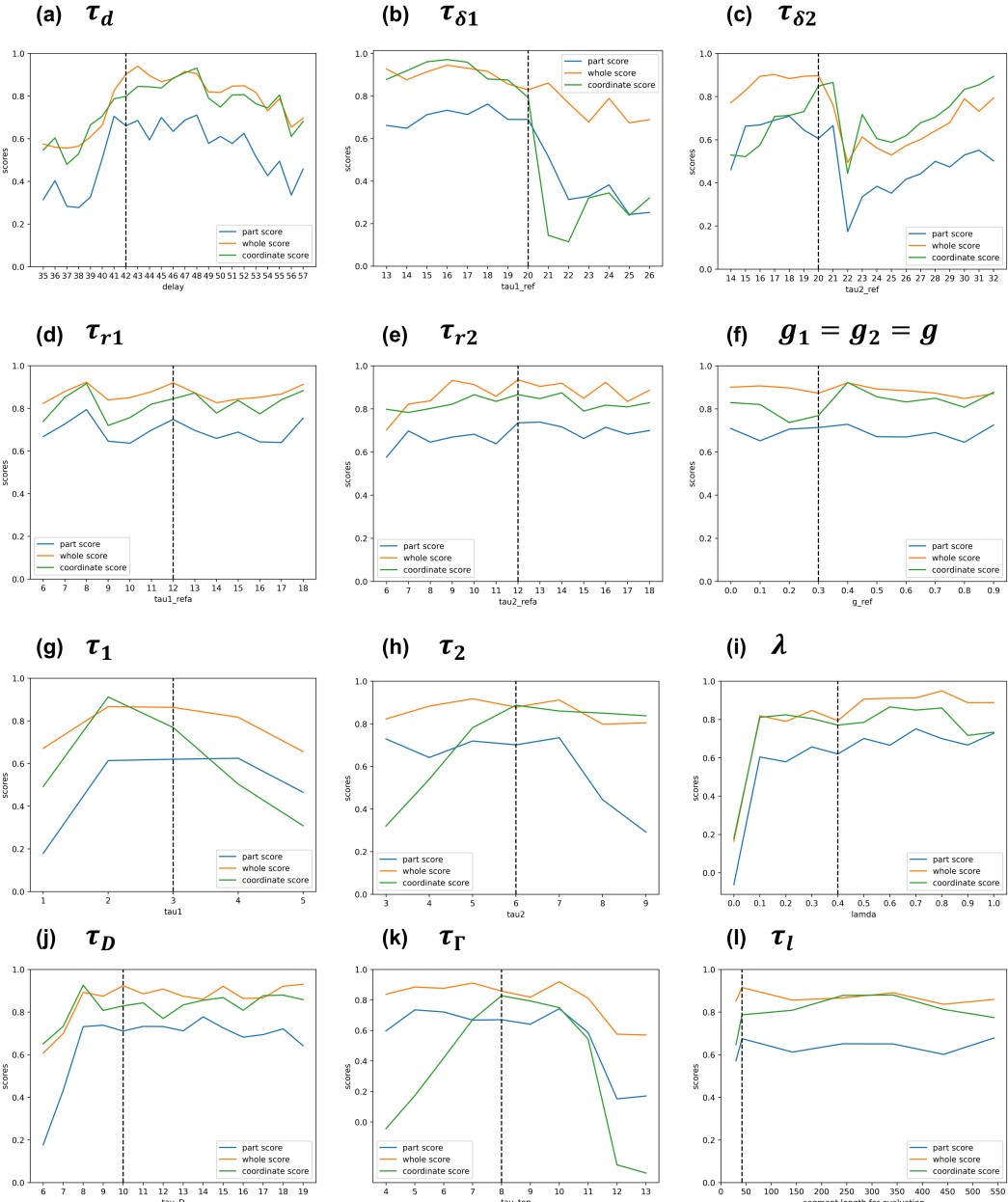

Figure 30: The sensitivity test on SHOPs dataset. Black-dashed line indicates the value used in the main text. The value of parameters are perturbed to show how parsing degrades w.r.t parameter change. (a) The delay parameter of DAE feedback, same for part / whole level; (b) entire refractory period for part-level; (c) entire refractory period for whole level; (d)absolute refractory for part level; (e) absolute refractory for whole level; (f) the inhibitory effect of the relative refractory function; (g) the integration time window for part-level spiking neurons; (h) the integration window for whole-level spiking neurons; (i) the factor of the partial influence from skip connection; (j) the integration time window from part-level to whole level; (k) the integration time window from whole-level to part level; (l) the length of (spike train) segment used for evaluating the parsing quality.

As pointed out in Fig.2e in the main text, refractory period and delay coupling are essential to change the attractor dynamics into metastable rhythmic dynamics (equilibrium states into non-equilibrium states). Thus, the removal of these parameters indeed degrades the system.

Thirdly, the removal of the relative refractory period slightly degrades the coordination of the Composer. The explanation is that: Representing the part-whole hierarchy is a combinatorial problem in nature, which needs to be iteratively searched. For example, when the object number increases as in the Ts dataset, the possible configuration of the parse tree gets exponentially larger. However, while hard refractory period forces the system to switch among different states (spike fires at wrong timings), the 'hardness' could prevent efficient self-correcting once the system gets into a wrong state (because the hard refractory period constraints the available next firing timing). Thus, introducing a relative refractory period can help the system jump out of the local minimum, once it 'finds' much better states. It is likely that for this reason, enforcing $g = 0$ in Fig.31 in the main text slightly degrades the Coordination Score.

**Ablation of DAE modules**

In Fig.32, we conduct ablation studies to the DAE modules, both part-level and whole levels, in order to find out the relation between the parsing score and the quality of DAEs.

We randomly selected 100 learning rates from $(10^{-3}, 1)$ and for each selected learning rate we trained one part-level DAE and one whole-level DAE. So there are 100 part-level DAEs and 100 whole-level DAEs for each dataset (100 DAE pairs of different denoising capabilities). Then we evaluate the parsing score of the Composer equipped with each of the 100 DAE pairs.

As shown in Fig.32, the positive relationship between lower denoising loss and higher scores indicates that there are direct interplays between the DAE and the parsing ability of the Composer.

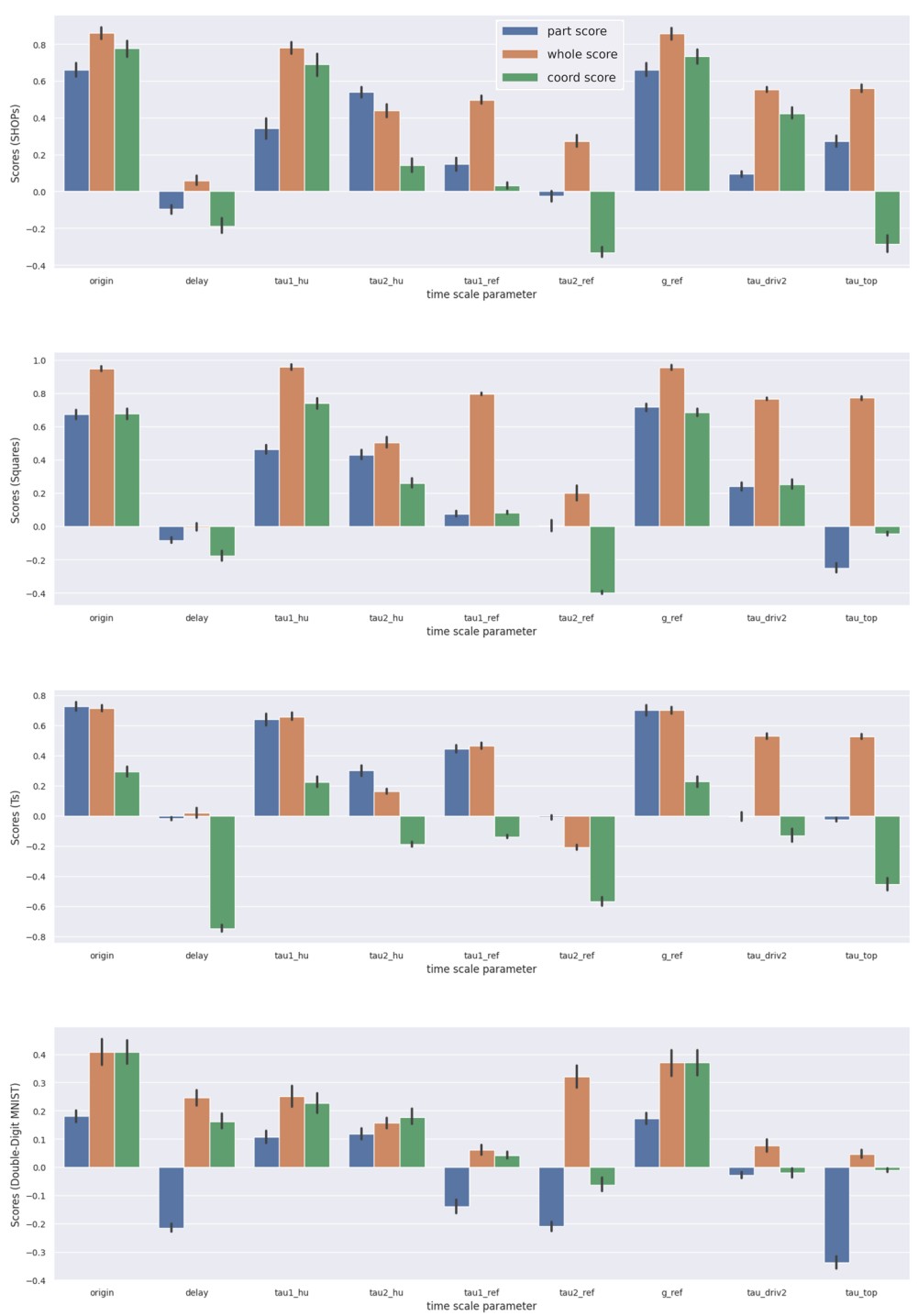

Figure 31: copied from Figure 20 in Appendix. Ablation study on all datasets same as Fig.10b in main text.

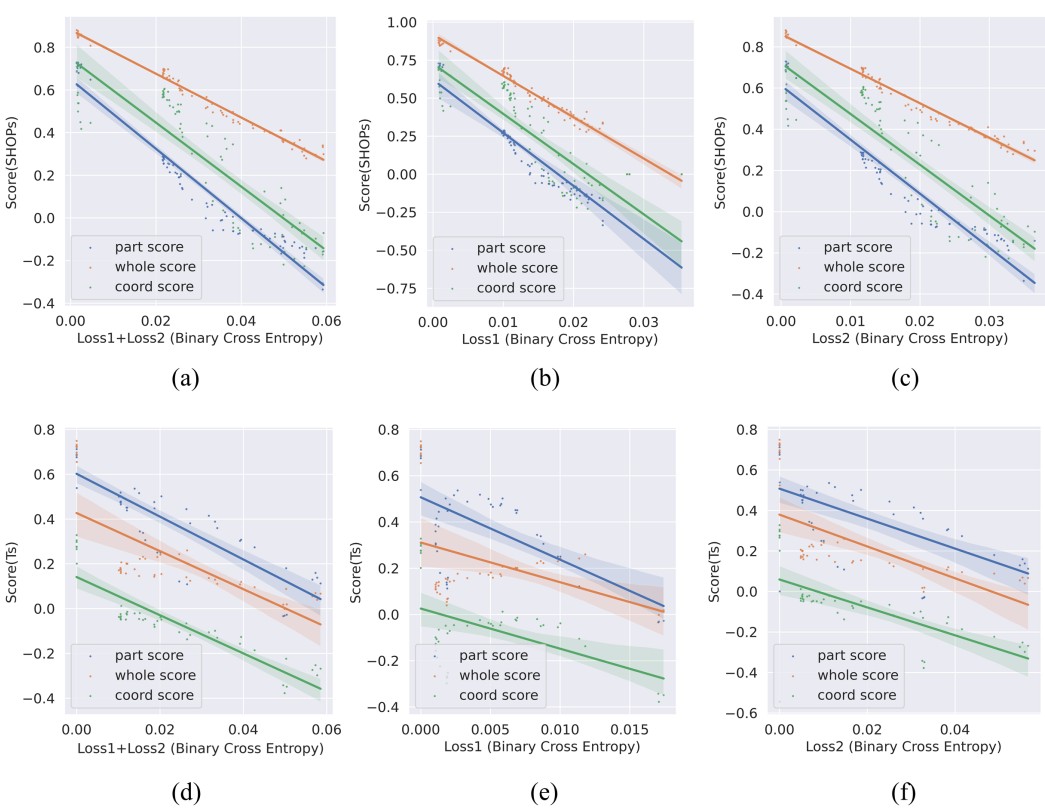

Figure 32: copied from Figure 19 in Appendix. Loss vs Score. More results. (a)(b)(c) results on SHOPs dataset; (d)(e)(f) results on Ts dataset. (a)(d) Relations between scores and total loss of part / whole level DAE ($loss_1 + loss_2$); (b)(e) Relations between scores and total loss of part-level DAE ($loss_1$); (c)(f) Relations between scores and loss of whole-level DAE ($loss_2$); All results are consistent.

