# OpenReview forum: "Representing part-whole hierarchy with coordinated synchrony in neural networks"
_ICLR.cc/2024/Conference — ICLR 2024 Conference Withdrawn Submission_

### Official Review · Reviewer_iE1G · 2023-10-31

**Soundness:** 2 fair
**Presentation:** 3 good
**Contribution:** 1 poor
**Rating:** 3
**Confidence:** 4

**Summary:**

In this manuscript the authors present their COMPOSER scheme how object hierarchies might be represented in biological neural networks by synchrony of neural responses. The synchrony is created by using a denoising autoencoder on the network representation whose output drives neurons one period of the oscillation later. By linking the representations at two levels of a hierarchy the authors can additionally create a rough synchronisation of the parts and the whole representations that are represented at the two depths.

**Strengths:**

Synchrony of responses is still discussed as a possibility for binding in neuroscience and this manuscript implements a model that creates such synchrony for neurons encoding the same object or part. By running a similar process at different levels of the hierarchy a object and part hierarchy can be represented in a fixed neural architecture. And Composer is a spiking neural network with full temporal dynamics.

**Weaknesses:**

I do not think this is a convincing model of processing in visual cortex or a good starting point for machine learning and computer vision models.

From the perspective of artificial systems synchrony based approaches are not really of interest from the start as similarity for inferring objects and parts can be implemented much more efficiently by simply computing the information that is encoded in the phase explicitly.
The scheme proposed here is only evaluated on extremely simple tasks, without any comparisons to other techniques, which is not convincing that the authors’ technique works particularly well, even for these simple tasks.

Additionally, COMPOSER is implemented only for simple displays and simple shallow architectures. As the authors admit it additionally has a bad scaling behaviour because a different phase is required for each part of every object in the scene. For natural scenes this approach is not expected to work. And we have working segmentation algorithms that can do segmentation at any level of the object hierarchy. These approaches appear clearly better suited for computer vision than the one proposed here.

I also think the less relevant aspect of a model of biological vision is not really fulfilled by this model, because of the scaling issues and because this model implies parallel processing of objects everywhere. Clearly humans can deal with natural levels of complexity and it is well documented that object parsing takes time and proceeds sequentially, i.e. humans cannot do segmentation for all objects in a scene at once as this model would suggest. Also, the levels of synchrony observed in the networks here go far beyond anything I have ever seen in a visual cortex.

**Questions:**

The manuscript seemed fairly clear to me and I don’t have questions for the authors.
In general options to improve my impression of this work would be any push towards realistic levels of complexity and/or demonstrations of overall segmentation behaviour that aligns with our knowledge about human vision.
- For example, making this work on natural stimuli, or at least something with a background or more objects and parts than fit into different phases.
- Or inclusion of a spatial propagation of the recruitment or an attention like mechanism that can choose which object to extract.
- Or just comparisons to any other algorithms performing the same tasks.
- Or comparisons to the alternative ideas how object segmentation might be represented in visual cortex like attentional facilitation spreading, an explicit representation of local similarity, or the hierarchy assignment.
- Etc.

These would be substantial revisions of the manuscript that I do not expect to see in the revision period though and thus expect to keep my rating.

---

> ### Author Response · Authors · 2023-11-21
> **Response to reviewer iE1G (part1/8)**
>
> >We thank the reviewer for spending the valuable time to read the paper. However, we find that there are significant factual errors in the review which may mislead other reviewers or readers. Besides, some comments do not accurately reflect the content and contribution of the paper. And the argument is not very informative due to vagueness. As a result, we would like to first point out those **factual errors** or **inaccurate understandings**. Then we provide a point-to-point response to the reviewer’s concerns.
> ---
> Factual errors:
> >- (1)	On the “Summary” of the paper: “In this manuscript the authors present their COMPOSER scheme how object hierarchies might be represented in biological neural networks by synchrony of neural responses.”
> >>- No, how object hierarchies are represented in real biological neural networks is not the main focus of the paper. Instead, we take inspirations from theories and evidences in neuroscience (See Appendix A.9) to explore how part-whole hierarchy could be represented in neural networks so as to provide a solution to the challenging computational problem.
> >- (2) “The synchrony is created by using a denoising autoencoder on the network representation whose output drives neurons one period of the oscillation later.”
> >>- Not exactly, the coupling delay do not equal to the period of oscillation. For example, the delay in the part / whole level shares the same value for simplicity in this work, but the oscillation period is not equal for the two levels.
> >- (3)	On the “Weakness“ of the paper: “From the perspective of artificial systems synchrony based approaches are not really of interest from the start as similarity for inferring objects and parts can be implemented much more efficiently by simply computing the information that is encoded in the phase explicitly.”
> >>- Not True. Actually, synchrony-based methods are of interests even in artificial neural networks [1,2,3], especially for encoding relationship. As stressed by Schmidhuber in his review [4]: “… it is clear that temporal synchronization plays an important role in neural information processing, and perhaps one that is still unaddressed in current artificial neural networks…”. There are rich theoretical frameworks of synchrony, which can help to conquer computational problems in ANNs. The challenge really lies on how to integrate the synchrony in ANNs with minimal biological details that really matters. The minimal implementation of brain-inspired model in this paper contributes to reducing the gap. While both synchrony and phase can encode similarity, time dimension (synchrony) is different from spatial dimension (phase).
> >>>- For example, time is natually distinguished from space (e.g. for simulation); but space (for phase) is entangled with space (for other information) and need to be specified artificially in most cases. This reduces the flexibility of representation (eg. how should we divide the dimension? and each way of division induces a different coding scheme, so that generalizability would be limited).
> >>
> >>- Besides, the efficiency is not informative without a clear constraint. When there are unlimited spatial resources, encoding everything in space is efficient, but when there are constraints / cost on spatial resources, time / recurrence become efficient. In total, the cost to infer by phase should not really lower than that by synchrony. Therefore, exploiting both space and time is desirable even for ANNs.
>
> >- (4)	“The scheme proposed here is only evaluated on extremely simple tasks, without any comparisons to other techniques, which is not convincing that the authors’ technique works particularly well, even for these simple tasks.”
> >>- No, we have compared our model to related models in the main text (Figure 9 in the main text) and shows that the COMPOSER outperforms the benchmark model. More details on benchmarking can be found in Appendix A.5.
>
>
> >[1] Hao Zheng et al. Dance of snn and ann: Solving binding problem by combining spike timing and reconstructive attention. Neurips 2022.
> >
> >[2] Hao Zheng et al. GUST: Combinatorial Generalization by Unsupervised Grouping with Neuronal Coherence. Neurips 2023.
> >
> >[3] Reichert, David P. and Thomas Serre. “Neuronal Synchrony in Complex-Valued Deep Networks.” CoRR abs/1312.6115 (2013): n. pag.
> >
> >[4] Greff, Klaus et al. “On the Binding Problem in Artificial Neural Networks.” ArXiv abs/2012.05208 (2020): n. pag.

---

> ### Author Response · Authors · 2023-11-21
> **Response to reviewer iE1G (part 2/8)**
>
> Factual errors: (**continued**)
>
> >- (5)	“As the authors admit it additionally has a bad scaling behavior because a different phase is required for each part of every object in the scene. For natural scenes this approach is not expected to work.”
> >>- It is not precise what ‘scaling’ really means by the reviewer because we did not use this term in our paper. As far as we understand, the ‘scaling’ stands for the capacity of the **number of objects** the network can represent.
> >>>- However, such constraint is shared by both human and our model. In [5], it is argued that the number of objects that can be processed together by human is around 7 (magic number), far away from unlimited capacity as the reviewer might have suggested.
> >>>- Further, representing neural syntax with temporal structure is one potential cause (“a different phase is required for each part of every object in the scene” as reviewer suggested). As argued by Buzaki [6]:
> >>>>- “the temporal compression mechanism can limit the ‘attention span’ and the ‘register capacity’ of the memory ‘buffer’ of the gamma-nested theta-cycle to seven to nine items”.
> >>>
> >>> Therefore, the constraints in representing capability does not necessarily cause the failure in representing more complex scenes (at least human does not fail). Actually, our model has shown representing capacity up to 12 objects, therefore the scaling should not be regarded as ‘bad’.
> >>>- An intuitive answer to the reviewer’s concern is that brain use different strategies in different cases and different stages. When the object number is beyond the working memory capacity of the brain, the brain “attends” to a local region to restrict the object number to be within its capacity. By sequentially switch to different local regions, the brain can form a perception of the entire scene. **Our model focuses on the cases** where object number is within the capacity or the attention has already been focused on a local region of the scene (but there may still multiple objects of part-whole relationship in the local region).
> >>- In short, asserting that limited working memory capacity is expected to cause failure is an overstatement, since these limitations are shared by both our model and the brain, even taking temporal constraints as the common cause.
>
> >- (6)	“And we have working segmentation algorithms that can do segmentation at any level of the object hierarchy. These approaches appear clearly better suited for computer vision than the one proposed here.”
> >>- First of all, the argument is very vague since the reviewer did not provide any reference to support the over-strong argument. And it is not clear whether ‘segment any level’ the reviewer referred to is actually equal to representing part-whole relationship that this work really cares about.
> >>- It is likely that the reviewer misunderstands the core motivation and challenge of the paper. In short, to represent part-whole hierarchy, not only ‘content’, but also the ‘**relationship**’ among contents should be represented by the neuronal activity of the neural networks. Therefore, the representation at a higher level (relation instead of content) should be a much more challenging problem.
> >>- Even if a segmentation model can segment object at many levels like [7], these segmented pieces do not have a relation among them, which is essential for representing part-whole hierarchy.
> >>- As far as we know, representing part-whole hierarchy is an unsolved problem in CV, regarded as a cornerstone in the CV field once realized as argued by Hinton [8].
> >>- However, the reviewer’s argument indicates that it is a well-solved problem with plenty of available methods and algorithms. Such inaccurate understanding of background knowledge is very misleading in evaluating the motivation and contribution of the paper.
>
> >[5] Miller, George A.. “The magical number seven plus or minus two: some limits on our capacity for processing information.” Psychological review 63 2 (1956): 81-97 .
> >
> >[6] Buzsáki, György and Brendon O. Watson. “Brain rhythms and neural syntax: implications for efficient coding of cognitive content and neuropsychiatric disease.” Dialogues in Clinical Neuroscience 14 (2012): 345 - 367.
> >
> >[7] Kirillov, Alexander et al. “Segment Anything.” ArXiv abs/2304.02643 (2023): n. pag.
> >
> >[8] Hinton, Geoffrey E.. “How to Represent Part-Whole Hierarchies in a Neural Network.” Neural Computation 35 (2021): 413-452.

---

> ### Author Response · Authors · 2023-11-21
> **Response to reviewer iE1G (part3/8)**
>
> >- (7)	“I also think the less relevant aspect of a model of biological vision is not really fulfilled by this model, because of the scaling issues and because this model implies parallel processing of objects everywhere. Clearly humans can deal with natural levels of complexity and it is well documented that object parsing takes time and proceeds sequentially, i.e. humans cannot do segmentation for all objects in a scene at once as this model would suggest.”
> >>- The bio-correlates of the model is summarized in full detail in Appendix A.9, from coding principle, to architecture, to dynamics, to neuron type and related theories. Actually, the proposed model is highly-inspired from various facts in neocortex, but we make simplifications to balance the bio-plausibility with implementational succinctness, without loss of generality, which is essential to bridge neuroscience with machine learning field.
> >>
> >>“because of the scaling issues”.
> >>- As argued in (5), scaling issue is a common issue of both our model and the brain, so that such limitation should not lead to reviewer’s conclusion.
> >>
> >>“because this model implies parallel processing of objects everywhere. Clearly humans can deal with natural levels of complexity and it is well documented that object parsing takes time and proceeds sequentially, i.e. humans cannot do segmentation for all objects in a scene at once as this model would suggest.”
> >>- On the one side, the parallel processing is a basic way of processing in visual cortex. Specifically, both parallel and sequential processing are important for scene understanding, either in **different cases**[9] or in **different stages**[10]. Therefore, it is an overstatement to say that human deal with object parsing only sequentially. It would be more accurate to say that the brain uses both strategies and the current model mainly accounts for the parallel case. For example, if the complexity of natural scene exceeds the working memory capacity (e.g. around 7), alternative strategies like sequential spatial attention (or sequential saccade) can help to resolve the problem. But even in each attended area, it is possible that there are still multiple objects of part-whole relationships that may need parallel processing and representation. As stated in [11]:
> >>
> >>> “…Memory is a key for feature binding in obtaining coherent object representations. An influential conception of visual working memory is each of a small number of discrete memory “slots” stores an integrated representation of a single visual object and includes all its component features as well. If a scene contains more objects than slots there have, then visual attention will control which objects to gain access to memory…”
> >>
> >>- On the other side, the model does not imply “parallel processing of objects everywhere” or “humans do segmentation for all objects in a scene at once” as the reviewer understood. We are aware of the common sense that brain do not use a single strategy for all cases. But we choose to focus on understanding the parallel aspect. For example, the input image for the model is not necessarily the whole scene as the reviewer thought, **but an attended local patch of the scene**. The model provides a solution of how each attended patch are understood. Actually, such view is shared in almost all temporal binding models and theories[cite]: temporal binding theory do not exclude the possibility of sequential processing of the scene, though most of related models process images in a parallel manner[11].
>
> >[9] McElree, Brian and Marisa Carrasco. “The temporal dynamics of visual search: evidence for parallel processing in feature and conjunction searches.” Journal of experimental psychology. Human perception and performance 25 6 (1999): 1517-39 .
> >
> >[10] Li, Kang et al. “Distinguishing between parallel and serial processing in visual attention from neurobiological data.” Royal Society Open Science 7 (2018): n. pag.
> >
> >[11] Ding, S., Meng, L., Han, Y. et al. A Review on Feature Binding Theory and Its Functions Observed in Perceptual Process. Cogn Comput 9, 194–206 (2017).

---

> ### Author Response · Authors · 2023-11-21
> **Response to reviewer iE1G (part 4/8)**
>
> Factual errors: (continued)
> >- (7) (continued)
> >>-  Thirdly, the COMPOSER does have sequential processing (not “parallel processing of objects everywhere” as the reviewer understood). For example, Figure 6 (e) in the main text and **Figure 21 (e) to Figure 28 (e) in Appendix** shows that there are emerged sequential attention along the parsing process.
> >>- We provided brief discussion in section 5.1 in the main text (“Emergence of the DAE attention map”) and provided more detailed explanation in A.10.6 in Appendix. The **emerged sequential DAE attention map** can be regarded as the “sequential processing” or “spatial propagation of the recruitment or an attention like mechanism” the author suggested, in the setting of this paper. In other words, in a complex system, sequential and parallel is often not absolute but **a matter of the timescale** we choose to view the system. For example, if we take each timestep as 1ms in the brain, then viewing the system from a timescale of 100 ms provides a picture that the brain uses spike timing code (nested synchrony) to represent the part-whole hierarchy in parallel. However, if we view the system from a timescale of 10 ms, we would find that the neuronal assembly emerges sequentially and there is spatial propagation of the attention-like mechanism (DAE feedback, etc). However, psychological evidence implies that the timescale of perceptual awareness is around 200 ms, therefore we take the “sequence” of neuronal assemblies as a “complete representation” of the visual scene. In fact, our model **bridges** the sequential processing viewpoint and parallel processing viewpoint consistently.
> >>
> >>- Lastly, the object parsing in our model also takes time, consistent with that in human. In **Figure 23 and Figure 24 in the Appendix**, it can be seen that there is even hierarchical processing during the parsing. More specifically, the whole level is processed before the part-level, so that the COMPOSER solve the part-whole problem in a divide-and-conquer manner. The reviewer’s arguments, “humans cannot do segmentation for all objects in a scene at once as this model would suggest”, is misleading because it implies that the model treat everything simultaneously, which is the strategy used by most artificial neural networks. Instead, our model attends to each object sequentially, when viewed from short timescale, and in total integrates all object information in its “working memory” as nested synchrony when viewed from longer timescale. The parsing takes time, and even processed hierarchically.
> >>
> >- (8)	“Also, the levels of synchrony observed in the networks here go far beyond anything I have ever seen in a visual cortex.”
> >>
> >> Actually, representing nested relation with nested neuronal activity (nested neuronal assemblies along with nested oscillation) is a long-standing viewpoint in neuroscience, so called “neural syntax” [12]. For example, as Buzaki said in this nice review paper:
> >>
> >>- “In general, syntax (grammar) is a set of principles that govern the transformation and temporal progression of discrete elements (e.g., letters or musical notes) into ordered and hierarchical relations (e.g., words, phrases, sentences or chords, chord progression, and keys) that allow for a congruous interpretation of the meaning of language or music by the brain……The second hypothesis of this review is that temporal sequencing of discrete assemblies by neural syntax can generate neural words and sentences. ……I chose the term ‘‘neural word’’ to emphasize that words consist of multiples of the fundamental assemblies. Gamma oscillation episodes, containing a string of assemblies, are typically short lasting and often grouped by slower oscillations. Such a relatively short sequence of cell assemblies may be regarded as a neural word……Linking strings of fundamental assemblies requires readers with longer time integration abilities……”
>
>
> [12] Buzsáki, György. “Neural Syntax: Cell Assemblies, Synapsembles, and Readers.” Neuron 68 (2010): 362-385.
> [13] Uhlhaas, Peter J. et al. “Neural Synchrony in Cortical Networks: History, Concept and Current Status.” Frontiers in Integrative Neuroscience 3 (2009): n. pag.

---

> ### Author Response · Authors · 2023-11-21
> **Response to reviewer iE1G (part 5/8)**
>
> Factual errors: (continued)
> >- (8) (continued)
> >
> >> In a review of the cortical network in 2009 [13]:
> >>
> >>- “……Another important and unresolved question is related to the coexistence of oscillations in different frequency bands. We are only at the beginning of understanding their mutual interactions and spatial organization. Neuronal synchrony is found on different temporal and spatial scales. The spatial scale of synchrony can range from local synchronization of small numbers of neurons within a single cortical column to the synchronization of large populations that are distributed across different cortical regions and even across both hemispheres. The temporal scales of neuronal synchrony range from very precise sub-millisecond spike-spike synchronization to the synchronization of slow oscillatory activity as low as < 0.1 Hz. Remarkably, synchronization is not confined to oscillations of the same frequency band but occurs across different frequencies as n:m synchrony. This allows for the concatenation of rhythms and for the establishment of partial correlations. An attractive hypothesis is that this could serve as a mechanism to encode nested relations – an indispensable function for the neuronal representation of composite objects and movements……”
> >>
> >>- The COMPOSER follows closely to these spirits (Appendix A.9) and the dynamical picture / network architecture / experimental results provide new insights to the theory, which is contributes to unlock the neural syntax. Note that **part-whole hierarchy is the syntax for vision**. Though the idea of representing neural syntax by nested rhythm idea is still hypothetical, which needs constant experimental verifications, our model provides a working model to understanding the process behind the theory.
> >>
> >>-  Moreover, nested spatial temporal neuronal activity is at the core of the spatial temporal theory of consciousness (TTC, Appendix A.9) [14,15]. More specifically, the **nestedness accounts for the level of consciousness**. Related evidence in visual cortex[16] is reviewed in the theory [14]. In a point of view of this paper, the level of consciousness is related to the quality of the representation of the neural syntax (nested synchrony).
> >>
> >> - Also, in Appendix A.9, we summarized the recent neuroscience evidences of the timescale hierarchy along the cortical hierarchy of visual cortex [17].
> >>
> >>- Therefore, the nested synchrony emerged in this paper is well-documented in theories and supported by experiments in neuroscience. Though debates also exist, the proposed work could serve as potential computational models to understand neural syntax in cortical circuit.
> >
> >[13] Uhlhaas, Peter J. et al. “Neural Synchrony in Cortical Networks: History, Concept and Current Status.” Frontiers in Integrative Neuroscience 3 (2009): n. pag.
> >
> >[14] Northoff, Georg and Zirui Huang. “How do the brain’s time and space mediate consciousness and its different dimensions? Temporo-spatial theory of consciousness (TTC).” Neuroscience & Biobehavioral Reviews 80 (2017): 630-645.
> >
> >[15] Huang, Zirui. “Temporospatial Nestedness in Consciousness: An Updated Perspective on the Temporospatial Theory of Consciousness.” Entropy 25 (2023): n. pag.
> >
> >[16] He, Biyu J. et al. “The Temporal Structures and Functional Significance of Scale-free Brain Activity.” Neuron 66 (2010): 353-369.
> >
> >[17] Ana Maria Manea, Anna Zilverstand, Kamil Ugurbil, Sarah R. Heilbronner, and Jan Zimmermann. Intrinsic timescales as an organizational principle of neural processing across the whole rhesus macaque brain. eLife, 11, 2021

---

> ### Author Response · Authors · 2023-11-21
> **Response to reviewer iE1G (part 6/8)**
>
> Response to the questions or concerns of the reviewer.
>
> Q1. For example, making this work on natural stimuli, or at least something with a background or more objects and parts than fit into different phases.
>
> >“making this work on natural stimuli” and “push towards realistic levels of complexity” are  over-demanding requirements for this model. Actually, self-supervised or unsupervised object-centric representation, even without hierarchical relation, are mostly evaluated on synthetic datasets instead of natural images [18,19,20,21,22]. Since these works are more theoretical than practical, it is the architecture, interpretable representation and insights behind that are more important, which are also illustrated in our paper. Applying to natural image is a common limitation of all related works in this field, not just our work. And these “pushes” are of less interests in the field if there are no theoretical insights or the representation is not interpretable. Since our work further taking hierarchical relations into account, it is over-demanding to require the model be tested on natural images. To convince the reviewer of the potential “pushes”, it is helpful to review a line of work in object-centric representation. Starting from toy models that is pre-train on single objects and tested on toy datasets (binary, without background)[18], a line of following work incrementally increase the performance of the model[19,20,21,22], so as to be applied to more and more complex dataset (eg. CLEVER)[22]. The whole process takes about 6 years (from 2016 to 2022). The important thing for this process is that, while the model is simple, the idea behind is general and elegant, so that it could be gradually improved with increasing engineering works. In this paper, we also try to provide a minimal model that is only based on elegant set of principles: denoising, timescale hierarchy, etc, which could be generalized to future works. More detailed discussion of limitation and future direction can be found in Appendix A.1.
>
> > “or at least something with a background”.
>
> >- We provide a demo in the **Appendix A.11 and Figure 29** of updated paper. The demo shows that background does not necessarily destroys the representation, since the model can selectively attend to the objects of interests and ignore others.
>
> >“or more objects and parts than fit into different phases.”
>
> >- Actually, objects and parts are not ‘fit’ into phases. Instead, the phases (ups and downs) emerges during the evolution of the system, not predefined. It is not clear how large the number should be to “not fit”. On the other hand, in Ts dataset, the object number is very large (6-whole objects and 12-part objects). As shown in Figure 25 (c) in the Appendix, when the object number is large, the spikes flexibly exploit time dimension to express correlation information. Specifically, different assemblies can occur with different frequency (some Ts occur more than others), so as to relieve the representational burden of the time dimension. In Figure 25(c) right, blue assemblies (top) occur six times while red assemblies (bottom) occur three times. Such flexible use of time is one interesting phenomenon in our model.
>
> Q2. Or inclusion of a spatial propagation of the recruitment or an attention like mechanism that can choose which object to extract.
>
> >Since the “spatial propagation” is a very big concept and it is not clear what reviewer really refers to without providing the reference, we provide the response based on our understanding that could potentially resolve the reviewer’s concern.
> >- As explained above, DAE feedback is a form of spatial attention in our model and can select object of interests from noisy background (Figure 29 in updated Appendix). The sequence of attention patterns can be seen in Figure 6(e) in the main text and Figure 21(e) to Figure 28(e) in Appendix. Attention in the brain have different levels (eg. from external (saccade), to internal).
> >- In this paper, the attention is regarded as the endogenous top-down feedback from higher-level layers of the cortex (eg. layer5/6). The attention emerges through iterative top-down / bottom-up dynamics given the input. The emergence of attention-like mechanism and the emergence of neuronal coherence are the two highly related sides of the model, consistent with neuroscience evidence [23].
>
>
> >[18] Greff, Klaus et al. “Binding via Reconstruction Clustering.” ICLR (2015).
> >
> >[19] Greff, Klaus et al. “Tagger: Deep Unsupervised Perceptual Grouping.” Neurips (2016).
> >
> >[20] Greff, Klaus et al. “Neural Expectation Maximization.” Neurips (2017).
> >
> >[21] Greff, Klaus et al. “Multi-Object Representation Learning with Iterative Variational Inference.” ICML (2019)
> >
> >[22] Locatello et al. Object-Centric Learning with Slot Attention. Neurips 2020.
> >
> > [23] Andreas Karl Engel et al. Dynamic predictions: Oscillations and synchrony in top–down processing. Nature Reviews Neuroscience, 2001.

---

> ### Author Response · Authors · 2023-11-21
> **Response to reviewer iE1G (part 7/8)**
>
> Q3. “Or just comparisons to any other algorithms performing the same tasks.”
> >- We do have provided comparison in Figure 9 in the main text.
>
> Q4. “Or comparisons to the alternative ideas how object segmentation might be represented in visual cortex like attentional facilitation spreading, an explicit representation of local similarity, or the hierarchy assignment.”
>
> >- As discussed in Appendix A.9 (“Temporal binding theory and feature integration theory” section). Two main steam theories for segmentation in the brain is attention-based (feature integration theory) and synchrony-based (temporal binding theory)[11]. It is still open question which idea captures the nature of the brain computation. In this model, we naturally integrated both sides into the model.
> >>- On the one hand, at representational level, the parsing tree is represented as nested synchrony of spiking neurons;
> >>- On the other hand, at computational level, sequential top-down attention (at each level / across level) emerges to modulate the parsing process.
> >>- Actually, the COMPOSER provides a good starting point to unified the both sides within a closed system and a consistent framework.
> >
> >- As far as I understand, attentional facilitation spreading [24,25] holds the spirit that there are multiple levels of attentions, so that the lower-level attention can spread even within a (high-level) attended object (eg. hand movement vs endogenous attention). This fact matches very well with our framework. As shown in Figure 6(e) in the main text and Figure 21(e) to Figure 28(e) in Appendix, there are emergent selective attentions of different levels (part / whole).
> >- When the whole-level attention attends to an object, **part-level attention sequentially spread over the parts of the attended objects**. Therefore, our model is consistent with the “attentional facilitation spreading” and further shows how the attention emerges and potentially related to nested synchrony.
>
> >[24] Wu W, Li Z, Miura T, Hatori Y, Tseng CH, Kuriki I, Sato Y, Shioiri S. Different Mechanisms for Visual Attention at the Hand-movement Goal and Endogenous Visual Attention. J Cogn Neurosci. 2023 Aug 1;35(8):1246-1261. doi: 10.1162/jocn_a_02005. PMID: 37172135
> >
> >[25] Richard AM, Lee H, Vecera SP. Attentional spreading in object-based attention. J Exp Psychol Hum Percept Perform. 2008 Aug;34(4):842-53. doi: 10.1037/0096-1523.34.4.842. PMID: 18665730.

---

> ### Author Response · Authors · 2023-11-21
> **Response to reviewer iE1G (part8/8)**
>
> Q5. “I do not think this is a convincing model of processing in visual cortex or a good starting point for machine learning and computer vision models.”
>
> >While the claim of the reviewer is harsh, we understand the underlining concerns of the reviewer. “Can this paper really contribute to either field?” On the one hand, we hope the response above partially resolved the reviewer’s concern. On the other hand, we would like to restate the contributions of the paper to either side and to both sides.
> >
> >- From the neuroscience side,
> >>- the model is inspired by a list of neuroscience theories and findings and is consistent with main stream views of the brain (eg. neural syntax by rhythmic assemblies). We provided detailed discussion in A.9 in Appendix. To bridge the idea in neuroscience with ANNs, we have to make several simplifications and preserves the main insights. Such simplification should not be regarded as “not convincing”, because if necessary, the simplification process could be “reversed” to add increasing details to really model the cortical circuits. As far as we know, there are no intrinsic limitations to add most biological details into the model. Besides, we compare the model to visual cortex to visualize how the model could be bio-related. However, it is **not necessarily to be constraint for certain regions** or circuits of the brain.
> >>- For example, Hopfield network could be applied to understand diverse brain areas, not just CA3, depending on how we treat the model.
> >>- Similarly, even though the minimal model in this paper may not capture the future observation in visual cortex, which is very common in neuroscience, the model could still be modified and used to understand other parts of the brain.
> >>- As long as neural syntax, nested oscillation, nested assembly code, diverse attention-mechanism, delay-coupling are important features of the brain, our model could contribute to understand wherever these features play a role. Since hierarchical rhythmic activity is still an unknown phenomenon in the brain, much experimental and modeling works are needed to understand them. Our model has the advantage that,  compared to traditional neuroscience model, bridging with ANNs enlarges the capacity of the model (Figure 2 in the main text) to solve non-trivial parsing problems (larger number of stimuli).
> >
> >- From machine learning side,
> >>- as argued by Hinton in [8], representing part-whole hierarchy in neural network could be a cornerstone for CV field, if realized. This work is the first work to explicitly represent part-whole hierarchy with quantitative evaluation. Such a working model itself is a contribute to the field, regardless how it is implemented. While the biological details may not be necessary for pure AI researchers, basic representation insight (encoding relation by correlation), computational principles (denoising and parameter hierarchy), mechanism (dynamical picture), and evaluation pipeline (generalizable to evaluate ANNs, A.4.4 and A.5.4) can provide useful inspirations to build up future pure ANN models. Especially, quantitive evaluation is a challenge in part-whole field and we find that most related work was evaluated on less relevant metrics like classification accuracy. Lack of proper metric makes it questionable whether these models can really parse. As a result, the evaluation pipeline we developed (metrics for each node and for the parsing tree) can contribute to the field. We hope the reviewer captures the value / challenge of the problem of part-whole hierarchy itself and ground our framework as a working solution to this unsolved problem.
> >
> >- In sum, our work can contribute to both sides. **Lastly, the author ignores a third way of evaluating this paper**, beyond viewing it from either neuroscience or machine learning in isolation: to bridge the both sides.
> >>- To bridge the both sides, the biological details need to be simplified. The representation framework, network architecture, training pipeline and even evaluation metric, need to integrate considerations from both sides, which is a challenge. For example, the attention-mechanism is realized as the DAE feedback and emerged in a closed system. For example, how nested synchrony code could emerge and be exploited in ANNs (augmented with minimal modifications)? For example, how synchrony and parsing can be evaluated by integrating Silhouette score in machine learning and Victor-Purpura metric in neuroscience (See A.4 in Appendix). From a point of view of bridging neuroscience and machine learning, which is an important direction, our work provides systematic contributions at all levels.

---

> ### Comment · Reviewer_iE1G · 2023-11-23
> **Read the comments**
>
> I read the (rather long) author replies. They did not improve my opinion of the paper.

---

### Official Review · Reviewer_JZAs · 2023-11-01

**Soundness:** 2 fair
**Presentation:** 3 good
**Contribution:** 2 fair
**Rating:** 5
**Confidence:** 3

**Summary:**

This work presents a dynamical network-based solution to the problem of representing part-whole hierarchy in artificial neural networks. They propose a model, Composer, which consists of two-layers of neurons that mimic lower and higher visual areas, which form assemblies to represent parts and whole, respectively. The model uses a denoising autoencoder to encourage finding metastable attractors in the spike coding population, which consists of (largely) rate-based neurons that emit spikes with refractory periods. Furthermore, the authors create a set of toy tasks and a metric that evaluates the performance of the proposed model, and compares it to a SOTA model for visual part-whole parsing.

**Strengths:**

The introduction of the paper (and Fig.1) explains the part-whole hierarchy representation problem quite clearly, with behavioral and neuroscientific evidence as motivation. The intuition behind the proposed model (e.g., Figure 2) is also nicely illustrated. The creation of several benchmark toy tasks as well as a metric is commendable, so is the comparison to a SOTA model, and ultimately, demonstrates that the proposed model works. Overall, it’s an interesting paper which tackles a very high-level problem with a novel brain-inspired model.

**Weaknesses:**

1. it’s unclear how the model actually works, even though the explanation of the architecture is clear. I’m not sure if I understood correctly the important mechanisms, but it feels a bit magical that it “just worked”, and that assemblies naturally emerged. Was there parameter learning or network training, either beforehand or during a single stimulus? Or is it really the case that a stimulus is simply presented and both whole and part assemblies simply arise over time? If the latter case, how sensitive are the results to specific values of the parameters. While the ablation study is nice, setting the various values to zero is somewhat dramatic and uninformative, and I’m rather wondering how precise the parameter values must be, e.g., tau_r, for the network to work. It’s also a bit unclear to me what exactly the DAE is doing, though Figure 10a suggests that they are trained a priori?

2. the proposed model, given the stated motivation of the paper, was somewhat underwhelming for me. If I understood correctly, the “spike coding space” layers simply receive input, one neuron per pixel, and while they do emit spike, there is no recurrent interaction within the SCS (as a standard spiking NN would), nor is it really “spiking”, since the rates are the governing variables and multiplicatively gated across layers, but the spikes are simply emitted with a given—and precisely set—refractory time. If this is correct, it’s difficult to judge how robust the setup is (i.e., parameter sensitivity), as well as how generalizable this mechanism is. The connections to neural circuits are quite loose, and I think calling it a “bio-plausible” framework is an overstatement.

Taken together, my limited reading of the work is that it’s a very “handcrafted” model / toy solution to an important and general problem, but ultimately falls short of making a convincing contribution.

**Questions:**

- Some clarification or intuition on how the model works would be informative, i.e., are there just naturally emerging assemblies after letting it run for long enough? What does the DAE do and how is it trained (if it is)?
- Is there no parameter tuning? how does the “whole”-level population naturally fire with longer periods (e.g., Fig7b?)? Or is it very sensitive to specific parameter values (e.g., delay and refractory timescales), and if so, are the findings of the study generalizable to either learning something about the brain or improving practical ML algorithms?
- axis scaling between different tasks for Fig 8, esp panel d, is misleading
- What is the downstream decoder? Since the entire image is represented partially through time, how is the parsing score computed? Is it aggregated / smoothed over some time window? If so, how does one choose this window and is it realistic for a decoder in the brain to perform this?
- The study does not really acknowledge any limitations, in particular how much the proposed model deviates from its stated goal of “bio-realistic” implementation. The most obvious example is that the spiking is quite artificial (as explained above).
- Similarly, the lack of recurrent activity between neurons within a SCS layer seems very implausible, compared to cortical assemblies in the real brain. In this work, the assemblies arise due to the DAE forcing them into metastable attractors (I think?), whereas cortical assemblies are typically formed via connectivity, especially to and from inhibitory neurons.
- The introduction raises several points that existing models fail at, e.g., “the parse tree could switch among multiple reasonable forms even given a single scene”, i.e., “correct” parsing is context-dependent. While this is true, the proposed model also does not deal with this (or does it?)
- The proposed relationships to the canonical cortical microcircuit, e.g., Figure 2g, doesn’t really add much value, in my opinion
- Does the model even need the timescale hierarchy? At it’s core, each layer simply has assemblies that represent some specific part of the image, and it seems like persistent firing with the different assemblies that temporally coincide is sufficient for solving this task
- in-line citations are not bracketed, which is very distracting
- figure text is way too small, e.g., Fig 5c & 8a legend box is illegible; similarly, very hard to tell different colors of spikes in 7c
- Some citation on the formation of, and computing via neuronal assemblies may be appropriate (e.g., L. Mazzocato, J Gjorgjieva, etc.)

---

> ### Author Response · Authors · 2023-11-21
> **Response to the reviewer JZAs (Part 1/7)**
>
> >We thank the reviewer for spending the valuable time to thoroughly read the paper. The questions and advice are very important to improve our work. Here, we would like to provide detailed responses to the reviewer’s concerns.
>
> Q1. “it’s unclear how the model actually works” and “it feels a bit magical that it just worked”.
>
> >We understand the reviewer’s concern.
> >
> >- On the one hand, the emergent phenomenon usually looks like a ‘magic’ (eg. phase change of magnetic field in physics, or the clustering process during EM algorithm). Besides, there is no explicit supervision to guide the parsing. These are exactly why the model and results are non-trivial and interesting.
> >- On the other hand, to understand emergent behavior, it is helpful to have a dynamical system picture, which we tried to explain in Figure2 in the main text. In short, intuitively, **the emergent synchrony states are local minimums of the energy landscape** of the dynamical system, which is shaped by DAEs and biological constraints (eg. time-scale parameters).
> >>- For example, DAEs of different levels are pre-trained to know” what objects look like at that level” (See Figure 18 in the Appendix), which are essential prior knowledge for parsing and can be taken as innate structure in the brain. We ask how could the model parse a scene based on those minimal prior knowledges of objects.
> >>- For another example, the timescale hierarchy (whole level slower than part level) biases the network to form hierarchical spatial temporal patterns at different level (the nested synchrony). Figure 2 provides a more systematic explanation of how the entire system is built-up step by step.
> >
> >To understand the property of the single column at each level, [1] provided nice theoretical proofs of how synchrony emerges at each level. We hope the explanation helps the reviewer to capture the basic insight of how the model works.
>
> Q2. Was there parameter learning or network training, either beforehand or during a single stimulus? Or is it really the case that a stimulus is simply presented and both whole and part assemblies simply arise over time?
>
> >The DAEs are pre-trained to reconstruct single objects (to know what single objects look like, detailed in Appendix A.7.3; A.7.4), which is a convention in related works [1], and also bio-plausible (A.9 ‘Preconfigured Brain section’). However, how to represent the ‘relationship’ is not trained, but a self-organized behavior of the model. During parsing, a stimulus is simply presented and both part and whole assemblies arise over time (no training is needed anymore).
>
> Q3. “How precise the parameter values must be” and “parameter sensitivity test”.
>
> >We provide the parameter sensitivity test on SHOPs dataset in Appendix A.11 and **Figure 30** (updated version). It is shown that the performance of the model is **robust to parameter perturbations** within a wide range. Actually, we did not fine-tune the parameters and they are set based on requirement shown in eq (26) in the Appendix.
>
> Q4. what exactly the DAE is doing
>
> >We are sorry that the unclearness of our writing masked such essential mechanism of our model. The role of DAE is visualized in the Figure6(e) in the main text and analyzed in Figure 10 (a) in the main text. Its bio-correlates are explained in Appendix A.9 (“Top-down attention as autoencoder”). The overall picture of how DAE works as feedback is shown in Figure 2(a)(c) in the main text.
> >
> >Briefly, the output of DAE provides **attentional feedback to modulate the spiking neurons**. Therefore, as shown in Figure6(e), starting from random guessing (left panel), selective attention (of different timescales) gradually **emerges** in the DAE feedback (right panel). The attention pattern from DAE (Figure6(e)) is related to the synchrony pattern of spiking neurons (Figure 6(c)). The attentional feedback and neuronal coherence promote each other during the convergence process (more synchrony, more selective attention feedback from DAE, more synchrony).
>
> >[1] Hao Zheng et al. Dance of snn and ann: Solving binding problem by combining spike timing and reconstructive attention. Neurips 2022.

---

> ### Author Response · Authors · 2023-11-21
> **Response to the reviewer JZAs (Part 2/7)**
>
> Q5. Concerns on its relation to neuroscience.
> >We appreciate the reviewer’s question on the limitation of our model in a viewpoint of bio-plausibility. We admit that the current model is a minimal implementation of the ideas so that the complexity can not match that in the real neural circuit. We provided detailed explanation of such limitation and how the limitation can be overcome in future works in Appendix A.9 (“Inner-layer recurrent connection” and “Plasticity” section).
> >
> >Such limitation exists for all neuroscience modeling: “what biological details should be integrated into the model and what should be ignored temporally”. It is desirable to consider the details as long as they really play certain computational roles. Therefore, in the current model, we consider those biological constraints that really matters to solve the problem (at current task complexity).
> >
> >>- Other details, for example, inner-layer recurrence can be considered if we need to further require the representation to be self-sustained even if the input is removed (working memory).
> >
> >There is no fundamental limitation to add those details. In other words, we focus on bottom-up / top-down recurrence as core mechanisms in current model and regard inner-layer recurrence as secondary. The bio-correlates of the model are summarized in Appendix A.9 in detail (up to 18 supports).
> >>- For example, the delay coupling and time-scale hierarchy are all essential property of the neocortex.
> >We believe the model is highly inspired by neural system, consistent with a list of basic biological evidence or hypothesis, instead of “loose” and “overstatement” as reviewer suggested.
> >
> >To be more specific, we provide **point-to-point** response to the reviewer’s arguments in the following section.
> >>- (1)	“the “spike coding space” layers simply receive input, one neuron per pixel”.
> >>>- Yes, and we have discussed such limitation and how the limitation can be overcome as future work in Appendix A.1 (Visible Layer section). On the other hand, it is a reasonable simplification based on topographical mapping in cortex (Appendix A.9 Topographical mapping section). Each pixel stands for one location in the topographical map and can be generalized into a ‘column’ or a vector if needed (similar to [2]).
> >>- (2)	“while they do emit spike, there is no recurrent interaction within the SCS (as a standard spiking NN would)”
> >>>- Firstly, not all neural circuit have recurrent interactions within the layer. For example, striatum does not contain any inner layer recurrent connection. Cerebellum is also modeled as feedforward network. More relevant fact is that different layers in cortical column have different recurrence level: while layer5/6 have high recurrence level, **layer 2/3 has relatively low recurrence level**. More interestingly, layer 4 has even lower recurrence and usually serve as a feedforward layer. This picture is the underlining motivation that we ignore the inner-layer recurrence in the SCS, since SCS is treated as layer 2/3 in a cortical column.
> >>>- Secondly, as discussed in A.9 in Appendix, different types of recurrences usually have **different functional roles**. The underlining picture is that it is the top-down / bottom-up recurrence that contributes to the prior-knowledge-based grouping. The inner-layer recurrence can serve other roles. For example, the ‘spatial’ connection (e.g. grid-like connection) among neurons could further help to group objects based on spatial rules like closeness, similarity, proximity and so on [3]. So that such inner-layer connection can be complementary for the grouping in future models. For another example, inner-layer recurrence helps to sustain the activity if the input is removed and to complete the pattern if the input has noise. However, these considerations are not the focus in the current paper, so that we treat these facts as **secondary details** and ignores these details to make the model more elegant and the paper more readable.
> >>>- Thirdly, if the reviewer still concerns about inner-layer connection, the entire column could be recognized as a single layer, so that the DAE parameterize the connection weight among spiking neurons within the layer.
>
>
> >[2] Hinton, Geoffrey E.. “How to Represent Part-Whole Hierarchies in a Neural Network.” Neural Computation 35 (2021): 413-452.
> >
> >[3] Johan Wagemans, James H. Elder, Michael Kubovy, Stephen E. Palmer, Mary A. Peterson, Manish Singh, and Rüdiger von der Heydt. A century of Gestalt psychology in visual perception: I. Perceptual grouping and figure-ground organization. psycnet.apa.org, 2012a.

---

> ### Author Response · Authors · 2023-11-21
> **Response to the reviewer JZAs (Part 3/7)**
>
> Q5. Concerns on its relation to neuroscience (**continued**).
> >>- (3)	“nor is it really “spiking”, since the rates are the governing variables and multiplicatively gated across layers, but the spikes are simply emitted with a given—and precisely set—refractory time.”
> >>>- First, spiking neuron is not necessarily to be not rate-based. In our model, synchrony code of very precise temporal structure emerges and spike timing is exploited to represent information (grouping and parsing), which is the core feature of spiking neurons. To some extent, all spiking neurons, taken **stochasticity** into account, has a form of rate neuron. For example, the rate model can be considered as a threshold point neuron with noisy background input. In other words, spike firing based on membrane potential as firing rate is a very general case for spiking neurons. Instead, it is the timescale of integration window, the relative value of threshold (low threshold or high threshold), other constraints like refractory period, and the overall network dynamics that really determine whether the neuron works in ‘rate-mode’ or ‘spike-mode’. Therefore, we do not agree with the reviewer’s argument that the neurons are not spiking.
> >>>- Second, multiplicative interaction between driving input and modulatory input is a core non-linearity of dendritic computation of pyramidal neurons in cortex[4]. Actually, driving inputs arrive at proximal site of dendrites and modulatory input arrives at distal site of dendrites and both signals interact multiplicatively (Appendix A.9 section8). Therefore, multiplicative gating is not a reason to degrade the model.
> >>>- Third, the refractory dynamics in our model actually stands for any self-inhibition process that is common in neural systems, which could be realized either at single neuron level (our model) or at population level. For example, such self-inhibition can be realized by inhibitory neuronal populations. We choose single-neuron-level refractory dynamic as a minimal realization in the current model, but the basic idea behind is more generalizable.
>
> Q6. If this is correct, it’s difficult to judge how robust the setup is (i.e., **parameter sensitivity**), as well as how generalizable this mechanism is.
>
> >We have provided a parameter sensitivity test in updated appendix, please see **A.11 and Figure 30 in the updated Appendix**. The analysis shows that the model is relatively **robust to parameter perturbations** within a reasonable range. As argued in Q5, the simplifications in the current minimal model is reasonable approximations without loss of generality. On the one hand, the core mechanism only depends on very general principles, like denoising, attention, timescale hierarchy, delay-coupling, etc. Since it does not require domain-specific engineering (like bounding box for image segmentation [5]) or heavy supervised training [5], the model can potentially be generalized to other domains like auditory or language. On the other hand, there is no fundamental limitation that the SCS can only be realized in the way this paper did, and various further details (lateral recurrence, plastic neurons, etc) could be considered if necessary.
>
> >[4] Nelson Spruston. Pyramidal neurons: dendritic structure and synaptic integration. Nature Reviews Neuroscience, 9(3):206–221, 2008
> >
> >[5] Girshick, Ross B. et al. “Rich Feature Hierarchies for Accurate Object Detection and Semantic Segmentation.” 2014 IEEE Conference on Computer Vision and Pattern Recognition (2013): 580-587.

---

> ### Author Response · Authors · 2023-11-21
> **esponse to the reviewer JZAs (Part 4/7)**
>
> Q7. Taken together, my limited reading of the work is that it’s a very “handcrafted” model / toy solution to an important and general problem, but ultimately falls short of making a convincing contribution.
>
> >On the one hand, we agree with the reviewer that this paper focus on how to ‘construct’ a solution to the challenging problem. On the other hand, we disagree with the reviewer that the contribution is necessarily limited by the simplicity of the model and task.
> >
> >We would like to provide our reasons and restate the contribution of this paper as followings.
> >>- (1)	Construction of a solution is **a theoretical foundation** for further searching for such solution (eg. learning), especially on such a challenging problem. Actually, we are currently working on how to learn such a parsing system in a fully unsupervised and end-to-end manner (no pre-training is needed). In this case, biological constraints (e.g. time-scale hierarchy) acts as inductive bias to bias the learning towards hierarchical synchrony states.
> >>- (2)	While the parameter seems complicated, **the basic principle and idea behind is very elegant**. After all, these timescale parameters are unavoidable when considering spiking network of different levels (e.g. integration window, delay, refractory period, etc). We provided a summary of time scale parameters in Appendix A.6.2. The basic requirement of the parameters is general (eq 26 in Appendix). As explained in Q6, the framework is generalizable.
> >>- (3)	The contribution of the work is three-fold, as summarized in the main text.
> >>>- (1) framing of the problem at representation level;
> >>>- (2) developing a model that bridge AI and neuroscience to solve the problem;
> >>>- (3) developing metric and dataset to explicitly evaluate the part-whole representation.
> >>>
> >>>It is notable that each aspect is a challenging problem.
> >>>+ For example, dynamically forming a parsing tree in neural network is non-trivial, as the reviewer suggested it as ‘magical’. As argue by Hinton in his 2021 paper [2], such a model could be a **cornerstone for CV** field.
> >>>+ For another example, evaluating parsing representation of different quality is itself a challenging problem, which is lacking in this field.
> >>>+ It is also notable that representing part-whole hierarchy is an unsolved problem since it cares not only how to represent ‘content’, but how to represent **‘relations’** among content. Therefore, even a minimal solution on simple datasets can contribute to this field considering the insights it provides.
> >>>+ Lastly, our model has potential to **bridge AI and neuroscience** at flexible levels (minimal details as our work or more details as potential future works) to solve basic CV problems which can also benefit neuroscience modeling.

---

> ### Author Response · Authors · 2023-11-21
> **Response to the reviewer JZAs (Part 5/7)**
>
> *Response to questions*
>
> ---
>
> Q8. How DAE train and work.
>
> >See Q1 and Q2
>
> Q9. Is there no parameter tuning? how does the “whole”-level population naturally fire with longer periods (e.g., Fig7b?)? Or is it very sensitive to specific parameter values (e.g., delay and refractory timescales), and if so, are the findings of the study generalizable to either learning something about the brain or improving practical ML algorithms?
>
> >We did not optimize the timescale parameters by exhausted parameter tuning, the basic relationship among parameters is eq26 in Appendix. Specifically, neurons in the whole level has longer integration timewindow (timescale parameters) than neurons in part level to bias the model generate spatial-temporal patterns of different timescales. Such design principle is consistent with neuroscience theory [6]: **timescale difference of readout mechanism contributes to the neural syntax** (represent hierarchical information by hierarchical spatial temporal neuronal activity).
> >1. **_The sensitivity test can be found in Q3._**
> >>- The performance of the model is robust to the perturbation of timescale parameters within a wide range. The timescale parameter only 'bias' the parsing, intead of 'determining' the parsing.
> >
> >1.	learning something about the brain:
> >>- The timescale hierarchy is consistent with recent experimental findings of the brain (Appendix A.9 “Time scale hierarchy” section). If we take each timestep in simulation as 1 ms in the brain, the refractory timescale, coupling delay, integration window and oscillation frequency matches that in the brain (Appendix A.6.2, last paragraph).
> >>- Besides, the findings of the study indicate several insights about visual cortex that could be tested in the future.  For example, visual concepts are organized as synchronized assemblies at different scales and the coordination of the assemblies corresponds to the ‘awareness’ of a visual scene. For another example, the top-down recurrent dynamics and inner-layer dynamics serve different computational goals and top-down attention mainly contributes to object-based grouping.
> >>- Lastly, current model provides a working model for how neural syntax can be represented by neural rhythm [6] and how nestedness of temporal-spatial patterns in the brain can account for the level perceptual awareness [7]
> >
> >1.	improving practical ML algorithms
> >>- The initial motivation of representing part-whole hierarchy in neural network, as argued in [2], is theoretical rather than practical (to build artificial vision system like human). And our work follows this motivation. While current version of the implementation is too simple to directly improve practical ML algorithms (compared with supervised method on classification), progress in this field can help us better imagine what the representation should be like for a **human-like visual system**. Such representation is desirable because it captures both content and relation in a visual scene, which is believed to be the case for human [2].
> >>
> >>- Besides, if what the reviewer means by ‘practical’ refers to apply the current model to classify real images, we need to point out that the literature of self-supervised object-centric representation (even without hierarchical relationship) is mostly limited on synthetic (toy) datasets. It is because for this line of works, interpretability and theoretical insights are of more importance than ‘usefulness’ or ‘performance’. Therefore, ‘practical’ in real-images is an over-demanding requirement.
> >>
> >>- Lastly, this work does not aim to ‘improve’ at the very beginning, but to provide a ‘novel’ roadmap to unsolved problems.
>
> Q10. axis scaling between different tasks for Fig 8, esp panel d, is misleading.
> >We thank the reviewer’s advice. We have corrected it in the updated paper.
>
> >[6] Buzsáki, György. “Neural Syntax: Cell Assemblies, Synapsembles, and Readers.” Neuron 68 (2010).
> >
> >[7] Huang, Zirui. “Temporospatial Nestedness in Consciousness: An Updated Perspective on the Temporospatial Theory of Consciousness.” Entropy 25 (2023).

---

> ### Author Response · Authors · 2023-11-21
> **Response to the reviewer JZAs (Part 6/7)**
>
> Q11. What is the downstream decoder? Since the entire image is represented partially through time, how is the parsing score computed? Is it aggregated / smoothed over some time window? If so, how does one choose this window and is it realistic for a decoder in the brain to perform this?
>
> >The downstream decoder could be any integrators[8] that sensitive to either precise synchrony (decode single part/whole representation) or longer-scale trajectories (decode the entire parsing relationship). Such integrator is very common in the brain and can be realized as either single neuron or a population of neurons [8]. As shown above, the decoder to readout the pattern can be flexibly realized with the time window realistic in the brain (up to tens of millisecond if we take each timestep as 1 ms in the brain). It is notable that a decoder to readout the synchrony pattern (e.g. by a template) is totally different from computing a parsing score.
>
> >We detailed how parsing score is defined in **Appendix A.4.4** and it is an important challenge that we solve and an important contribution we made to the field. The window for evaluating parsing score is not limited as long as it includes all relevant assemblies at least once (40-time steps to 60-time steps will suffice for SHOPs dataset, see Figure 30 in the updated Appendix). We do not need to '**smooth**' the spike train for evaluation. The parameter for evaluation is very flexible. Evaluating parsing score do not need to have its neural correlates in the brain, because it is simply the evaluation of the model.
>
> Q12. The study does not really acknowledge any limitations, in particular how much the proposed model deviates from its stated goal of “bio-realistic” implementation. The most obvious example is that the spiking is quite artificial (as explained above). Similarly, the lack of recurrent activity between neurons within a SCS layer seems very implausible, compared to cortical assemblies in the real brain. In this work, the assemblies arise due to the DAE forcing them into metastable attractors (I think?), whereas cortical assemblies are typically formed via connectivity, especially to and from inhibitory neurons.
>
> >Actually, we provide detailed discussion of **limitations in Appendix A.1** and provides detailed discussion of **limitation on bio-plausibility in Appendix A.9 (16,17,18)**, which exactly includes the concerns of the reviewer. As argued in Q5, the limitations the reviewer pointed out is actually reasonable simplifications instead of fundamental limitations of the architecture that really harm the framework. The bio-plausibility of our model is not just on implementational level (spiking neuron, recurrent dynamics) as the reviewer focused on, but representational level (assembly code, synchrony code), dynamics level (nested rhythm) and computational level (nested rhythm for nested neural syntax[6]). See Appendix A.9 for more evidences. Lastly, metastable states forced by top-down modulation (mimicked by DAE) is itself a **promising mechanism** in neural systems[9]. After all, the DAE modulation is itself an emergent property instead of being hard-coded. We hope the reviewer captures the overall picture of the model, eg. how neuronal coherence and top-down attention promote each other from randomness to convergence during the iteration.
>
> Q13. The introduction raises several points that existing models fail at, e.g., “the parse tree could **switch among multiple reasonable forms** even given a single scene”, i.e., “correct” parsing is context-dependent. While this is true, the proposed model also does not deal with this (or does it?)
>
> >We appreciate the insightful question the reviewer asked, since multi-stability is an essential aspect that distinguishes our model with others (eg. feedforward network by supervised training). The multi-stable property is inherent for a stochastic recurrent dynamical system, and our model is one such system. Actually, **as shown in Figure 6(c)** in the main text, the parsing representation changes from the middle phase to the final phase. Specifically, the order of the leaves of the ‘green’ parse tree **inversed**. Similar phenomenon is very common in our model due to its dynamical nature. Therefore, in principle, the framework proposed in this paper could support the switch of forms of the parsing tree in general cases.
>
> >[8] Khona, Mikail and Ila Rani Fiete. “Attractor and integrator networks in the brain.” Nature Reviews Neuroscience 23 (2021): 744 - 766.
> >
> >[9] Andreas Karl Engel, Pascal Fries, and Wolf Singer. Dynamic predictions: Oscillations and synchrony in top–down processing. Nature Reviews Neuroscience, 2:704–716, 2001.

---

> ### Author Response · Authors · 2023-11-21
> **Response to the reviewer JZAs (Part 7/7)**
>
> Q14. The proposed relationships to the canonical cortical microcircuit, e.g., Figure 2g, doesn’t really add much value, in my opinion
>
> >We are sorry that we have to remove more detailed explanation of bio-relations into the Appendix A.9 due to limited space in the main text, which confuses the reviewer about the motivation of Figure 2g. Actually, the model is highly inspired by cortical circuits, though with simplifications to bridge with machine learning field. Please see Appendix A.9 (section1,2,3,4,5,6,8,9)
>
> Q15. **Does the model even need the timescale hierarchy?** At its core, each layer simply has assemblies that represent some specific part of the image, and it seems like persistent firing with the different assemblies that temporally coincide is sufficient for solving this task
>
> >The author asked a very insightful question. **The answer is yes**. If only persistent firing are considered, then there would be the binding problem [10]. In other words, at any level, we do not know what is an object is at all! We cannot group features into objects (tree node) because grouping information is provided by synchrony in time dimension.
> >>- Imagine that the sets of neurons representing different nodes fire at the same time (persistent firing), how can we distinguish which subset of neurons constructs a tree node?
> >
> >Without well-defined node representation, tree structure can not be specified at the first place. It is exactly the insight of our representation framework and can be considered as one of the contributions to the field.
>
> Q16. in-line citations are not bracketed, which is very distracting.
>
> >We are sorry about the confusion caused by the formatting issue, we have corrected that in the updated paper.
>
> Q17. figure text is way too small, e.g., Fig 5c & 8a legend box is illegible; similarly, very hard to tell different colors of spikes in 7c.
>
> >We have corrected the issues in the updated paper.
>
> Q18. Some citation on the formation of, and computing via neuronal assemblies may be appropriate (e.g., L. Mazzocato, J Gjorgjieva, etc.)
>
> >We thank the reviewer’s advice and we have added those in the main text of updated paper. Besides, Mazzocato’s recent work on metastable states was included in Appendix A.9 “Meta-stability of cortical network” section. And recent work by Gjorgjieva has been added to Appendix A.9 “Neuronal assembly as code words” section.
>
> >[10] Malsburg, Christoph von der. “The What and Why of Binding: Review The Modeler's Perspective.” (1999).

---

> > ### Comment · Reviewer_JZAs · 2023-12-05
> >
> > Many, many thanks to the authors for their extensive response to my questions and concerns. Altogether, I think it's a nice paper that attempts to address a very big and important problem in neuroscience, but ultimately "only" presents a "proof of principle" that tries to tie together a feasible solution to a benchmark toy problem and limited bio-plausibility. I say "only" because I don't necessarily think this problem needs to be solved in one go in an ICLR conference paper. Nevertheless, I don't feel confident enough to argue for its acceptance.

---

### Official Review · Reviewer_FiYe · 2023-11-01

**Soundness:** 1 poor
**Presentation:** 1 poor
**Contribution:** 3 good
**Rating:** 3
**Confidence:** 3

**Summary:**

The authors propose a spiking neural network to parse images into parts and wholes. A denoising auto-encoder is trained on single-object images. This auto-encoder is integrated into a hierarchical network with both bottom-up and top-down connections. Running the network forward eventually leads it to converge to oscillatory dynamics which are indicative of the whole-parts relationship in the image.

**Strengths:**

This work addresses an important question in the literature, for which much ink has been spilled: how does binding work? From the Gestalt psychologists to Singer and Buszaki to Hinton and his capsule networks, this question has been identified as both interesting and very hard to pin down. The authors do a good job of framing the problem (Figures 1 and 2). They assemble some reasonable toy datasets and show that it works on those. The evaluation using Victor-Purpura spike train metrics is interesting.

**Weaknesses:**

I found this paper really hard to read past the introduction, and I cannot identify its main technical contribution. They detail the training method in the 30+ page appendix (unacceptably long) and I cannot figure out why it should do any useful work given that it's so simple. I cannot convince myself that the network does something useful; the benchmarks are very easy, and I don't understand why their baseline gets a score of 0.

The citations are badly formatted and there are many awkward phrasings. It needs proofreading–automated tools like Grammarly or ChatGPT would suffice.

**Questions:**

How is the network trained? It seems like a DAE is trained in isolation and that network is assembled out of the pieces of that network. I don't understand why the network weights don't need to be adjusted once embedded in a spiking recurrent neural network. This info needs to be in the main text.

---

> ### Author Response · Authors · 2023-11-21
> **Response to the reviewer FiYe (part 1/4)**
>
> >We thank the reviewer for appreciating the motivation of the work and the attempt this paper made from framing the question to evaluation, which we also regard as the contribution of the paper. We apologize for any unclearness that confused the reviewer and we would like to provide a detailed explanation to resolve the concerns.
> ---
> Q1. Hard to read past the introduction:
> >We apologize for the unclearness that caused the confusion to the reviewer. Here we would like to briefly outline the basic idea and insights of the paper to resolve the concerns.
> >
> > 1. **How to represent hierarchical relation in neural network?** As argued in [1], nested spatial temporal activity in the brain is likely to be used to represent information of nested relations, so called neural syntax [1] (eg. Word-Phrase-Sentence is represented by neural activity of different spatial temporal timescales[1]). Part-whole hierarchy is one of such neural syntax in vision domain and therefore we aim to represent it also with nested spatial temporal neuronal activities.
> >
> >1. But **how to realize that in a neural network**? Is it just a magic? To represent a ‘tree’, we firstly need node representation and somehow connect the nodes into a tree. That is where different levels, similar column structure at each level, and connection between levels comes in. The insights are that
> >>- different levels are needed to represent nodes of different levels.
> >>- nodes at each level can be formed by iterative interaction between a SNN and a DAE, and be represented as alternating synchronized neuronal groups.
> >>- different levels should have different time-scales and different priors of ‘what an object should look like’.
> >>- gating between levels may help to coordinate the nodes into nested structures.
> >>
> >>All these insights are shown in the Figure2 in the main text. Here, we stress that the parsing ability and grouping ability should be acquired in an unsupervised manner, without explicit annotations (eg. Labels of bounding boxes [2,3]).
> >
> > 1. Lastly, even if the parsing representation emerges in a neural network, it could either be very salient (perfect synchrony and perfect nestedness) or very weak (not well-structured). **How should we measure it quantitatively?** The insight is that we could regard grouping or parsing or synchrony as a coherence state so that we could measure it through coherence measure like Silhouette Score (equipped with proper metric of spike trains). Details of evaluations are shown in A.4 in Appendix. In total, we develop a minimal but complete story of how part-whole hierarchy can be represented and measured in neural networks.
>
> >[1] Buzsáki, György. “Neural Syntax: Cell Assemblies, Synapsembles, and Readers.” Neuron 68 (2010): 362-385.
> >
> >[2] Kirillov, Alexander et al. “Segment Anything.” ArXiv abs/2304.02643 (2023): n. pag.
> >
> >[3] Girshick, Ross B. et al. “Rich Feature Hierarchies for Accurate Object Detection and Semantic Segmentation.” 2014 IEEE Conference on Computer Vision and Pattern Recognition (2013): 580-587.

---

> ### Author Response · Authors · 2023-11-21
> **Response to the reviewer FiYe (part 2/4)**
>
> Q2: Main contributions:
> >As briefly summarized at the end of the introduction section, the contribution is three-fold:
> >1. We develop a **representation framework** to represent part-whole hierarchy that is both self-consistent and bio-plausible, which is missing in most other works in this domain. It is notable that a clear representation framework is the pre-requisite to really conquer the problem and to evaluate the representation.
> >
> >1. We develop **COMPOSER** to solve the problem, which itself is a contribution given that the part-whole hierarchy is a very challenging problem for neural networks. Besides, there are several other sub-contributions which can inspire readers in related fields:
> >>-    The model is highly brain-inspired and bio-plausible (explained in Appendix A.9).
> >>
> >>-   The model provides a potential direction to combine artificial neural network (denoising autoencoder or its more advanced variants like diffusion model or just general generative models) and spiking neural network (spike coding, hierarchical rhythmic dynamics, dendritic computation, various biological constraints like time-scale hierarchy, delay-coupling. See Appendix A.9). The insight is that spiking representation/dynamics + generative attention (DAE) + hierarchical organization is a way to solve the problem.
> >>
> >>-   The representation in the model is highly interpretable and consistent with neuroscience findings (assembly code, nested oscillation).
> >>
> >>-    The dynamical system viewpoint could contribute to a list of more theoretical works like Hopfield network and energy-based models. For how to construct many associative memories as sequence / cycle attractors.
> >
> >1. We develop datasets and metrics to **explicitly evaluate** the part-whole hierarchy, which itself is a challenge (See Appendix A.4 for detailed explanation). For example, how a parsing structure that potentially has different coherence levels could be measured based on neuronal activity? Note that the evaluation is not limited to spiking representation but can be generalized to general cases in ANNs (See Appendix A.4.7, A.5.4)
> >
> >Taken together, the overall contribution is that we developed **a systematic framework** from framing the question at representation level to modeling and to evaluation, which is both relevant to machine learning field and neuroscience field. Such a systematic framework should be essential to solve the problem of part-whole hierarchy. Potential future works are discussed in A.1 in Appendix.
>
> Q3: Training method:
> >Please see A.7.3 and Figure 18 for **training dataset**.
> >
> >Please see A.7.4 and eq.27 for the **training loss**.
> >
> >The training pipeline is very simple and generalizable. We discussed the effect of training on the parsing performance in A.10.3 and Figure 19. The assumption behind the training pipeline is that, to parse the scene, the neural system should have certain priors of ‘what an object should look like’ and ‘what a part should look like’. These priors should be acquired from evolution and implemented as pre-configured or **innate connections** for the brain[1] and is acquired from **pre-training** the DAE of part (whole) levels to denoise single part (whole) objects. Note that the self-supervised pre-training only provide necessary priors and **do not introduce any explicit supervision** of grouping or parsing. Thus, it is the dynamics of the model that really parse the scene.
> >
> >
> >Once the DAE is pre-trained on each training dataset, the model has acquired all the ‘needed prior’ information and just use that information to dynamically parse visual scenes. Therefore, it is not a must to further train the DAE during the parsing process (“once embedded in a spiking recurrent neural network”). The overall picture is explained in Figure2 in the main text. Still, we thank the reviewer’s advice and we have clarified more clearly in main text.
> >
> >
> >Further, we have discussed **the limitation** of the current learning scheme in Appendix A.1 (“Learning Scheme” section) and provided a more ambitious picture of how to replace the pre-training by end-to-end training as **future work**. It is notable that this work provides a foundation of how the biological constraints work together (integration window, refractory period, delay-coupling, hierarchical timescales…) and possibly serve as **inductive biases** to bias the fully-unsupervised learning of parsing in the future. Similar strategies of extending pre-training model to unsupervised training models can be seen in [2,3]
>
> >[1] Stöckl, Christoph et al. “Structure induces computational function in networks with diverse types of spiking neurons.” bioRxiv (2022).
> >
> >[2] Hao Zheng et al. Dance of snn and ann: Solving binding problem by combining spike timing and reconstructive attention. Neurips 2022.
> >
> >[3] Hao Zheng et al. GUST: Combinatorial Generalization by Unsupervised Grouping with Neuronal Coherence. Neurips 2023.

---

> ### Author Response · Authors · 2023-11-21
> **Response to the reviewer FiYe (part 3/4)**
>
> Q4: I cannot figure out why it should do any useful work given that it's (the loss) so simple.
> >Denoising loss (or reconstruction loss) is a classical type of loss in representation learning, which is widely applied in popular models like Bert, Diffusion model, VAE and so on.  Therefore, the denoising loss suffice to learn prior knowledge (explained for Q3) and desirable for self-supervised pre-training because it does not need any annotation. In this paper, we focus on general parsing instead of the ‘use’ of parsing for downstream tasks. To do useful work, **additional loss terms**, eg. supervised loss for classification, can be flexibly included and **additional readout head** can be added (to latent space of DAE/ or to SCS) to serve downstream tasks. Actually, it is a natural logic of a line of research on object-centric representations (e.g. Agglomerator [2] and Slot Attention [7]) to “use” the representation. In short, additional loss terms and modules can be added to serve downstream tasks. And the simple denoising loss is not an intrinsic limitation of the model.
>
> Q5: I cannot convince myself that the network does something useful
> >We understand the concern of the reviewer and we would like to explain in several aspects.
> >
> >- Firstly, everything starts from representation. How to represent (and evaluate) part-whole hierarchy with neuronal activity in pure neural networks is currently **a theoretical problem** in the representation learning field [1] instead of an application problem. And we aim to provide a theoretical framework for possible solutions. As argued in [2], such representation facilitates the interpretability of the network representation and promote human-like vision. Related motivations are nicely reviewed in [1].
> >
> >- Secondly, while the datasets are composed of relatively simple objects, so that classification of objects can be easy, the representing part-whole **relation** is not as easy as argued in the main text and [1]. For example, imagine how an image could be dynamically decomposed into parts and wholes and recomposed into a structured tree in neural network? As far as we know, our model is the first model to realize meaningful part-whole hierarchy in pure neural networks.
> >
> >- Thirdly, simple architecture and simple loss function do not necessarily make a model useless. For example, Hopfield network has simple structure and simple energy function (loss), but is very useful in various fields. Actually, minimal implementation is desirable as a starting point for further generalization. It is more important what insight a simple architecture brings (eg. Figure1,2 in the main text and how DAE and SNN can work together, and how time-scale hierarchy potentially serves as inductive bias).
> >
> >- Fourthly, in A.1 in the Appendix, we discuss how the model could be generalized to account for diverse cases. For example, what the emerged representation is used for (eg. classification) can depend on downstream modules and related supervised loss, as shown in [2]. In this paper, we focus on developing a basic model serving as theoretical foundation to various future works.
> >
> >- Fifthly, there is an interesting comparable tendency in object-centric representation literature of CV, from ‘seemingly useless’ simple model[3] (evaluated in binary simple datasets, pre-trained with simple loss, simple network) to more and more useful complex models[4,5,6,7] (evaluated on more complex datasets, with more engineering works and improvements). We hope such comparison helps the reviewer to see current simple model as a promising starting point of a list of future works. For example, the capability of the model may increase if the diffusion model replaces the DAE.
> >
> >- Lastly, self-supervised object-centric representation (even without hierarchical relationship) is mostly limited on synthetic datasets[3,4,5,6,7]. Therefore, **the concern of the reviewer is actually a common limitation of the domain** [3,4,5,6,7], not just of our work.
>
> >[1] Hinton, Geoffrey E.. “How to Represent Part-Whole Hierarchies in a Neural Network.” Neural Computation 35 (2021): 413-452.
> >
> >[2] Nicola Garau et al. Interpretable part-whole hierarchies and conceptual-semantic relationships in neural networks. 2022 IEEE/CVF Conference on CVPR, 2022.
> >
> >[3] Greff, Klaus et al. “Binding via Reconstruction Clustering.” ICLR (2015).
> >
> >[4] Greff, Klaus et al. “Tagger: Deep Unsupervised Perceptual Grouping.” Neurips (2016).
> >
> >[5] Greff, Klaus et al. “Neural Expectation Maximization.” Neurips (2017).
> >
> >[6] Greff, Klaus et al. “Multi-Object Representation Learning with Iterative Variational Inference.” ICML (2019)
> >
> >[7] Locatello, Francesco et al. Object-Centric Learning with Slot Attention. Neurips 2020.

---

> ### Author Response · Authors · 2023-11-21
> **Response to the reviewer FiYe (part 4/4)**
>
> Q6. Why baseline gets a score of 0.
> >We provided the explanation in A.5.7 in Appendix.  In short, it is because
> >
> >- the dataset we made, while composed of simple objects, is still a challenging task for representing part-whole hierarchy (**it is the relationship that really matters**).
> >
> >- Further, in binary images, objects or parts may be harder to be distinguished since **color does not differ** among objects or between parts/wholes.
> >
> >- Also, as stressed in the main text, while many related works attempted / claimed to achieve part-whole representation, they do **not** provide clear representation framework or explicit **quantitative evaluation** method. Therefore, it is questionable whether they really capable of parsing. The baseline is one such case. It is why we insist that proposing the explicit parsing dataset and evaluation metric are important contributions to the field.

---

### Official Review · Reviewer_denz · 2023-11-03

**Soundness:** 3 good
**Presentation:** 2 fair
**Contribution:** 3 good
**Rating:** 5
**Confidence:** 3

**Summary:**

The authors propose COMPOSER, a new framework for modeling part-whole hierarchical object representations in neural networks. Composer is an architecture obtained by combining aspects of deep learning (denoising autoencoders trained in a self-supervised manner) and computational neuroscience (spike timing, synchrony, layered-organization in the neocortex, etc) trained it in a self-supervised manner. The authors propose a suite of benchmarks to evaluate the ability of models to parse a visual scene into part-whole hierarchies of objects present in the scene, and propose metrics corresponding to this evaluation. The authors show evaluations on the proposed datasets both qualitatively and quantitatively showing that neurons in Composer show emergent part-whole grouping, with lower-level neurons encode part- features and higher-level neurons encode whole- features.

**Strengths:**

- Representing part-whole hierarchy in neural representations is a challenging open problem that both machine learning and cognitive science researchers find interesting. The paper proposes a complete package to model and evaluate part-whole parsing in neural networks.
- The proposed datasets and corresponding metrics are a useful contribution to further research in this domain. That being said, the proposed metrics are suitable only for evaluating spiking neural network models of scene parsing (authors, please correct me if I'm wrong in my understanding here with more elaboration on how to evaluate non spiking networks too)
- I appreciate that authors share the code to reproduce the results presented here.

**Weaknesses:**

- Lack of ablations: Composer is quite a complicated architecture and has many moving parts to it. The authors haven't justified how they arrived at this architecture, and if all its components are useful for the downstream task. They need to ablate individual components and report the value of each of them to justify such a complicated architecture.
- Metrics used aren't explained very clearly: The authors could further explain part-score and whole-score more clearly, maybe with visualizations of part- and whole- segmentation with low and high part- / whole- scores to make the readers understand the used metrics. Without a clear understanding of what these metrics are doing, many figures and visualizations aren't accessible.
- Writing: It feels like the authors have tried to compress a lot of important information into the main text (and appendix) as a result of which clarity of the writing has dropped. There is some inspiration from neuroscience in the spiking and synchrony modeled in Composer, however I felt the description of composer to be unnecessarily heavy on neuroscience-oriented jargon.
E.g. 1) superficial spike-coding space, what does 'superficial' stand for here? 2) what does 'abstract' refractory period mean? did you mean to say absolute refractory period? 3) I didn't find it necessary to mention that neurons in the architecture are pyramidal neurons unless there is some anatomical/computational inspiration of pyramidal neurons in composer.
- (nit) Formatting issues: I believe many citations are in the wrong format, quotes are wrongly formatted (two close quotes used in place of open and close quotes), multiple spelling errors (Cotical -> cortical in the first mention of Composer's full form) need to be corrected.

**Questions:**

Please refer to my main review's weaknesses section for questions.

---

> ### Author Response · Authors · 2023-11-21
> **Response to Reviewer denz (part 1/2)**
>
> We thank the reviewer for the careful reading and insightful questions. We appreciate that the reviewer accurately captured the motivation of the work and view the work as a complete package including framing the question, modeling, and evaluation. We apologize for any confusion caused by the writing and formatting issues. We have corrected those issues in the updated paper. Here, we would like to response to the reviewer’s concerns one-by-one.
>
> ---
>
> Q1: “That being said, the proposed metrics are suitable only for evaluating spiking neural network models of scene parsing”
> >We appreciate that the reviewer acknowledges the contribution of the proposed datasets and corresponding metrics to future researches in the domain.
> >
> >Actually, the dataset and evaluation are not necessarily limited to spiking neural network. On the one side, the binary dataset could be used as input for either SNN [1] or ANNs [2].
> >
> >On the other side, the metric we proposed for evaluating grouping (part-score, whole-score) and parsing (parse-score) is generally based on the coherence measure of clusters (Silhouette Score), which, as far as we know, is the first explicit evaluation of part-whole relationship in neural network. Whether it is used to evaluate spiking representation (SNN) or real-valued vector representation (ANN) is based on the “distence metric”. While VP-metric is used to measure distance between spike trains, many other metrics can be used to measure distance between real-valued vectors (eg. L2 norm). For example, the metric is directly **generalized to measure the parsing representation in the Agglomerator** for benchmarking (the model is briefly introduced in A.4.5.4), which is ANN instead of SNN. We have provided detailed discussion of metric and how the metric can be generalized in Appendix (A.4.3 and A.4.4 for understanding the metric. A.4.7 and **A.5.4** for generalizing the metric). To be more specific, the metric could generally be used to evaluate the idea of representing the part-whole relationship by the similarity measure (eg. synchrony in Composer or islands of identical vectors in Agglomerator) in pure neural networks. In short, **yes, the 3 metrics are not limited to spiking neural network**.
>
> Q2:” Lack of ablations.
> >We have provided the ablation study in Figure 10 in the main text and Figure 19, Figure 20 in the Appendix (Copied in **A.11** of updated appendix). Generally, the model is composed of (two) spiking modules and (two) DAE modules. So that the ablation is made for both. To (continuously) ablate DAE, we trained DAE to have different denoising abilities (by setting different learning rate). Since spiking module has different parameters, so we ablated it by perturbing each parameter.
>
> >To be specific, the Figure 10 (a) and Figure 19 show how continuously **degrading the DAE** linearly harm the parsing performance of the entire model, so that DAE modules (both part/whole level) are essential for the model.
>
> >The Figure 10 (b) and Figure 20 show how **perturbing the spiking layer** and some other parameters degrades the parsing ability of the model. For example, setting $\tau_d=0$ accounts for removing the delay coupling and setting $\tau_1=0$ accounts for removing the coincidence detector and setting $g=0$ accounts for removing the relative refractory period. We provide related explanation in the main text and appendix (A.6.3 and A.10.3).
>
> >To explain how we arrived at the seemingly complicated architecture as a system step by step, we provide the **dynamical system viewpoint of the model** in the main text (Figure 2). For example, how large number of candidate assemblies at each level becomes transient attractors of the denoising dynamics formed by ‘DAE+recurrence+refractory+delay’ and how different level assemblies nested with each other by ‘attentional gating’.
>
> >Besides, to further resolve the concern of the reviewer, and as suggested by another reviewer, we add more ablation study in the updated Appendix (A.11 or **Figure 30**). For example, we further showed the role of the skip connection (Figure 30(i)). The result indicates that the model is relatively robust w.r.t the specific parameter tuning but the removal of those part would be harmful to the model, to different extent.
>
> >[1] Zheng, Hao et al. “Dance of SNN and ANN: Solving binding problem by combining spike timing and reconstructive attention.” ArXiv abs/2211.06027 (2022): n. pag.
>
> >[2] Greff, Klaus et al. “Binding via Reconstruction Clustering.” ArXiv abs/1511.06418 (2015)

---

> ### Author Response · Authors · 2023-11-21
> **Response to Reviewer denz (part 2/2)**
>
> Q3: Metrics used aren't explained very clearly:
>
> >We provided very detailed explanation of metric in **Appendix A.4** (A.4.1 to A.4.7, especially Figure 15 to Figure 18). We apologize that we found it very difficult to compress the whole explanation into the main text, so we firstly provided an intuitive explanation by experiment in the main text (Figure 5) and then expanded the whole story in the Appendix (exactly as the reviewer suggested). Figure5 is more clearly shown in **Appendix Figure 15 to Figure 18**, which are the complete visualizations the reviewer suggested. Maybe it is the unclearness of Figure 5 (tiny label and legend) in main text that caused the confusion and we have updated a more clear Figure.5 in the updated paper. By the way, the parsing score is the most challenging one when we developed the evaluation metric, because it is to measure the cross-level coherence in a hierarchical structure. However, most related measures are only for single-levels, instead of cross-level. Therefore, the idea of measuring the parsing by coherence measure (Silhouette score) could potentially contribute to this field.
>
> Q4: Concerns on neuroscience-oriented words.
>
> >1. **Superficial** spike coding space (SCS) is in comparison with the more abstract representation in the latent space of DAE. ‘Superficial’ stands for that the neurons in SCS is mapped to the image in pixel-wise manner, without abstraction or dimension reduction. In “neuroscience jargon”, they are topographically mapped to the external word. We use this term to imply a picture that <1> SCS and <2> latent space of DAE represent features at different ‘feature levels’ as the <1’> layer2/3 and <2’> layer 5/6 in the cortex (explained in Appendix A.9 point 4,6).
> >
> > 1. **abstract refractory** should be ‘absolute refractory’: Thank you for pointing out the typo.
> >
> > 1.  **Pyramidal cell** is relevant to the spiking neuron in our model in several senses that we explained in Appendix (please see A.9 point 8,9). In sum, anatomically and computationally, pyramidal cell has dendritic computation of driving input (at proximal site) and modulatory input (at distal site) (Spruston 2008) and serve as coincidence detectors that has small integration window and low threshold (Abeles 1982). These details are integrated in the model in a simplified manner and have essential computational roles. Besides, pyramidal cells are more dominant in ‘superficial layers 2/3’ than in deeper layers 5/6.
>
> >While the model is minimally implemented in the current paper, the coding scheme, architecture, dynamics, are inspired by and consistent with a list of facts or theories in neuroscience, which are thoroughly explained in **Appendix A.9**. We thank the reviewer’s response and we will find ways to balance the clarity and neuroscience inspiration better.
>
> Q5: Formatting issues:
>
> >Thank you for pointing out the issues and we have corrected them in the updated paper.

---

> > ### Comment · Reviewer_denz · 2023-11-30
> > **Thank you for your response to our reviews**
> >
> > Dear authors,
> >
> > Thank you for your response to our reviews. I have looked at the other reviewers comments as well, and it looks like all reviewers unanimously find the working of this model difficult to understand. It doesn't help that important details are explained in several pages of appendix. I see that the ablations are found in Appendix A.10.4, which is still very hard to follow unfortunately with references back to the main text and other parts of the Appendix. I believe the authors can work on clearly describing their work in a more clearly readable format and at this stage, I do not recommend acceptance of the work at ICLR.
> >
> > Thank you.